# Phase separation of initiation hubs on cargo is a trigger switch for selective autophagy

Autophagy is a key cellular quality control mechanism. Nutrient stress triggers bulk autophagy, which nonselectively degrades cytoplasmic material upon formation and liquid–liquid phase separation of the autophagy-related gene 1 (*Atg1*) complex. In contrast, selective autophagy eliminates protein aggregates, damaged organelles and other cargoes that are targeted by an autophagy receptor. Phase separation of cargo has been observed, but its regulation and impact on selective autophagy are poorly understood. Here, we find that key autophagy biogenesis factors phase separate into initiation hubs at cargo surfaces in yeast, subsequently maturing into sites that drive phagophore nucleation. This phase separation is dependent on multivalent, low-affinity interactions between autophagy receptors and cargo, creating a dynamic cargo surface. Notably, high-affinity interactions between autophagy receptors and cargo complexes block initiation hub formation and autophagy progression. Using these principles, we converted the mammalian reovirus nonstructural protein μNS, which accumulates as particles in the yeast cytoplasm that are not degraded, into a neo-cargo that is degraded by selective autophagy. We show that initiation hubs also form on the surface of different cargoes in human cells and are key to establish the connection to the endoplasmic reticulum, where the phagophore assembly site is formed to initiate phagophore biogenesis. Overall, our findings suggest that regulated phase separation underscores the initiation of both bulk and selective autophagy in evolutionarily diverse organisms.

Autophagy is a highly versatile cellular degradation pathway that targets diverse components, ranging from membrane-bound organelles to distinct protein complexes. Bulk autophagy, also known as nonselective autophagy, involves the nonspecific engulfment and degradation of cytoplasmic material within autophagosomes. Starvation induces the formation of the autophagy-initiating Atg1 complex[1,2]. Under nutrient-rich conditions, autophagy and Atg1 complex formation are inhibited due to target of rapamycin complex 1 (TORC1)-mediated phosphorylation of Atg13, a member of the Atg1 complex[3]. In contrast, starvation inhibits TORC1, promotes Atg1 complex assembly and activates the Atg1 kinase[4–6]. The Atg1 complex undergoes liquid–liquid phase separation and the resulting condensates are anchored to the vacuolar membrane via Atg13–Vac8 interactions[7–10]. This process

ultimately builds the phagophore assembly site (PAS), which is key for phagophore initiation[8]. Thus, phagophore formation occurs without a templating cargo[11].

Unlike bulk autophagy, selective autophagy relies on autophagy receptors that recognize and bind to specific cargo, which templates membrane formation and thereby facilitates its exclusive packaging into autophagosomes for subsequent degradation. Before their degradation through selective autophagy, a variety of cargoes undergo phase separation[12]. In budding yeast, for example, dodecamers of the aminopeptidase 1 (Ape1) proteins organize into a complex via phase separation, which is transported to the vacuole through a selective autophagy process known as the cytoplasm-to-vacuole targeting (Cvt) pathway. Amino acid substitutions in Ape1 that hinder assembly also impede the Cvt

✉e-mail: florian.wilfling@biophys.mpg.de; kraft@biochemie.uni-freiburg.de

pathway, suggesting that cargo liquidity is a critical determinant for selective autophagic targeting[13]. Similarly, phase separation drives the formation of p62 (also known as SQSTM1) condensates and PGL granules, which both are selective autophagy cargoes[14–16]. Autophagy receptors link their cargo to Atg8 on the growing autophagosome membrane, thus, phagophore formation in selective autophagy requires a templating cargo[17]. In addition, cargo-bound autophagy receptors interact with the adaptor and scaffolding protein Atg11 to activate the Atg1 kinase[18–20].

We recently discovered that the endocytic protein Ede1 functions as a selective autophagy receptor in budding yeast[21]. When endocytosis is impaired, Ede1 directly links early clathrin-mediated endocytosis proteins to Atg8 on the autophagosomal membrane. These Ede1-dependent endocytic protein deposits (ENDs) are degraded via autophagy[21]. Notably, selective autophagy of ENDs depends on the phase separation of Ede1 and its autophagy receptor function. Ede1 contains a low-complexity region and coiled-coil domains, which promote the formation of higher oligomeric structures[22] and are required to form phase separations together with further END components[21,23]. In addition, Ede1 contains several binding motifs that form interactions with the autophagy machinery. Disrupting the ability of Ede1 to undergo phase separation or its receptor function abolishes the Ede1-dependent autophagy pathway for endocytic proteins.

Overall, these findings reveal an interplay between phase separation and autophagy. Nevertheless, how phase separation influences phagophore initiation in selective autophagy is incompletely understood.

## Results

### Low-affinity interactions promote receptor mobility

To address the role of phase separation in END turnover in more detail, we manipulated the END cargo properties. We generated two yeast strains expressing a double green fluorescent protein (GFP)-tagged Ede1 to visualize END condensation and degradation. In one strain, we also introduced a copper-inducible triple GFP-binding protein (3×GBP) fused to blue fluorescent protein (BFP)[24]. Treatment of these strains with copper to induce protein expression did not induce autophagy, in contrast with previous reports in mammalian cells[25] (Extended Data Fig. 1a). To induce END turnover by autophagy[21,26], we treated cells with rapamycin. Although 2×GFP–Ede1-positive ENDs were eliminated in the control strain lacking BFP–3×GBP (−3×GBP), they persisted in the strain expressing BFP–3×GBP (+3×GBP) (Fig. 1a,b and Extended Data Fig. 1b,c) and showed no major effects on the size and number of ENDs per cell before addition of rapamycin (Extended Data Fig. 1c). Of note,

fluorescence recovery after photobleaching (FRAP) revealed rapid exchange of 2×GFP–Ede1 between the bleached and nonbleached pool in the control strain but not in the strain expressing BFP–3×GBP (Fig. 1c). Mutations in 3×GBP that reduce its affinity for GFP (3×GBP^low, F103A E104R; Extended Data Fig. 1d [27]), restored END fluidity (Fig. 1c) and END turnover (Fig. 1b). This 3×GBP^low mutant still localized to ENDs, indicating that its interaction with GFP was significantly reduced but not completely abolished (Extended Data Fig. 1e). These data suggest that Ede1 within the END compartment is liquid-like in the control strain but solidified in the strain expressing BFP–3×GBP, and this change in mobility altered its autophagic degradation. Despite their solidification, END assemblies still co-localized with the autophagy protein Atg8, suggesting that the autophagy receptor properties of Ede1 were not altered (Fig. 1d and Extended Data Fig. 1f). We infer that high-affinity interactions between 2×GFP–Ede1 and BFP–3×GBP render the receptor Ede1 immobile and that receptor mobility is required for selective END degradation.

To determine whether receptor mobility is an important feature in selective autophagy more generally, we manipulated the Cvt pathway. First, we used an in vitro binding assay to examine the interaction between the cargo Ape1 and the Atg19 receptor. This assay detects the high-affinity interaction between GFP and GST–BFP–GBP, but not the low-affinity interaction between GFP and GST–BFP–GBP^low (Fig. 1e). Subsequently, we immobilized a GST–BFP-tagged Ape1 propeptide (Ape1 residues 1–45, which interacts with Atg19 (ref. 20)). After washing, we did not detect the interaction between mCherry–Atg19 and GST–BFP–Ape1^1–45. Similarly, Atg19 bound stably to a C-terminal fragment of Atg11 immobilized on Glutathione Sepharose (GSH) beads (GST–Atg11^685–1178), but showed substantially reduced binding to GST–Ape1^1–45-coated beads in pulldown experiments (Fig. 1f), consistent with a low-affinity interaction between Ape1 and Atg19. This suggests that receptor mobility in the Cvt pathway is established by low-affinity receptor–cargo interactions.

To examine the impact of cargo–receptor affinity on degradation, we coexpressed GFP–Ape1 with either untagged Atg19 or GBP-tagged Atg19 in *atg19Δ* cells. The presence of GBP is expected to change the endogenous low-affinity interaction between Atg19 and GFP–Ape1 to a high-affinity (ectopic) interaction. Notably, free GFP was generated in *atg19Δ* cells coexpressing GFP–Ape1 and untagged Atg19 but was strongly reduced in those coexpressing GFP–Ape1 with Atg19–GBP (Fig. 1g). Moreover, expression of Atg19–GBP^low restored free GFP generation, excluding effects of receptor tagging and further suggesting that the high-affinity interaction of Atg19–GBP with GFP–Ape1 inhibits its delivery to the vacuole and autophagic turnover. Notably, the single

**Fig. 1 | High-affinity receptor–cargo interactions impair selective autophagy. a**, Schematic of END condensate solidification. Dashed grey arrows show low-affinity interactions. **b**, Cells expressing 2×GFP–Ede1 without (light grey) or with (orange) BFP–3×GBP or with BFP–3×GBP^low (dark grey) under the control of a copper-inducible promoter were grown to mid-log phase in the presence of 1 mM CuSO₄ for 6 h. Autophagy was then induced by rapamycin addition. Quantification: cells with END condensates. Data are mean values (*n* > 150 cells per condition and replicate, three biological replicates). Circles show mean values of each replicate, bars show mean. Statistical analysis was carried out by two-tailed unpaired *t*-test. *P* values are as follows. Untreated: −3×GBP (--) versus +3×GBP, *P* = 0.0978; −3×GBP (--) versus +3×GBP^low, *P* = 0.5104; rapamycin: −3×GBP (--) versus +3×GBP, ****P* < 0.0001; −3×GBP (--) versus +3×GBP^low, *P* = 0.1363. **c**, The −3×GBP (−), the +3×GBP and the +3×GBP^low strains were grown to mid-log phase as described in **b**. 2×GFP–Ede1 assemblies were photobleached and recovery of the signal was monitored. White arrowheads indicate the photobleached area. Scale bar, 1 μm. Quantification: recovery of the GFP signal. Data are mean values ± s.e.m. (*n* > 26 ENDs per condition across replicates, three biological replicates). **d**, A 2×GFP–Ede1 strain without (−) and with (+) 3×GBP and a 2×GFP–Ede1^AIM (Atg8 binding mutant[21]) strain with +3×GBP coexpressing mCherry–Atg8 were grown to mid-log phase in the presence of 1 mM CuSO₄. GFP/mCherry-positive structures were quantified. Data are mean values (*n* > 100 ENDs per condition and replicate, three biological replicates). Circles show mean values

of each replicate, bars show mean. Statistical analysis was conducted by a two-tailed unpaired *t*-test. *P* values are WT(−) versus WT(+), *P* = 0.525; WT(+) versus AIM(+) ****P* < 0.0001. AIM, Atg8 binding mutant. **e**, GST–BFP, GST–BFP–GBP, GST–BFP–GBP^low and GST–BFP–Ape1^1–45 were expressed in *Escherichia coli* and bound to GSH beads. Protein-bound beads were incubated with *E. coli* cell lysates containing 6×His–GFP or mCherry–Atg19, and bound GFP or mCherry–Atg19 was analysed before and after washing. Scale bar, 20 μm. Quantification: ratio of bead-bound protein to soluble protein in a box plot. Horizontal lines show the median, box shows the 25th to 75th percentiles, whiskers indicate the 5th and 95th percentiles, circles show mean value of each replicate, outliers show black dots (*n* > 25 beads per condition across replicates, three technical replicates). Statistical analysis was carried out by a one-way analysis of variance (ANOVA) followed by Dunnett's post hoc test. *P* values: GBP versus GBP^low, ****P* < 0.0001; GBP versus Ape1^1–45, ****P* < 0.0001. **f**, GST, GST–Ape1^1–45 and GST–Atg11^685–1178 were expressed in *E. coli* and bound to GSH beads, incubated with purified recombinant Atg19^3D and analysed by immunoblotting. One experiment out of three technical replicates is shown. **g**, *atg19Δ* cells expressing GFP–Ape1 and Atg19, an empty control vector (−), Atg19–GBP, Atg19–GBP^low or Atg19–GBP^med were grown to mid-log phase. Cell extracts were analysed by immunoblotting. One out of three biological replicates is shown. Source numerical data and unprocessed blots are available in Source data.

E104 mutation (GBP^med), was not sufficient to restore autophagic Ape1 delivery to the vacuole to the extent of the wild type (Fig. 1g). Of note, wild-type Atg19 showed a mobile fraction on the Ape1 cargo surface, whereas the stabilization of Atg19 on Ape1 by GBP–GFP resulted in minimal recovery upon photobleaching, (Extended Data Fig. 1g). However, the mobility of Ape1 itself was unaffected by this stabilization (Extended Data Fig. 1h).

Together, these findings suggest that receptor mobility is a key principle of selective autophagy.

## Autophagy biogenesis factors form initiation hubs

The autophagy machinery initiates phagophore biogenesis at the cargo surface, selectively engulfing the cargo. A key step in this process is the recruitment of the scaffolding protein Atg11 to the autophagy

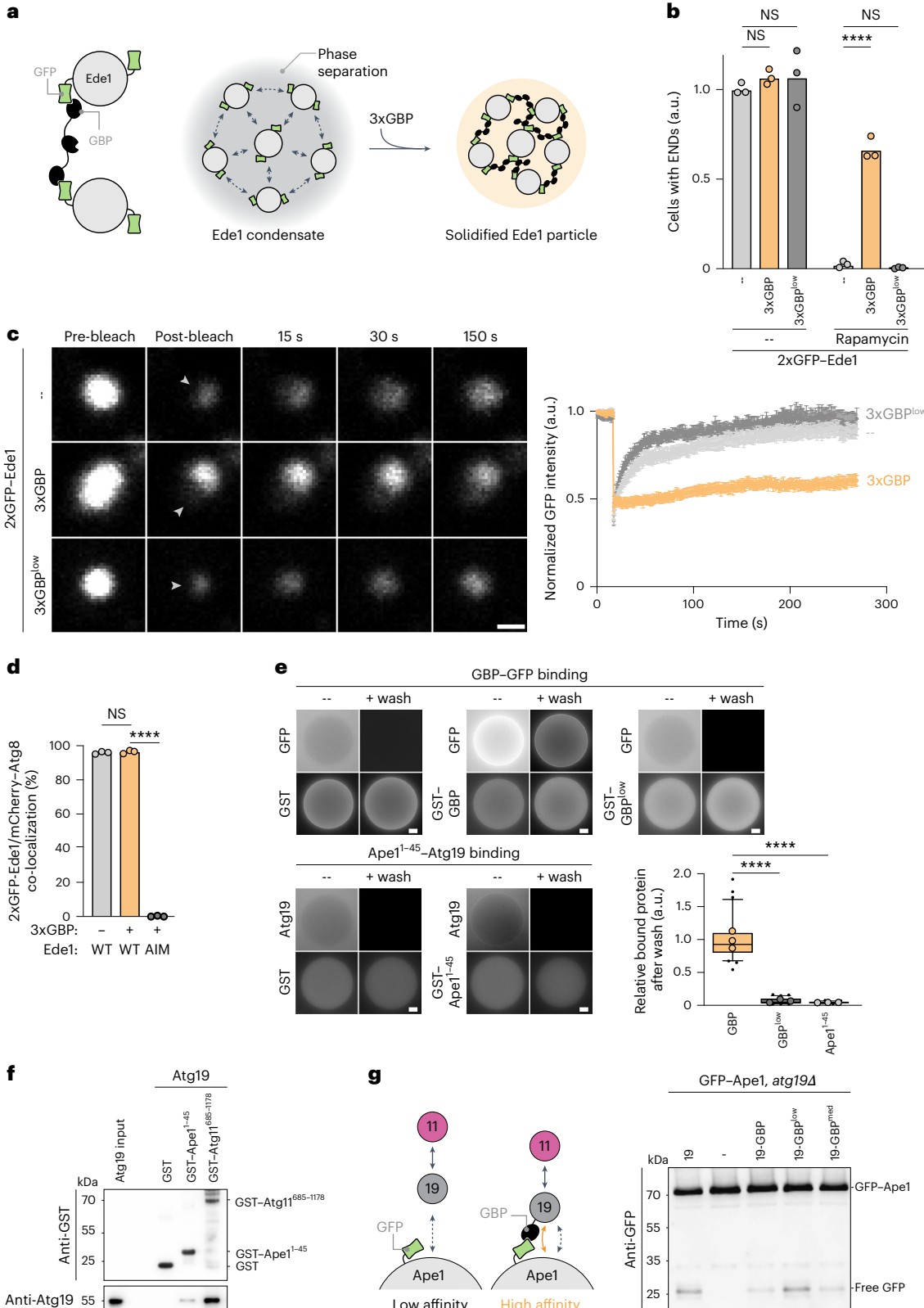

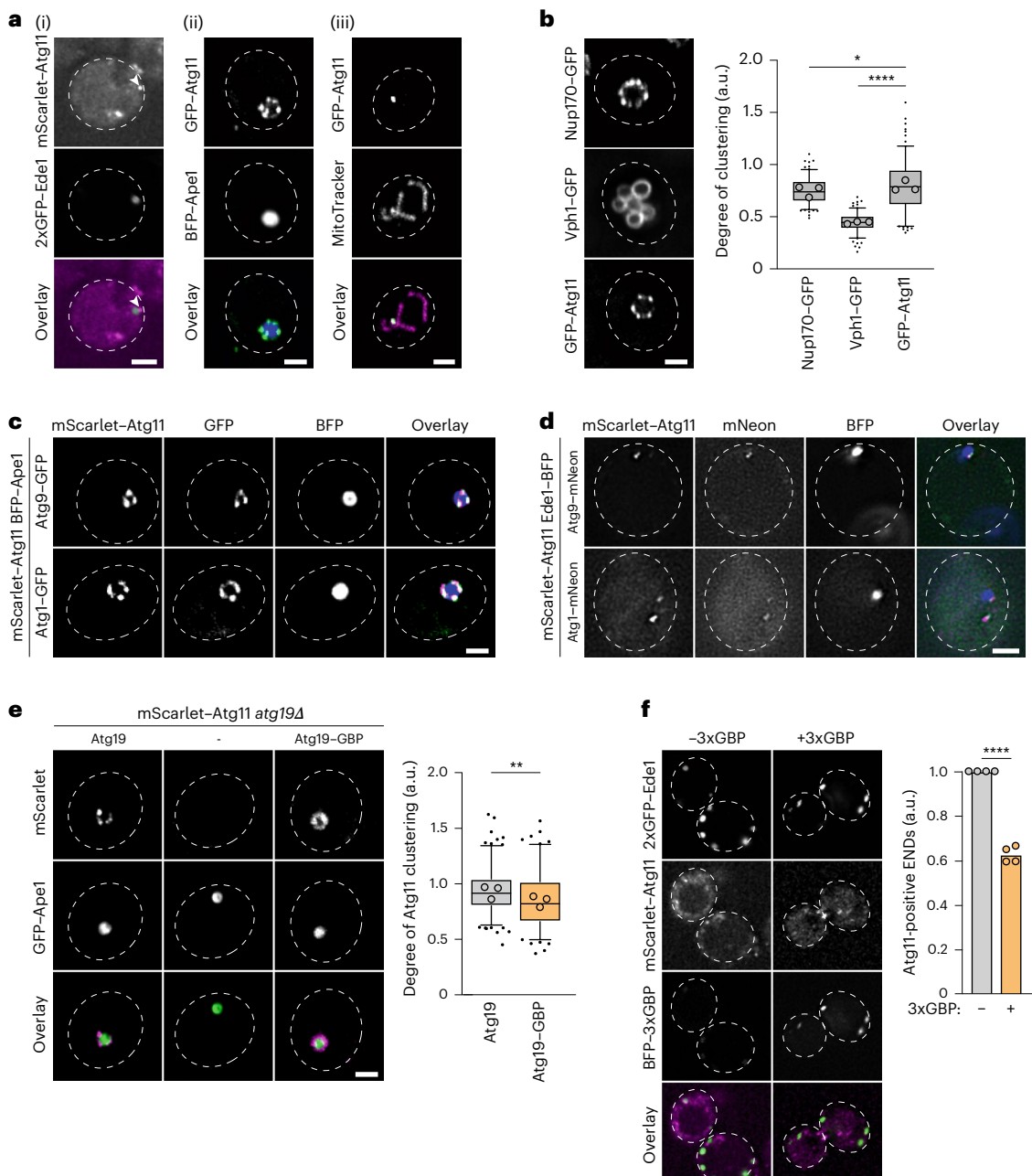

**Fig. 2 | Receptor mobility enables the formation of initiation hubs.**
**a**, (i) *atg19Δ* cells expressing 2×GFP–Ede1 and mScarlet–Atg11 were grown to mid-log phase. (ii) GFP–Atg11 cells expressing endogenous BFP–Ape1 and copper-inducible untagged Ape1 were grown to mid-log phase in the presence of 50 μM CuSO₄. (iii) *atg19Δ* cells expressing GFP–Atg11 were grown to mid-log phase. Mitochondria were stained with MitoTracker Red and mitophagy was induced by starvation. Images of one out of three biological replicates are shown. Scale bar, 2 μm. **b**, Nup170–GFP, Vph1–GFP *vac8Δ atg19Δ* and GFP–Atg11-expressing cells (as in **a**(ii)) were grown to mid-log phase. Scale bar, 2 μm. Quantification: GFP clustering as the coefficient of variance (s.d./mean GFP intensity) in a box plot. Horizontal lines show the median, box shows the 25th to 75th percentiles, whiskers show the 5th and 95th percentiles, circles show the mean value of each replicate, outliers are indicated by black dots (*n* = 50 structures per condition and replicate, three biological replicates). Statistical analysis was conducted by a one-way ANOVA followed by a Dunnett's post hoc test. *P* values: GFP–Atg11 versus Vph1–GFP, ****P* < 0.0001; GFP–Atg11 versus Nup170–GFP, **P* = 0.0439. **c**, Atg9–3×GFP *atg11Δ* or Atg1–3×GFP cells expressing mScarlet–Atg11, endogenous BFP–Ape1 and copper-inducible untagged Ape1 were grown to mid-log phase in the presence of 50 μM CuSO₄. Images of one

out of three biological replicates are shown. Scale bar, 2 μm. **d**, Cells expressing Ede1–BFP, mScarlet–Atg11 and either Atg1–mNeon or Atg9–mNeon were grown to mid-log phase. Images of one out of three biological replicates are shown. Scale bar, 2 μm. **e**, mScarlet–Atg11 *atg19Δ* cells coexpressing either Atg19, an empty control vector (−) or Atg19–GBP and endogenous GFP–Ape1 and copper-inducible untagged Ape1 were grown to mid-log phase in the presence of 50 μM CuSO₄. Scale bar, 2 μm. Quantification: mScarlet–Atg11 clustering as the coefficient of variance (s.d./mean mScarlet intensity) in a box plot. Horizontal lines show the median, box shows 25th to 75th percentiles, whiskers show the 5th and 95th percentiles, circles show the mean value of each replicate, outliers are indicated by black dots (*n* = 50 structures per condition and replicate, three biological replicates). Statistical analysis: two-tailed unpaired *t*-test. ***P* = 0.0012. **f**, *atg19Δ* cells expressing mScarlet–Atg11 and 2×GFP–Ede1 without (−) or with (+) 3×GBP under the control of a copper-inducible promoter were grown to mid-log phase. Scale bar, 5 μm. Data are mean values (*n* > 37 ENDs per condition and replicate, four biological replicates). Circles show mean values of each replicate, bars show the mean. Statistical analysis was carried out by two-tailed unpaired *t*-test. *P* value: 3×GBP(−) versus 3×GBP(+), ****P* < 0.0001. Source numerical data are available in Source data.

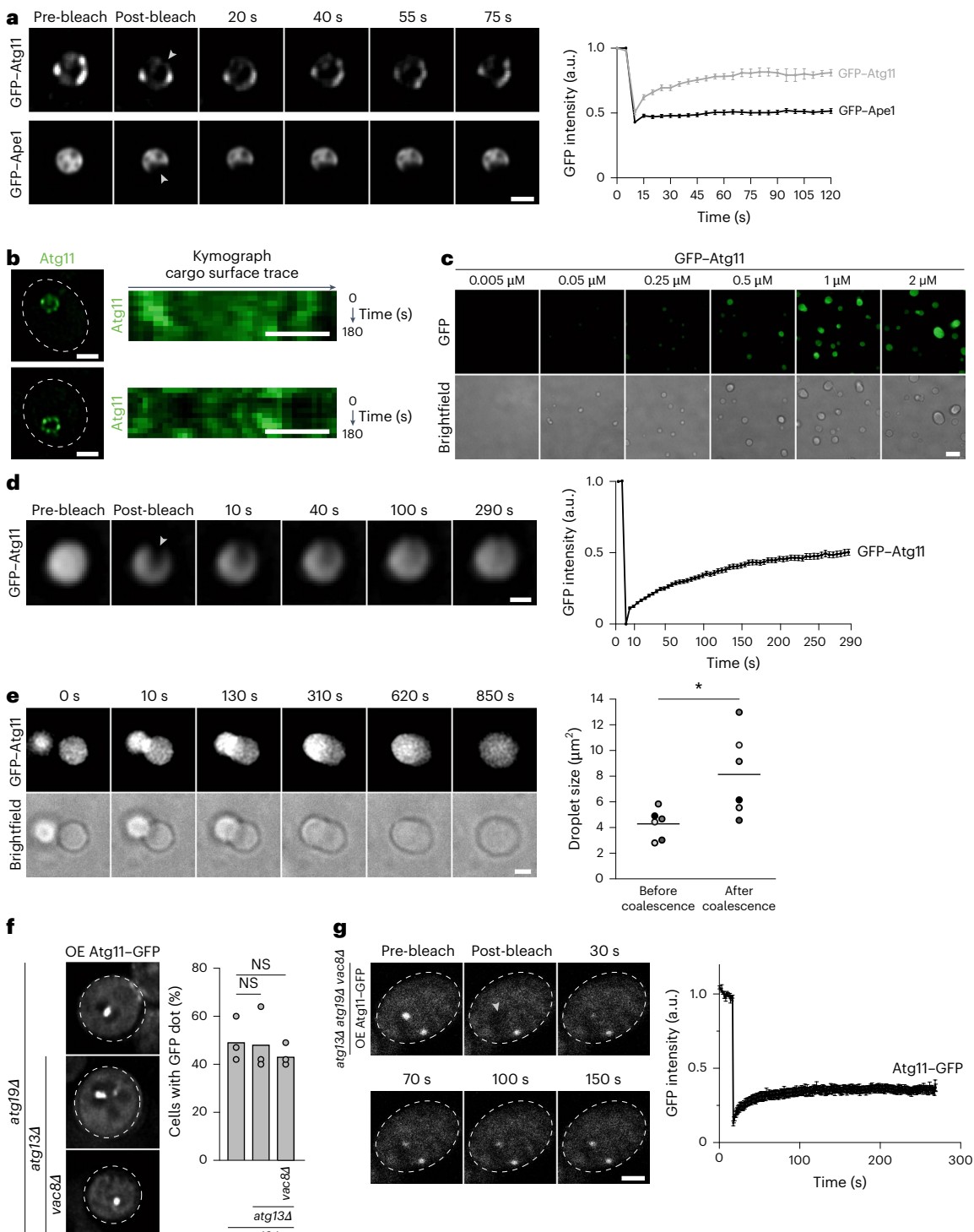

receptor on the cargo surface. We examined the recruitment of Atg11 to the surface of diverse selective autophagy cargoes with different physical properties, including ENDs, Ape1 and mitochondria. Whereas mitochondria represent a membrane-delimited cargo, ENDs and Ape1 are membrane-less cargoes that undergo phase separation[13,22]. Unexpectedly, Atg11 did not distribute uniformly around the cargo surface but instead accumulated in distinct foci (Fig. 2a). We compared this localization to that of Nup170–GFP a nuclear pore complex subunit, which forms foci around the nucleus, and to Vph1–GFP, which homogenously localizes around the vacuole. Analysing the coefficient of variance of fluorescence intensity around these structures, we observed that Atg11 formed prominent foci around Ape1, similar to the patterns

observed with Nup170–GFP (Fig. 2b)[28]. Important autophagosome biogenesis factors, such as the autophagy proteins Atg1 and Atg9, also clustered in these foci with Atg11 for both the Ape1 complex and the ENDs (Fig. 2c,d). Because these clusters contain multiple factors that are involved in phagophore initiation, we termed them 'initiation hubs'. We consider initiation hubs to be precursors of the PAS, forming multiple foci around the cargo to recruit autophagy machinery proteins such as Atg1 and Atg9.

We noticed that Atg11 clustering was reduced in cells with increased affinity of receptor–cargo interactions, such as the strains coexpressing Atg19–GBP with GFP–Ape1 (Fig. 2e and Extended Data Fig. 2a) and those coexpressing 2×GFP–Ede1 with BFP–3×GBP

**Fig. 3 | Phase separation of Atg11 drives initiation hub formation. a**, GFP–Atg11 cells expressing endogenous BFP–Ape1 and copper-inducible untagged Ape1 or wild-type cells expressing GFP–Ape1 and copper-inducible untagged Ape1 were grown to mid-log phase in the presence of 50 μM CuSO₄. GFP–Atg11 and GFP–Ape1 structures were photobleached and recovery of the signal was monitored. White arrowheads show the photobleached area. Scale bar, 1 μm. Quantification: recovery of the GFP signal. Data show mean ± s.e.m. (*n* > 32 foci per condition across replicates, three biological replicates). **b**, GFP–Atg11 cells expressing endogenous BFP–Ape1 and copper-inducible untagged Ape1 were grown to mid-log phase in the presence of 50 μM CuSO₄. The dynamics of GFP–Atg11 foci were monitored and represented as kymographs. Images of one out of three biological replicates are shown. Scale bar, 2 μm. Kymograph scale bar, 1 μm. **c**, GFP–Atg11 was purified from Sf9 insect cells and droplet formation was monitored in vitro at different concentrations by fluorescence microscopy after 20 min incubation at room temperature in a buffer containing 150 mM NaCl. Images from one out of three biological replicates are shown. Scale bar, 5 μm. Quantification is shown in Extended Data Fig. 2e. **d**, In vitro formed GFP–Atg11 droplets were photobleached and recovery of the signal was measured. White arrowheads indicate the photobleached area. Scale bar, 1 μm. Quantification: recovery of the GFP signal. Data show mean ± s.e.m. (*n* = 30 structures per condition and replicate,

three biological replicates). **e**, Coalescence of in vitro formed GFP–Atg11 droplets was monitored. Scale bar, 2 μm. Quantification: droplet size, represented in a scatter-plot. Statistical analysis was carried out by a two-tailed unpaired *t*-test. Circles show mean values of each replicate, horizontal lines show the median (*n* = 6 coalescence events, three biological replicates). *P = 0.026. Further examples are shown in Extended Data Fig. 3a. **f**, *atg19Δ*, *atg13Δ atg19Δ* or *atg13Δ atg19Δ vac8Δ* cells were transformed with Atg11–GFP overexpressed under a GPD promoter. The formation of Atg11 condensates was monitored. Scale bar, 2 μm. The percentage of cells with Atg11–GFP foci was quantified, displayed in a bar graph. Data are mean values (*n* = 100 cells per condition and replicate, three biological replicates). Circles show mean values of each replicate, bars show mean. Statistical analysis was conducted by one-way ANOVA followed by a Dunnett's post hoc test. *P* values: *atg19Δ* versus *atg19Δ atg13Δ*, *P* = 0.9885; *atg19Δ* versus *atg19Δ atg13Δ vac8Δ*, *P* = 0.688. **g**, Atg11–GFP was overexpressed in *atg19Δ atg13Δ vac8Δ* cells and cells were grown to the mid-log phase. GFP condensates were fully photobleached and their recovery was monitored. White arrowheads indicate the bleached area. Scale bar, 2 μm. Quantification: recovery of the GFP signal. Data show mean ± s.e.m. (*n* = 25 structures per condition across replicates, three biological replicates). Source numerical data are available in Source data.

(Fig. 2f). Atg1 was only recruited to Atg19 wild-type bound cargo but not to the one bound to Atg19–GBP (Extended Data Fig. 2b). Notably, single and combined deletion of Atg9 and Vac8 did not reduce Atg11 clustering on Ape1, suggesting that initiation hub formation is independent of downstream factors (Extended Data Fig. 2c). These data suggest that the elevated mobility of low-affinity cargo–receptor complexes allows Atg11 to establish initiation hubs and supports selective cargo degradation.

## Phase separation of Atg11 drives initiation hub formation

Notably, FRAP analysis revealed less mobility of the cargo GFP–Ape1 when compared with that of GFP–Atg11 (Fig. 3a). Time-lapse microscopy analysis of GFP–Atg11 revealed that most of the Atg11 clusters on the surface of the Ape1 complex dynamically change their size and morphology (Fig. 3b, Extended Data Fig. 2d and Supplementary Videos 1–12). To investigate these morphological changes in more detail, we purified GFP–Atg11 from insect cells and found that it formed round droplets in the presence of physiological salt concentrations, resembling a liquid-like behaviour (Fig. 3c and Extended Data Fig. 2e). Droplet formation was absent at low protein concentrations but became visible at around 0.05 μM, and droplet size increased with increasing concentration of Atg11. FRAP experiments showed that

GFP–Atg11 is largely mobile within the droplet (Fig. 3d). In addition, we observed coalescence of individual droplets, another typical feature of phase-separating proteins (Fig. 3e, Extended Data Fig. 3a and Supplementary Videos 13–15).

When expressed under its endogenous promoter and in the absence of the receptor Atg19, Atg11 did not form condensates in cells (Extended Data Fig. 3b,c). In vitro Atg11 phase separation was only observed at a certain protein concentration (Fig. 3c), so we hypothesized that endogenous Atg11 phase separates on cargo due to its local concentration being increased by Atg19 binding. Thus, Atg11 should act as a scaffold for condensation, and increasing its concentration in vivo should drive its phase separation independent of the presence of cargo. Indeed, overexpression of Atg11 resulted in condensation independent of Atg19, Atg13 and Vac8 (Fig. 3f,g). Truncation experiments revealed that coiled-coil regions 2 and 3 are required for its condensation in the presence or absence of cargo (Extended Data Fig. 3b–d). Additionally, these Atg19-independent Atg11 condensates were capable of recruiting Atg9, supporting the idea that Atg11 condensates are the driving force behind initiation hub formation (Extended Data Fig. 3e). Recombinant, preformed GFP–Atg11 condensates were efficiently recruited to GST–BFP–Atg19³ᴰ, a phospho-mimetic mutant of Atg19 known to stably

**Fig. 4 | Initiation hubs coalesce at the vacuolar contact site to trigger phagophore initiation. a**, Initial (top) and final (bottom) snapshots of a molecular simulation corresponding to a very-low-affinity, low-affinity and high-affinity interaction of Atg11–Atg19 subcomplexes (grey to green) with a cargo (blue). The clustering is shown by visualizing the number of neighbours of Atg11–Atg19 particles. Stronger colours indicate that more neighbours are in proximity and will result in a stronger coupling. **b**, GFP–Atg8 cells expressing mScarlet–Atg11, endogenous BFP–Ape1 and copper-inducible untagged Ape1 were grown to mid-log phase in the presence of 50 μM CuSO₄. Images of one out of three biological replicates are shown. Scale bar, 2 μm. **c**, mScarlet–Atg8 *atg19Δ* cells expressing endogenous GFP–Ape1 and copper-inducible untagged Ape1 along with Atg19 or Atg19–GBP were grown to mid-log phase in medium containing CuSO₄ and rapamycin to induce phagophore formation³². Scale bar, 2 μm. Quantification: elongated mScarlet–Atg8-positive structures on cargo. Data show the mean (*n* = 100 structures per condition and replicate, three biological replicates). Circles show the mean of each replicate, bars show the mean. Statistical analysis was conducted by a two-tailed unpaired *t*-test, *P* = 0.0061. **d**, Atg9–3×GFP mScarlet–Atg8 cells expressing endogenous BFP–Ape1 and copper-inducible untagged Ape1 grown and treated as in **c**. Scale bar, 2 μm. Quantification: cargo-associated Atg9 clusters. Data are mean values (*n* = 100 cells per condition and replicate, three biological replicates). Circles show the mean of each replicate, bars show the mean. Statistical analysis was conducted

by a two-tailed unpaired *t*-test. *P* values: rich versus rapa (for 1–2 clusters), ***P* = 0.0002; rich versus rapa (for >2 clusters), ***P* = 0.0002. **e**, GFP–Atg19 cells overexpressing copper-inducible untagged Ape1 were grown to mid-log in medium containing CuSO₄ and rapamycin (one out of three identified examples shown). (i) In situ cryo-electron tomographic slice. Orange arrowhead indicates phagophore; V, vacuole. (ii) Different slice (+16.6 nm from i). (iii) Segmentation and 3D rendering of the tomographic volume. Orange, phagophore; grey, vacuole membrane. Scale bar, 200 nm (i), 50 nm (ii,iii). Correlative fluorescence images are shown in Extended Data Fig. 5e. **f**, Indicated strains expressing endogenous GFP–Ape1 and copper-inducible untagged Ape1, Atg19–GFP–μNS or pp–GFP–μNS were grown to mid-log phase in the presence of 50 μM CuSO₄. GFP cleavage was monitored by immunoblotting. pp, Ape1¹⁻⁴⁵. One out of three biological replicates is shown. **g**, Cells as in **f** coexpressing mScarlet–Atg11 were monitored by fluorescence microscopy. Scale bar, 2 μm. Quantification: GFP clustering as the coefficient of variance (s.d./mean GFP intensity) in a box plot. Horizontal lines show the median, box shows 25th to 75th percentiles, whiskers indicate the 5th and 95th percentiles, circles show the mean value of each replicate, outliers are shown as black dots (*n* = 50 structures per condition and replicate, three biological replicates). Statistical analysis was conducted by a one-way ANOVA followed by a Dunnett's post hoc test. *P* values: pp–GFP–μNS versus Atg19–GFP–μNS, ****P* < 0.0001; pp–GFP–μNS versus GFP–Ape1, *P* = 0.9826. Source numerical data and unprocessed blots are available in Source data.

interact with Atg11 (ref. 29), when immobilized on GSH beads. GFP–Atg11 remained bound after subsequent washes, consistent with a stable interaction between the scaffold and receptor (Extended Data Fig. 3f). Moreover, these GFP–Atg11 condensates on the Atg19[3D]-decorated beads recovered upon photobleaching (Extended Data Fig. 3g), supporting their phase separation. Given that increasing the affinity of the Atg19 receptor for the Ape1 cargo reduced

Atg11 cluster formation (Fig. 2e), we propose that low-affinity Atg19 receptor–cargo interactions enable phase separation of Atg11–Atg19 complexes and initiation hub formation. Such a high on and off rate of cargo–receptor interactions represents an effective diffusion rate similar to the lateral diffusion in membranes. In general, this suggests that receptor mobility on cargo is critical for autophagy initiation and progression of the pathway.

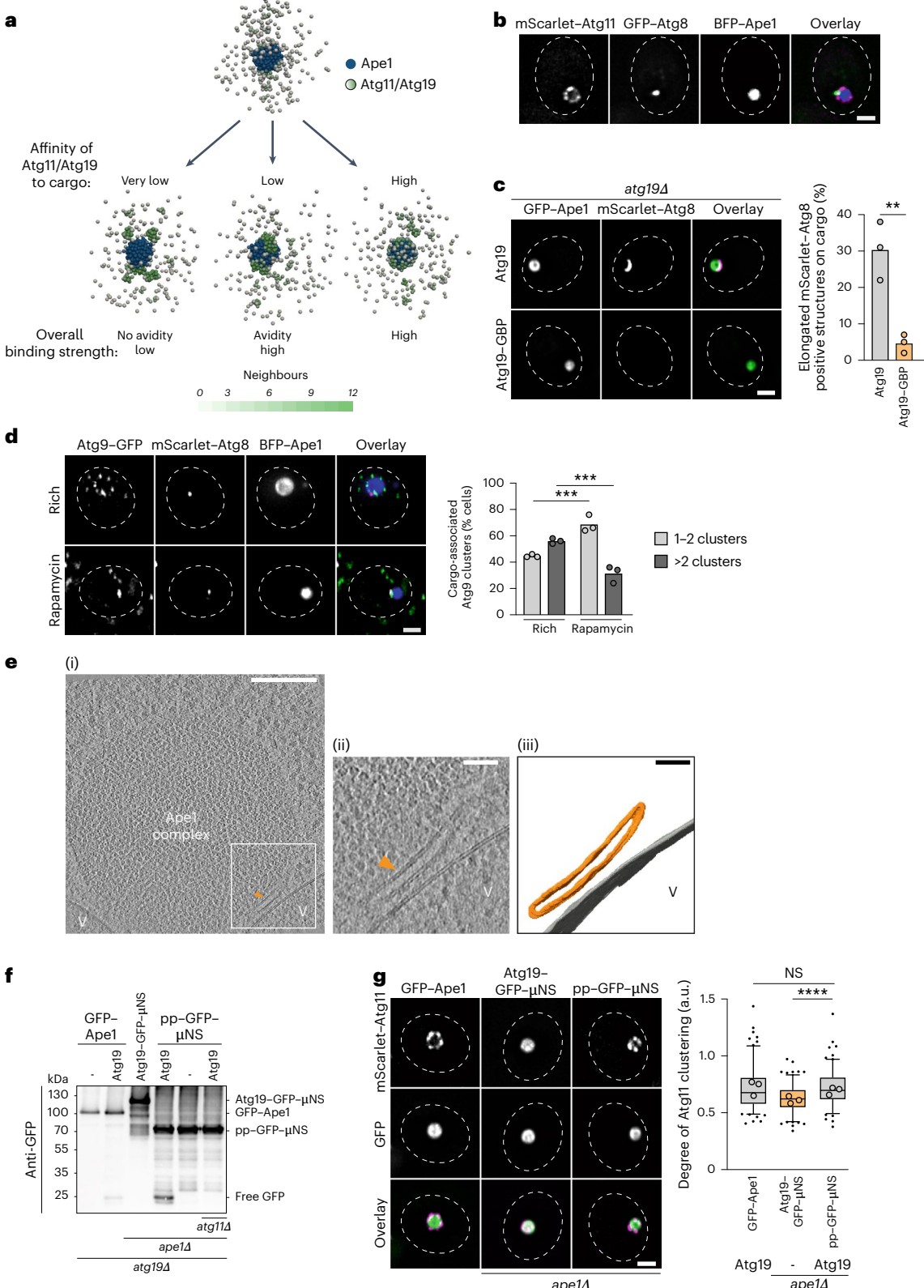

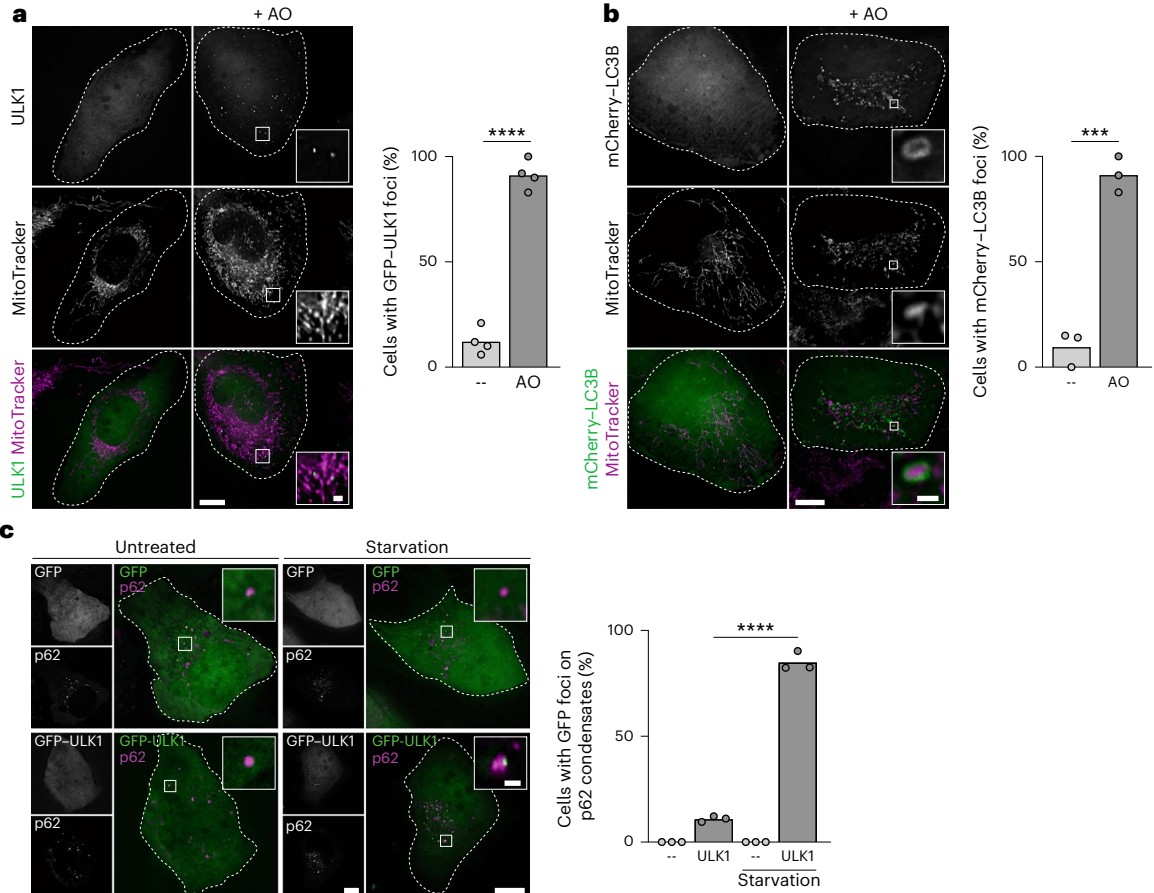

**Fig. 5 | Initiation hubs for selective autophagy are conserved in human cells.**
**a**, U2OS cells transfected with FKBP–GFP–ULK1 and mCherry–Parkin were stained with MitoTracker DeepRed. Mitophagy was induced with antimycin A and oligomycin (AO). Live cells were visualized by fluorescence microscopy. Scale bar, 10 µm; Scale bar inset, 1 µm. Quantification: cells containing GFP–ULK1 foci. Data are mean values (*n* > 130 cells per condition across replicates, four biological replicates). Circles show mean values of each replicate, bars show the mean. Statistical analysis was conducted by a one-tailed unpaired *t*-test, ****P < 0.0001. **b**, U2OS cells transfected with mCherry–LC3B and YFP–Parkin were stained with MitoTracker DeepRed and treated with AO. Images of one out of three biological replicates are shown. Scale bar, 10 µm. Quantification: cells containing mCherry–LC3B foci. Data are mean values (*n* > 50 cells per condition

across replicates, three biological replicates). Circles show mean values of each replicate, bars show mean. Statistical analysis was conducted by a one-tailed unpaired *t*-test, ****P = 0.0001. **c**, U2OS cells were transfected with FKBP–GFP or FKBP–GFP–ULK1 and mCherry–p62, and cultured in nutrient-rich medium or starvation medium (Earle's balanced salt solution; EBSS) for 4 h to induce bulk autophagy. Live cells were visualized by fluorescence microscopy. Images of one out of three biological replicates are shown. Scale bar, 10 µm, scale bar inset, 1 µm. Quantification: co-localization events between GFP and p62. Data are mean values (*n* > 100 cells per condition across replicates, three biological replicates). Circles show mean values of each replicate, bars show mean. Statistical analysis was conducted by one-way ANOVA followed by Sidak's multiple comparison test, ****P < 0.0001. Source numerical data are available in Source data.

Low-affinity interactions, such as those between the cargo and autophagy receptor, can be stabilized by increasing their number, resulting in avidity or high functional affinity[30]. To assess the importance of the avidity-driven interactions for the assembly of initiation hubs, we developed a mathematical model of the Atg11–Atg19 subcomplex and cargo using the modelling framework cellular_raza. We considered Atg11 and Atg19 as one entity due to their high-affinity interaction. We modelled a low-affinity interaction between Atg11 molecules, resembling its phase separation. We then simulated interactions between the cargo and the Atg11–Atg19 subcomplex at various affinity levels: very low affinity, low affinity and high affinity. The model with very low-affinity interactions resulted in a pattern of Atg11 clusters that were not located on the cargo (Fig. 4a and Supplementary Video 16). In the low-affinity model, Atg11 clusters formed on the cargo, as the avidity-mediated interactions were sufficiently strong to stabilize the clusters on the cargo surface, consistent with our in vivo observations (Fig. 4a, Supplementary Video 17). Conversely, the high-affinity interaction model produced uniform Atg11 binding on the cargo without forming large clusters, also reflecting our in vivo results (Fig. 4a

and Supplementary Video 18). This model supports that multivalent low-affinity interactions between the cargo and Atg11–Atg19 facilitate the formation of initiation hubs.

Taken together, our results suggest that receptor mobility on cargo allows Atg11 condensation and the formation of initiation hubs, which promote autophagy.

### Initiation hubs coalesce to trigger phagophore initiation

In selective autophagy, autophagosome formation is mediated through a spatial organization driven by numerous low-affinity but high-avidity-based interactions, with Vac8 coordinating these events at the vacuole at the site of the PAS[31]. Some of the Atg11 condensates co-localized with Vac8 (Extended Data Fig. 4a,b), and recombinant GFP–Atg11 condensates efficiently bound recombinant GST–Vac8 immobilized on GSH beads (Extended Data Fig. 4c). GFP–Atg11 condensates furthermore interacted with purified vacuoles in a Vac8-dependent manner (Extended Data Fig. 4d), resembling the native avidity-mediated binding between these proteins[31]. The interaction of the Atg11 condensates with Vac8 markedly deformed the

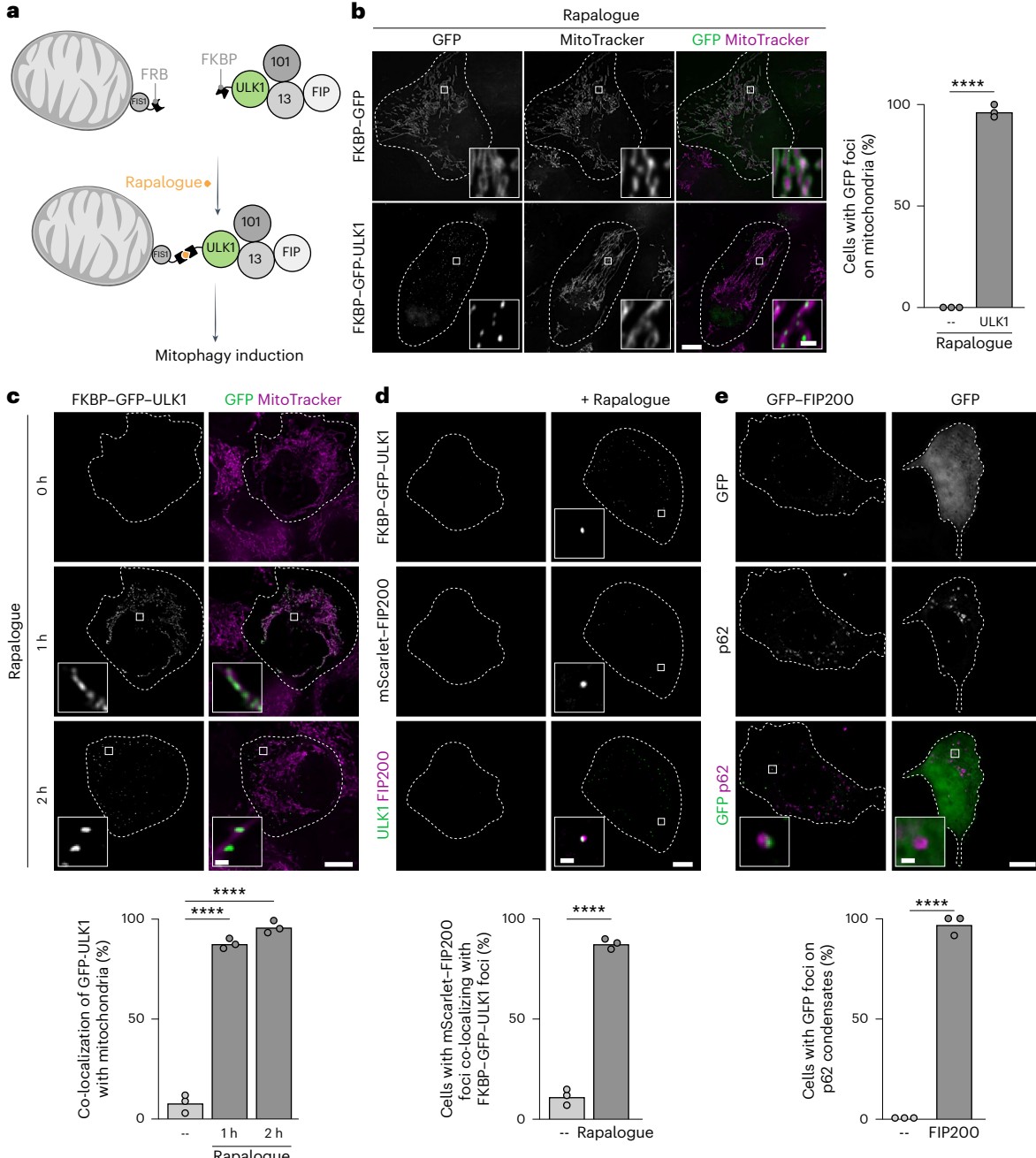

**Fig. 6 | FIP200 colocalizes with initiation hubs on mitochondria and p62 condensates. a,** Schematic of the synthetic tethering setup in U2OS cells. FRB is targeted to the outer mitochondrial membrane by its fusion with the tail anchor domain of the mitochondrial membrane protein FIS1 (FRB–FIS1$^{93-152}$, residues 93–152 of FIS1). Expression of an FKBP–GFP–ULK1 fusion construct allows its inducible tethering to mitochondrial FRB–FIS1$^{93-152}$ by rapalogue addition, resulting in mitophagy induction. **b,** U2OS cells stably expressing FRB–FIS1$^{93-152}$ and transfected with FKBP–GFP–ULK1 or FKBP–GFP were cultured in nutrient-rich medium and stained with MitoTracker DeepRed. Tethering was induced with rapalogue. Scale bar, 10 μm, scale bar inset, 1 μm. Quantification: GFP foci on mitochondria. Data are mean values ($n$ > 80 cells per condition across replicates, three biological replicates). Circles show mean values of each replicate, bars show mean. Statistical analysis was carried out by a one-tailed unpaired $t$-test. ****$P$ < 0.0001. **c,** U2OS cells stably expressing FRB–FIS1$^{93-152}$ and transfected with FKBP–GFP–ULK1 were cultured in nutrient-rich medium and stained with MitoTracker DeepRed. Tethering of FKBP–GFP–ULK1 to FRB–FIS$^{93-152}$ was induced with rapalogue. Scale bar, 10 μm; scale bar inset, 1 μm. Quantification: GFP foci on mitochondria. Data are mean values ($n$ > 100 cells per condition

across replicates, three biological replicates). Circles show mean values of each replicate, bars show mean. Statistical analysis was carried out by a one-way ANOVA followed by Sidak's multiple comparison test. $P$ values: 0 h versus 1 h, ****$P$ < 0.0001, 1 h versus 2 h, ****$P$ < 0.0001. **d,** U2OS cells stably expressing FRB–FIS1$^{93-152}$, transfected with FKBP–GFP–ULK1 and mScarlet–FIP200 were cultured in nutrient-rich medium. Tethering of FKBP–GFP–ULK1 to FRB–FIS$^{93-152}$ was induced by adding rapalogue for 1 h. Scale bar, 10 μm, scale bar inset, 1 μm. Quantification: mScarlet–FIP200 foci co-localizing with FKBP–GFP–ULK1 foci. Data are mean values ($n$ > 100 cells per condition across replicates, three biological replicates). Circles show mean values of each replicate, bars show mean. Statistical analysis was carried out by a one-tailed unpaired $t$-test. ****$P$ < 0.0001. **e,** U2OS cells transfected with FKBP–GFP or FKBP–GFP–FIP200 and mCherry–p62 were cultured in nutrient-rich medium. Scale bar, 10 μm; scale bar inset, 1 μm. Quantification: cells with GFP foci on p62 condensates. Data are mean values ($n$ > 100 cells per condition across replicates, three biological replicates). Circles show mean values of each replicate, bars show the mean. Statistical analysis was conducted by a one-tailed unpaired $t$-test. ****$P$ < 0.0001. Source numerical data are available in Source data.

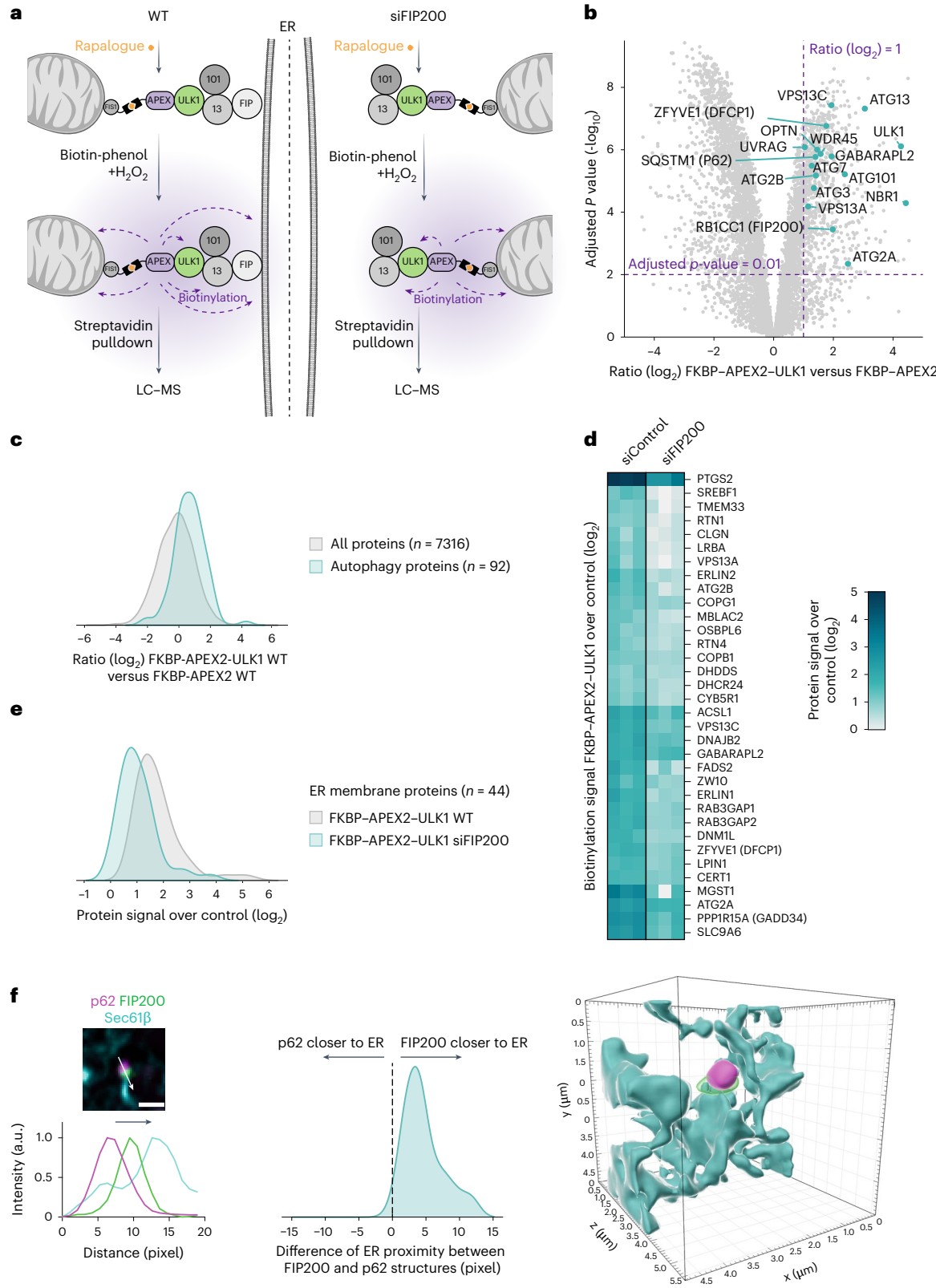

GFP–Atg11 droplet at the condensate-vacuole contact site, as expected for a liquid-like droplet (Extended Data Fig. 4d). Phase separations, much like subdomains in membranes, enhance the local concentration of proteins and their binding sites, creating an optimal environment for avidity-mediated interactions.

We noticed that some Atg11 clusters were largely immobile (Extended Data Fig. 4a,b). These clusters were mostly found at the contact site with the vacuole (Extended Data Fig. 5a,b) and were not only more stable compared with the surrounding clusters but also showed the highest fluorescence intensity. We hypothesized that the Atg11 clusters coalesce into a single, stable initiation hub at the contact site with the vacuole, eventually maturing into the PAS, from which autophagosomal membrane formation is catalysed. Indeed, cells lacking Vac8 also lost these high-intensity clusters (Extended Data Fig. 5c). Atg8, a marker

**Fig. 7 | Initiation hubs establish contact sites with the ER in mammalian cells.** **a**, Schematic representation of the proximity biotinylation setup. FKBP–APEX2–ULK1 is tethered to mitochondrial FIS1–FRB by rapalogue addition. FIP200 connects this mitochondrial assembly to the ER. Proximity biotinylation is induced by the addition of biotin–phenol and $H_2O_2$. Biotinylated proteins are isolated by affinity purification with streptavidin beads and analysed by mass spectrometry. FKBP–APEX2–ULK1 was compared with the unspecific control (FKBP–APEX) in each experiment. **b**, Mass spectrometry analysis of APEX2-based proximity labelling in HEK293 cells stably expressing 2×FKBP–APEX2–ULK1 or 2×FKBP–APEX2. Cells were grown under nutrient-rich conditions and rapalogue. Proximity labelling was induced by the addition of biotin–phenol and a short pulse of $H_2O_2$. The volcano plot shows the enrichment of biotinylated proteins in 2×FKBP–APEX2–ULK1 compared with 2×FKBP–APEX2. Dashed purple lines indicate the cutoffs used to identify ULK1-specific proteins ($\log_2$ ratio >1 and adjusted $P < 0.01$). Known autophagy proteins enriched in 2×FKBP–APEX2–ULK1 are highlighted in cyan. Ratios are calculated using mean values of three biological replicates. Statistical analyses carried out were moderated $t$-statistics using the limma-trend method and multiple testing correction with the Benjamini–Hochberg procedure. **c**, Mass spectrometry data from **b**. The kernel density estimate (KDE) plot compares all proteins against those listed under the Gene Ontology (GO) term 'autophagy' (GO:0006914) using the FKBP–APEX2–ULK1 versus FKBP–APEX2 ratio dataset. **d**, Heatmap displaying ULK1-specific ER membrane proteins with a significantly reduced signal upon knockdown of FIP200 (adjusted $P$ value < 0.05). The enrichment of protein signal in FKBP–APEX2–ULK1 over FKBP–APEX2 is shown for three biological replicates of wild-type and FIP200 knockdown cells. Statistical analysis: two-tailed unpaired $t$-test, multiple testing correction with the Benjamini–Hochberg procedure. **e**, Mass spectrometry data from **d**. The KDE plot shows the comparison FKBP–APEX2–ULK1 WT versus FKBP–APEX2–ULK1 siFIP200 of all proteins that are positive for the GO term 'ER membrane' (GO:0005789). The samples were control-corrected before the analysis. **f**, U2OS cells transfected with FKBP–GFP–FIP200, mCherry–p62 and BFP–Sec61β, were cultured in nutrient-rich medium. Intensity profiles were calculated for the 405 nm, 488 nm and 561 nm channels along the indicated line. A representative image from one out of three biological replicates is shown. Scale bar, 1 μm. The KDE plot shows the difference in ER proximity between FIP200 and p62 structures, calculated from 47 structures. The 3D surfaces of BFP–Sec61β (cyan), FKBP–GFP–FIP200 (green) and mCherry–p62 (magenta) were rendered with the Imaris software using the machine-learning tool for surface segmentation. A z-stack of 0.125 μm was taken to define the borders in z. Further examples are shown in Extended Data Fig. 7c,d. Source numerical data are available in Source data.

for phagophore formation, co-localized with vacuole-associated Atg11 initiation hubs in wild-type cells (Fig. 4b). Upon phagophore induction by rapamycin treatment[32], Atg8-positive structures elongated in about half of the wild-type cells, but not in cells containing the immobilized Atg19–GBP receptor (Fig. 4c). Atg9 and Atg1 co-localized with multiple Atg11 clusters, whereas Vac8 co-localized with only one (Fig. 2c and Extended Data Figs. 4a,b and 5a,b). These data are consistent with multiple membrane seeds established by Atg9 at initiation hubs eventually relocating to the Vac8-dependent PAS, where downstream factors such as the PI3K complex I are recruited for phagophore nucleation[31]. In line with this notion, similar to the phagophore membrane marker Atg8 (Fig. 4b), Atg9 relocalized to only a single focus upon induction of membrane expansion by rapamycin treatment, suggesting that phagophore formation is initiated at this single site (Fig. 4d). Similarly, also Atg11 redistributed upon induction of membrane expansion towards the vacuolar contact site (Extended Data Fig. 5d). To visualize the sites of phagophore biogenesis at the PAS directly in situ, we applied a correlative cryo-electron tomography workflow to target the site of the Ape1 complex in cells. Consistent with the fluorescence microscopy results, this revealed a phagophore membrane initiated between the vacuolar membrane and the Ape1 complex (Fig. 4e and Extended Data Fig. 5e). Taken together, these results suggest that initiation hubs coalesce at the vacuolar contact site and mature into the PAS, to trigger phagophore initiation.

## Ectopic formation of initiation hubs triggers degradation

Our work so far shows that receptor mobility on the surface of an autophagy cargo, rather than the biophysical nature of the cargo itself, is key to establish autophagy-competent Atg11-dependent initiation hubs. We hypothesized that artificially creating a low-affinity interaction with an autophagy receptor would trigger the degradation of a non-autophagic 'neo-cargo'. To test this, we manipulated the reoviral nonstructural protein μNS. The mammalian reovirus protein μNS is foreign to yeast, self-assembles into particles, accumulates in the cytosol of yeast cells and is not turned over by autophagy[31]. These features of μNS make it an ideal putative cargo for selective autophagy.

To engineer a low-affinity interaction between the Atg19 receptor and μNS, we constructed an Ape1 propeptide (pp)–GFP–μNS fusion. Notably, pp–GFP–μNS displayed autophagic degradation (Fig. 4f). In contrast, a direct Atg19–GFP–μNS receptor–cargo fusion, or an FK506-dependent high-affinity interaction between Cnb1–Atg19 receptor and FKBP–GFP–μNS cargo, did not trigger autophagic degradation (Fig. 4f and Extended Data Fig. 6a), despite the functionality of Cnb1–Atg19 in the absence of FK506 (Extended Data Fig. 6b), consistent with the high-affinity cargo–receptor interactions generated above (Fig. 1b,g and Extended Data Fig. 1b,c). Only the mobile pp–μNS but not the immobile Atg19–μNS restored Atg11 clustering, confirming, as predicted, that a low-affinity cargo–receptor interaction is necessary and sufficient for the formation of initiation hubs (Fig. 4g and Extended

**Fig. 8 | Receptor mobility is also a required feature for selective autophagy in human cells.** **a**, U2OS cells stably expressing FRB–FIS1[93–152] and transfected with 2×FKBP–GFP–ULK1 were cultured in nutrient-rich medium. To induce the homo-oligomerization of 2×FKBP–GFP–ULK1 the cells were treated with the homodimerizer AP20187 for 24 h. Tethering of FKBP–GFP–ULK1 to FRB–FIS[93–152] was induced by adding rapalogue for 2 h. Images of one out of three biological replicates are shown. Scale bar, 10 μm; scale bar inset, 1 μm. **b**, U2OS cells stably expressing FRB–FIS1[93–152] and transfected with 2×FKBP–GFP–ULK1 were cultured in nutrient-rich medium and stained with MitoTracker DeepRed. To induce the homo-oligomerization of 2×FKBP–GFP–ULK1 the cells were treated with AP20187 for 24 h. Tethering of 2×FKBP–GFP–ULK1 to FRB–FIS[93–152] was induced by adding rapalogue for 2 h. 2×FKBP–GFP–ULK1 clusters were photobleached, recovery of the signal was monitored. White arrowheads indicate the bleached area. Scale bar, 1 μm. Quantification: recovery of the GFP signal. Data are mean values ± s.e.m. ($n$ = 30 structures per condition across replicates, three biological replicates). **c**, U2OS wild-type cells stably expressing mito-mKeima and FRB–FIS[93–152] and FKBP–GFP–ULK1 were grown in nutrient-rich medium and treated with Bafilomycin A1 (Baf), rapalogue and AP20187 (24 h) as indicated. Cytosolic and lysosomal mt-mKeima fluorescence signal was monitored using flow cytometry and gating for GFP-expressing cells was performed. Data are mean values ($n$ > 50,000 cells per condition and replicate, three biological replicates). Circles show mean values of each replicate, bars show the mean. Statistical analysis was carried out by a one-way ANOVA followed by Sidak's multiple comparison test. $P$ values: 2 h, ****$P$ = 0.0001; 4 h, ****$P$ < 0.0001. **d**, U2OS WT cells were transfected with either GFP–p62–GFP alone or together with mCherry–3×GBP. GFP–p62–GFP clusters were photobleached and recovery of the signal was monitored. White arrowheads indicate the photobleached area. Scale bar, 1 μm. Quantification: recovery of the GFP signal. Data are mean values ± s.e.m. ($n$ > 20 structures per condition across replicates, three biological replicates). **e**, WT and ATG13KO U2OS cells were transfected with either GFP–p62–GFP alone or together with mCherry–3×GBP. The cells were starved in EBSS medium. GFP cleavage was monitored by immunoblotting. RFP, red fluorescent protein. Quantification: ratio between free GFP and 2×GFP–p62. Data are mean values ($n$ = 4 biological replicates). Circles show values of each replicate, bars show mean. Statistical analysis was carried out by a one-way ANOVA followed by Sidak's multiple comparison test. $P$ values: WT, ****$P$ < 0.0001; ATG13KO, $P$ = 0.9982. **f**, A universal model of selective autophagy (see text for details). Source numerical data and unprocessed blots are available in Source data.

Data Fig. 6c,d). In summary, these findings suggest that targeting an autophagy receptor to a multimeric neo-cargo via a low-affinity interaction is sufficient to trigger its autophagic degradation.

## Initiation hubs are conserved in human cells

To test whether the principles of initiation hub formation during selective autophagy initiation in yeast are conserved in humans, we induced Parkin-dependent mitophagy in human U2OS cells by treating them with antimycin A and oligomycin (AO). We found that the ULK1 kinase,

the human homologue of yeast Atg1, was diffuse in the cytoplasm of untreated U2OS cells but formed distinct foci at the mitochondria in AO-treated cells (Fig. 5a). Similarly, LC3B, a human homologue of yeast Atg8, was recruited to mitochondria in AO-treated cells (Fig. 5b). Upon starvation, ULK1 also formed clusters on phase-separated p62 condensates, another selective autophagy cargo (Fig. 5c and Extended Data Fig. 7a).

To assess mitophagy without inducing mitochondrial damage, we used the rapalogue-inducible FRB–FKBP dimerization system in U2OS

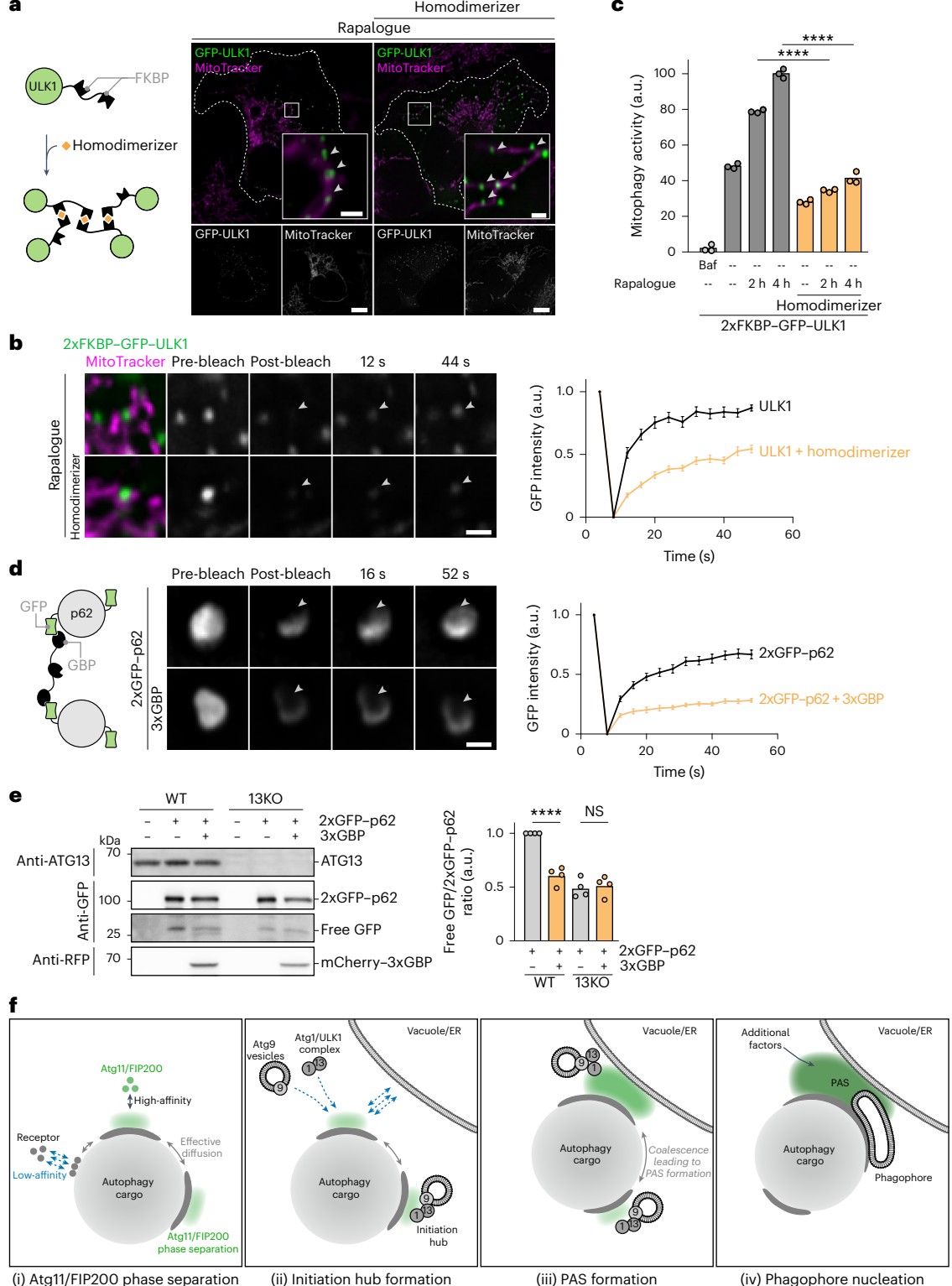

cells to target ULK1 to mitochondria[33]. Specifically, we fused FRB to the tail anchor domain of the mitochondrial membrane protein FIS1 (FRB–FIS1[93–152], residues 93–152 of FIS1). Coexpression of an FKBP–GFP–ULK1 fusion enables its rapalogue-inducible tethering to mitochondrial FRB–FIS1[93–152] and mitophagy induction (Fig. 6a and Extended Data Fig. 7b)[33,34]. This tethering approach recruits FKBP-tagged proteins along the entire mitochondrial network, as observed for FKBP–GFP (Fig. 6b). We hypothesized that targeting FKBP–GFP–ULK1 to the outer mitochondrial membrane would still promote the assembly of initiation hubs as it can diffuse laterally within the mitochondrial membrane and, therefore, rearrange. Indeed, treatment with rapalogue resulted in the formation of distinct FKBP–GFP–ULK1 foci along the mitochondrial network (Fig. 6b,c) that were not observed upon rapalogue-dependent targeting of FKBP–GFP (Fig. 6b), indicating a requirement for ULK1 and autophagy. FIP200, the human homologue of Atg11, undergoes phase separation during bulk autophagy induction[35], and we found that FIP200 co-localized in discrete foci with FKBP–GFP–ULK1 on the mitochondrial surface upon tethering (Fig. 6d), as well as to p62 condensates (Fig. 6e).

Collectively, these data support that initiation hub formation is a conserved feature of selective autophagy in human cells.

### Initiation hubs establish contact sites with the ER

Autophagy components assemble on the endoplasmic reticulum (ER) surface in mammalian cells in bulk autophagy[36,37]. We considered that autophagy machinery components also assemble on the ER surface during selective autophagy. To this end, we asked whether mitochondrial initiation hubs form contact sites with the ER in U2OS cells. We performed peroxidase-catalysed biotin proximity labelling by expressing the modified plant peroxidase APEX2 fused to either FKBP–ULK1 or as a control to FKBP alone (Fig. 7a,b)[38,39]. Biotinylated autophagy factors, including ATG3, ATG7 and GABARAPL2 were enriched in cells expressing FKBP–APEX2–ULK1 compared with those expressing FKBP–APEX2 (Fig. 7b,c). Notably, several biotinylated ER proteins were also enriched, including the ER-resident autophagic initiation factor DFCP1, and the lipid transfer proteins ATG2A and ATG2B and VPS13A and VPS13C, confirming the role of the ER as an assembly platform during selective autophagy. FIP200 has been reported to link the ULK1 complex to the ER during bulk autophagy[37]. In line with this function, we found that the biotinylation of ER membrane proteins in cells expressing FKBP–APEX2–ULK1 depended on FIP200 (Fig. 7d,e). To test whether the requirement of FIP200 for ER tethering is specific to mitochondria or also true for other types of selective cargo, we analysed its localization at contact sites between p62 condensates and the ER. Fluorescence intensity profile plots of FIP200, p62 and the ER marker protein Sec61β showed that FIP200 formed foci on p62, which were positioned between p62 and the ER marker Sec61β, suggesting a conserved role of FIP200 in ER tethering (Fig. 7f and Extended Data Fig. 7c). 3D reconstructions of the fluorescence images further supported these findings (Extended Data Fig. 7d and Supplementary Video 19).

The formation of initiation hubs and their maturation into the PAS in yeast required receptor mobility. Our engineered FRB–FIS1 tether provides such high mobility through its ability to diffuse laterally across the membrane. Indeed FRB–FIS1 targeted ULK1 clusters rapidly recovered after photobleaching, indicating their mobility on mitochondrial membranes (Extended Data Fig. 7e). To reduce this mobility and to test whether mobility is needed for initiation hub formation and PAS maturation in mammalian cells, we induced homo-oligomerization of 2×FKBP–ULK1 using the homodimerizing molecule AP20187, which dimerizes FKBP. As our ULK1 construct contains two FKBPs, this not only leads to dimerization but also to the formation of stable multimers. Upon homo-oligomerization and rapalogue treatment, distinct FKBP–GFP–ULK1 foci formed along the mitochondrial network, resembling those observed without the homo-oligomerizer (Fig. 8a). These foci exhibited significantly slower

recovery after photobleaching compared with non-oligomerized ULK1 foci, confirming the reduced mobility (Fig. 8b). To test the effect of this reduced mobility on mitophagy, we performed flux measurements using the mKeima reporter assay[40]. Whereas the mitophagy flux increased upon rapalogue-induced tethering of 2×FKBP–ULK1 to mitochondria as expected, the parallel homo-oligomerization significantly reduced this flux (Fig. 8c). Next, we manipulated the properties of the p62 condensate by coexpressing 2×GFP–p62 with 3×GBP, which, as expected, resulted in a strong reduction of p62 mobility (Fig. 8d). This decrease in p62 mobility also led to a marked slowdown in p62 turnover after autophagy induction by starvation, as cells expressing 3×GBP (Fig. 8e) showed significantly lower free GFP processing compared with those expressing only 2×GFP–p62.

Although additional experiments are needed to generalize the effect of cargo properties and their turnover in mammalian cells, these findings suggest that the formation and function of initiation hubs during selective autophagy, as well as their maturation into a fully functional PAS at ER contact sites, are conserved in mammalian cells.

## Discussion

### A universal model of selective autophagy

We propose that the mobility of receptors on selective cargo is key for cargo degradability by selective autophagy (Fig. 8f). Mobility of receptor molecules on the cargo surface support the recruitment of Atg11/FIP200, promoting its phase separation and the formation of initiation hubs. These initiation hubs stabilize low-affinity interactions with the autophagy machinery through high avidity, ensuring proper spatiotemporal regulation of phagophore initiation. Rearrangements further allow the coalescence of multiple initiation hubs to establish the PAS, where phagophore formation is ultimately initiated. For membrane-delimited cargo, rearrangements are supported by lateral diffusion of proteins within the membrane itself. For membrane-less cargo, such mobility on cargo can either be achieved by the phase separation of the cargo itself, or by low-affinity interactions between the cargo and receptors, which allow a high on–off rate and therefore an effective diffusion rate similar to the lateral diffusion in membranes. This together with other cargo properties such as size and shape will determine the cargo degradability. Our data further suggests that the introduction of a mobile surface on so-far nondegradable cargo might render these degradable and could open opportunities to engineer the targeting of aberrant structures such as amyloid fibres.

The best-known example of a balance between affinity and avidity is the effectiveness of antibodies in recognizing and responding to diverse pathogens. IgM antibodies are generated during the early phase of the immune response against pathogens. These antibodies possess ten low-affinity binding sites, which leads to high avidity and a high overall binding strength. This facilitates a swift screening of potential threats while allowing the reversible release of improperly bound non-antigens. In contrast, IgG antibodies produced in the later stages establish more stable and enduring bonds with antigens. This high-affinity mode of binding is essential for the sustained effectiveness of immune responses over an extended duration. Similarly, avidity-based interactions in autophagy are important at multiple stages throughout the pathway and are likely key in the forward progression and self-organization of the process.

Our findings, therefore, suggest that the concept of low affinity and high avidity is an underappreciated but important aspect in regulating the initiation of biological pathways.

## Online content

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

Article

Mariya Licheva [1,2,12], Jeremy Pflaum [3,12], Riccardo Babic [1,2,4,12], Hector Mancilla [1,12], Jana Elsässer [1,2,4], Emily Boyle [3], David M. Hollenstein [5,6], Jorge Jimenez-Niebla [1,2,4], Jonas Pleyer [7], Mio Heinrich [7,8,9], Franz-Georg Wieland [7,8,9], Joachim Brenneisen [1,2], Christopher Eickhorst [1,2,4], Johann Brenner [3], Shan Jiang [3], Markus Hartl [5,6], Sonja Welsch [10], Carola Hunte [1,8,11], Jens Timmer [7,8,9], Florian Wilfling [3] ✉ & Claudine Kraft [1,8] ✉

[1]Institute of Biochemistry and Molecular Biology, ZBMZ, Faculty of Medicine, University of Freiburg, Freiburg, Germany. [2]Faculty of Biology, University of Freiburg, Freiburg, Germany. [3]Mechanisms of Cellular Quality Control, Max Planck Institute of Biophysics, Frankfurt am Main, Germany. [4]Spemann Graduate School of Biology and Medicine (SGBM), University of Freiburg, Freiburg, Germany. [5]Department for Biochemistry and Cell Biology, University of Vienna, Center for Molecular Biology, Vienna Biocenter Campus (VBC), Vienna, Austria. [6]Mass Spectrometry Facility, Max Perutz Labs, Vienna Biocenter Campus (VBC), Vienna, Austria. [7]Freiburg Center for Data Analysis and Modelling (FDM), University of Freiburg, Freiburg, Germany. [8]CIBSS - Centre for Integrative Biological Signalling Studies, University of Freiburg, Freiburg, Germany. [9]Institute of Physics, University of Freiburg, Freiburg, Germany. [10]Central Electron Microscopy Facility, Max Planck Institute of Biophysics, Frankfurt am Main, Germany. [11]BIOSS-Centre for Biological Signalling Studies, University of Freiburg, Freiburg, Germany. [12]These authors contributed equally: Mariya Licheva, Jeremy Pflaum, Riccardo Babic, Hector Mancilla. ✉e-mail: florian.wilfling@biophys.mpg.de; kraft@biochemie.uni-freiburg.de

## Methods

### Yeast strains and plasmids

Mammalian plasmids are listed in Supplementary Table 1. Yeast, bacterial and insect cell expression plasmids are listed in Supplementary Table 2. Yeast strains are listed in Supplementary Table 3. Yeast genomic insertions and tagging were performed according to Janke et al.[41] and multiple deletions were generated by PCR knockout and/or mating and dissection. GFP–ATG11-containing strains were generated by crossing with yTB283 (ref. 19) and GFP–ATG8-containing strains were crossed with yTB281, which had been generated by seamless tagging[42].

### Growth conditions

Yeast cells were grown in a synthetic medium containing glucose (SD; 0.17% yeast nitrogen base, 0.5% ammonium sulfate, 2% glucose and amino acids as required) or lactate medium (Slac; 0.17% yeast nitrogen base, 0.5% ammonium sulfate, 2% lactic acid, 0.1% glucose and amino acids as required) or rich medium (YPD; 1% yeast extract, 2% peptone and 2% glucose), to the mid-log phase. To induce bulk autophagy, cells were washed and resuspended in a nitrogen starvation medium (SD-N; 0.17% yeast nitrogen base without amino acids and with 2% glucose) or treated with 220 nM (Fig. 4c,d and Extended Data Fig. 5d) or 100 nM (Fig. 1b and Extended Data Fig. 1b,c) rapamycin. Yeast liquid cultures were incubated with shaking at 200 or 220 rpm at 30 °C.

### Antibodies

The following antibodies were used in this study: mouse monoclonal anti-GFP (1:100 dilution; 2B6, Monoclonal Antibody Facility, Max Perutz Labs), mouse monoclonal anti-GFP (1:5,000; 7.1 and 13.1, ref no. 11814460001, lot no. 70378300, Roche), IRDye 800CW goat anti-mouse (1:1,000 dilution, ref no. 926-32210, lot no. D10825-15, Licor), anti-Pgk1 (1:10,000 dilution, 22C5D8, ref no. 459250 lot no. VC2958788, Invitrogen), rabbit polyclonal anti-Ape1 (1:15,000 dilution), was generated by immunizing rabbits with a synthetic peptide corresponding to amino acids 168–182 (ref. 43), rabbit polyclonal anti-Atg19 (1:5,000 dilution, Sascha Martens, Monoclonal Antibody Facility, Max Perutz Labs)[9], mouse monoclonal anti-GST (1:1,000 dilution, 2H3-D10, Monoclonal Antibody Facility, Max Perutz Labs), rabbit polyclonal anti-ATG13 (1:50 dilution, 5HY-C1-F8, Monoclonal Antibody Facility, Max Perutz Labs)[44] and mouse monoclonal anti-RFP (1:1,000 dilution, 6g6, ref no. 6g6-100 lot no. 51020014AB-05, Chromotek).

### Standard biochemical assays

Yeast cell cultures were either precipitated with 7% trichloroacetic acid (TCA) for 30 min on ice or overnight at −20 °C or with 10% TCA for 20 min on ice. Precipitated proteins were either pelleted at 16,000$g$ for 15 min at 4 °C, washed with 1 ml acetone, air-dried, resuspended in urea loading buffer (120 mM Tris-HCl, pH 6.8, 5% glycerol, 8 M urea, 143 mM β-mercaptoethanol and 8% SDS), boiled and analysed by SDS–PAGE (Figs. 1g and 4f and Extended Data Figs. 1a and 6a,b) or at 15,000$g$ for 3 min at 4 °C, washed twice with 1 ml ice-cold acetone, air-dried, resuspended in MURB buffer (50 mM $Na_2HPO_4$, 25 mM 2-(*N*-morpholino)ethanesulfonic acid (MES), pH 7.0, 1% SDS, 3 M urea, 0.5% 2-mercaptoethanol, 1 mM $NaN_3$ and 0.05% bromophenol blue), vortexed for 5 min with acid-washed glass beads and boiled[45]. Samples were loaded on 4–12% NuPAGE–SDS gels (Invitrogen), transferred to PVDF membranes and analysed by immunoblotting (Extended Data Fig. 1b).

### Tethering

Tethering in exponentially grown cells was induced by the addition of 3 μM FK506 (LC Laboratories) for 1 h.

### Ape1 expression for clustering analysis by live microscopy

Yeast cells were transformed with a plasmid containing GFP–Ape1 or BFP–Ape1 expressed under its endogenous promoter and a second copy of Ape1 under the copper-inducible CUP1 promoter. Cells were grown in a synthetic medium containing glucose (SD; 0.17% yeast nitrogen base, 0.5% ammonium sulfate, 2% glucose and amino acids as required) to the mid-log phase. Expression of Ape1 was induced by the addition of 50 μM $CuSO_4$ overnight. Overexpression of Ape1 largely results in intermediate-sized Ape1 particles that are still degraded by autophagy (Fig. 4f and Extended Data Fig. 6a). The degree of clustering for GFP- and mScarlet-tagged proteins was measured by the coefficient of variance of the fluorescence intensity around the cargo (s.d. divided by the mean GFP or mScarlet intensity).

### Mitochondria staining and mitophagy induction in yeast

Exponentially growing cells were stained with MitoTracker Red (Invitrogen) for 30 min at 30 °C. Cells were washed 1× with synthetic medium containing glucose and incubated for an additional 15 min at 30 °C. To induce mitophagy, stained cells were washed and resuspended in a nitrogen starvation medium (SD-N; 0.17% yeast nitrogen base without amino acids and with 2% glucose). Yeast cultures were incubated with shaking at 220 rpm at 30 °C for 3 h.

### Live-cell imaging of yeast

Exponentially growing, nitrogen-starved or rapamycin-treated cells were placed on 35-mm glass-bottom dishes (D35-20 1.5-N, In Vitro Scientific) or microscopy slides pretreated with 1 mg ml⁻¹ of concanavalin A type IV (Sigma-Aldrich) and live-cell imaging was performed at room temperature. Fluorescent microscopy images were recorded with a DeltaVision Ultra High Resolution microscope (GE Healthcare, Applied Precision) equipped with an UPlanSApo ×100/1.4 oil Olympus objective, an sCMOS pco. edge camera (PCO) and a seven-channel solid-state light source (Lumencor) (Fis. 3c,e and Extended Data Figs. 2e, 3a,f and 4d); or with a ×60/1.4 oil Olympus objective (Fig. 1e and Extended Data Fig. 4c) with a DeltaVision OMX Flex Microscope with UPlanSApo ×60/1.4 oil Olympus objective, using a PCO Edge 4.2 sCMOS camera (Figs. 2a–c,e, 3a,b,d,f and 4b,c,d,g and Extended Data Figs. 1g,h, 2a–d, 3c,d,g, 4a,b, 5a,c,d and 6c,d); or with a Leica Stellaris 5 system with a ×63/1.40 oil (HC PL APO CS2) objective (Figs. 1c and 3g) or with a Nikon Eclipse Ti2 with a ×100/1.49 oil (Apo TRIP) objective and a Hamamatsu C11440-22C camera (Figs. 1b,d and 2a,d,f and Extended Data Figs. 1c,e,f, 3e and 5b). Dashed lines indicate the contour of individual cells.

Raw microscopy images acquired with the DeltaVision Ultra High Resolution microscope or with the DeltaVision OMX Super Resolution microscope were deconvolved using the softWorX deconvolution plugin (v.R6.1.1 and v.7.2.1, respectively). Image analysis was performed using Fiji[46]. Images from each figure panel were taken with the same imaging setup and are shown with the same contrast settings. Single focal planes of representative images are shown. For quantification, at least three independent replicates were analysed and manual counting was performed blindly after randomizing image names.

Raw microscopy images acquired with Nikon Eclipse Ti2 system were deconvolved using the NIS Elements Batch Deconvolution v.5.20.00 and the chromatic shift was measured using fluorescent beads and corrected using Huygens 23.10.

The solidification of 2×GFP–Ede1 condensates was monitored by the co-localization of mTagBFP–3×GBP with 2×GFP–Ede1 after copper induction. For further co-localizations with ENDs, mid-section images were acquired. For the quantification of mScarlet–Atg11 co-localization to ENDs conditions of the same day were compared with each other and normalized to the amount of Atg11 co-localization observed in the −3×GBP strain.

To measure the turnover of solidified ENDs, the area of ENDs, the number of ENDs per cell or the co-localization of Atg8 to ENDs, mid-section images of cells after induction with copper before and after 24 h rapamycin treatment were acquired. For analysis, cells were identified with ilastik-1.4.0-OSX[47] via self-trained pixel-based classification and ENDs were identified and measured using Fiji. To investigate

binding of GBP[low] to 2×GFP–Ede1, cells were induced with copper, as carried out for the solidification, and z-stacks of 41 pictures with 0.2-μm distance were acquired. Maximum intensity projections were made and ENDs of similar brightness compared among the different strains. The co-localization of Atg9–3xmCherry to overexpressed Atg11–GFP clusters was observed by taking z-Stacks of 21 images with a step size of 0.4 μm.

For protein co-localization or analysis of GFP–Atg11 peak intensity and distribution on the surface of Ape1 images were generated by collecting a z-stack of 21 pictures with focal planes 0.25-μm apart. To quantify the degree of Atg11 clustering on different cargo structures, images were generated by collecting a z-stack of 21 pictures with focal planes 0.25-μm apart. For kymographs, time-lapse videos of at least ten frames were collected with a time interval of 20 s.

### In vivo FRAP analysis

2×GFP–Ede1 FRAP analysis was performed on exponentially grown DF5 cells. Expression of mTagBFP–3×GBP constructs was performed in LoFl medium (low fluorescence synthetic growth medium (yeast nitrogen base without amino acids and without folic acid and riboflavin (FORMEDIUM)) supplemented with all essential amino acids and 2% glucose) by the addition of 1 mM $CuSO_4$ for 6 h and incubation at 30 °C. The image resolution was set to 512 × 512 pixels, pixel size 11.1 pixels per μm, excitation wavelength to 488 nm, emission detection window to 490–750 nm, line time to 0.001 and line average to 1. For each sample, 420 frames were collected with a time interval of 0.518 s. Of those, 20 frames were collected before bleaching the region of interest with 100% laser power for two frames and 400 frames were collected immediately after bleaching. For FRAP of mTagBFP–3×GBP-solidified ENDs only ENDs with a strong co-localization between 2×GFP–Ede1 and mTagBFP–3×GBP were chosen. Note that 2×GFP–Ede1 contains an N-terminal and a C-terminal GFP tag on Ede1.

Atg11–GFP FRAP analysis was performed on BY474x cells overexpressing Atg11–GFP under a GPD promotor grown to mid-log phase. The image resolution was set to 512 × 512 pixels, pixel size 22.2 pixels per μm, excitation wavelength to 488 nm, emission detection window to 490–750 nm, line time to 0.001 and line average to 1. For each sample, 320 frames were collected with a time interval of 0.518 s. Of those, 20 frames were collected before bleaching the region of interest with 100% laser power for two frames and 300 frames were collected immediately after bleaching.

GFP–Atg11, GFP–Atg19 and GFP–Ape1 FRAP analysis was performed on exponentially grown BY474x cells. The image resolution was set to either 256 × 256 or 512 × 512 pixels and the excitation wavelength to 488 nm. For each sample, at least 13 frames were collected with a time interval of 3 or 5 s. An additional z-stack of three pictures with focal planes 0.25-μm apart was acquired for Fig. 3a. For each FRAP experiment, three frames were collected before bleaching, and at least ten frames were collected immediately after bleaching.

To correct for drift or rotations in the samples, images were processed using the Fiji plugin StackReg[48]. Double normalization was performed as described[49]. For this, the average fluorescence intensities were recorded within either the bleached region, the entire Ede1-positive structures, the entire GFP–Atg11, GFP–Atg19 or GFP–Ape1-positive structures using an in-house built analysis pipeline, and at a random cell-free spot for background subtraction before and immediately after photobleaching. Normalization was performed as described for overexpressed Atg11–GFP[49].

### In vitro FRAP analysis of Atg11 droplets

Purified GFP–Atg11 from Sf9 insect cells was diluted to a concentration between 0.005 μM and 2 μM in buffer containing 20 mM HEPES, pH 7.5, 150 mM NaCl and 5 mM dithiothreitol (DTT). For condensate formation, samples were incubated at room temperature for 20 min and transferred into a Chamber μ-Slide VI 0.1 (Ibidi) or incubated

with GST–mTagBFP–Atg19[3D]-coupled beads. FRAP analysis of GFP–Atg11 condensates was performed at room temperature. The image resolution was set to 256 × 256 pixels or 512 × 512, and the excitation wavelength to 488 nm. For each sample, at least 61 or 25 frames were collected with a time interval of 3 s or 5 s. Of those, two frames were collected before bleaching, and at least 58 or 21 frames were collected immediately after bleaching. Double normalization was performed as described elsewhere[48]. The analysis was conducted as described in the 'In vivo FRAP analysis' section.

### Protein expression in *E. coli*

GST, GST–mTagBFP, GST–mTagBFP–GBP, GST–mTagBFP–GBP[F103AE104Q], GST–Ape1[1–45], GST–mTagBFP–Ape1[1–45], GST–Atg11[685–1178], GST–Atg19[3D], GST–mTagBFP–Atg19[3D], GST–Atg19 and GST–mCherry–Atg19 fusion constructs were expressed from pGEX-4T.1 in *E. coli* BL21(DE3). 6×His–GFP was expressed from pRSET-A in *E. coli* BL21(DE3). Cells were grown in lysogeny broth medium supplemented with ampicillin at 37 °C to an $OD_{600}$ of 0.7, the temperature was reduced to 16 °C and expression was induced by the addition of 1 mM isopropyl-b-ᴅ-thiogalactoside (IPTG) for 18 h. Cells were pelleted at 3,000g for 15 min at room temperature, resuspended in GST-lysis buffer (50 mM Tris-HCl, pH 7.5, 150 mM NaCl, 5% glycerol, 1% Triton X-100, 1 mM phenylmethylsulfonyl fluoride (PMSF), 1 mM DTT and cOmplete protease inhibitor cocktail (Roche)) and lysed by sonication on ice. Cell lysates were cleared by centrifugation at 16,000g for 10 min at 4 °C. For purification of proteins, the cell lysates were incubated with GS 4B beads (GE Healthcare) for 1 h rotating at 4 °C. Beads were washed three times by pelleting at 300g for 30 s at 4 °C and resuspension in GST-wash buffer (20 mM HEPES, pH 7.5, 150 mM NaCl and 5 mM DTT). Elution from GSH beads was carried out by the addition of 30 units of thrombin protease and incubation at room temperature for 3 h. Atg19 was further purified over a high-load size exclusion Superdex200 column. The purified protein was stored at −80 °C.

### In vitro binding assay

GST fusion protein-coupled beads were incubated with purified mCherry–Atg19 or Atg19 from *E. coli* or GFP–Atg11 from insect cells for 20 min at room temperature. For microscopy analysis, unwashed samples were transferred into a Chamber μ-Slide VI 0.1 (Ibidi) and subjected to fluorescence microscopy. Afterwards samples were washed eight times with GST-wash buffer (20 mM HEPES, pH 7.5, 150 mM NaCl and 5 mM DTT) and subjected to further imaging. For immunoblotting analysis, samples were washed six times with GST-wash buffer containing 0.02% Tween-20.

### Protein expression in Sf9 insect cells

StrepII–GFP-Atg11 was expressed from baculovirus-infected Sf9 *Spodoptera frugiperda* insect cells (Expression Systems, cat. no. 94-001F). The ATG11 ORF was subcloned with an N-terminal StrepII–GFP tag into a pLIB library vector[50]. The recombinant bacmid carrying StrepII–GFP-Atg11 was assembled in DH10EMBacY *E. coli* strain (Geneva Biotech). For StrepII–GFP-Atg11 expression, Sf9 cells were grown in 1 l ESF 921 Insect Cell Culture Medium (Expression Systems) supplemented with penicillin and streptomycin at 27 °C to 1 × 10⁶ cells per ml, infected by the addition of 1 ml V3 virus, and grown for 4 days at 27 °C. Cells were pelleted at 500g for 10 min at room temperature and washed with 1× PBS, pH 7.4. Obtained pellets were resuspended in lysis buffer (500 mM Tris, pH 7.4, 1.5 M KCl, 50 mM $MgCl_2$, 10% glycerol, 5 mM β-mercaptoethanol, 1 mM PMSF, 0.1% Triton X-100, 10 mM imidazole and complete protease inhibitor cocktail (Roche)). Three freeze–thaw cycles were performed. Cell lysates were frozen in liquid nitrogen followed by thawing at 4 °C. Benzonase (25 U ml⁻¹; Merck) was added and cells were incubated for 10 min on ice. Cell lysates were cleared three times by centrifugation at 15,000g for 10 min at 4 °C and the supernatant was transferred to a new microfuge tube each time. The supernatants were recovered and

applied to a Strep-TactinXT 4Flow. The column was washed with Buffer W (100 Tris-HCl, 150 mM NaCl and 1 mM EDTA, pH 8.0) and the protein was eluted with Buffer BXT (100 Tris-HCl, 150 mM NaCl, 50 mM biotin and 1 mM EDTA, pH 8.0).

## In vitro Atg11 analysis

Purified GFP–Atg11 was diluted to a concentration between 0.005 μM and 2 μM in buffer containing 20 mM HEPES pH 7.5, 150 mM NaCl and 5 mM DTT. For condensate formation and condensate coalescence, samples were incubated at room temperature for 20 min and transferred into a Chamber μ-Slide VI 0.1 (Ibidi) and subjected to fluorescence microscopy. Coalescence of GFP–Atg11 droplets was followed for 60 min with a time interval of 10 s.

## Vacuolar purification

Vacuoles from Vph1–4xmCherry *atg15Δ pep4Δ* or Vph1–4xmCherry *vac8Δ atg15Δ pep4Δ* cells were enriched. Cells were grown at 30 °C, and a minimum of 1,000 $OD_{600}$ units were collected, washed and treated in DTT-containing buffer (100 mM Tris-HCl, pH 9.4 and 10 mM DTT) for 20 min at 30 °C. The cells were spheroplasted in YPD containing 600 mM sorbitol using recombinant lyticase for 30 min at 30 °C. Spheroplasts were collected at 1,500*g* for 10 min at 4 °C, and the pellet was resuspended in 15% Ficoll (in PS200 buffer: 20 mM PIPES, pH 6.8 and 200 mM sorbitol supplemented with cOmplete protease inhibitor (EDTA-free, Roche)) with 0.08 μg per $OD_{600}$ unit DEAE-dextran. The samples were centrifuged at 20,000*g* for 10 min, 4 °C using an ultracentrifuge (Optima MAX-130 K Ultracentrifuge (Beckmann)). The pellet was taken up in 15% Ficoll and overlayed with 8%, 4% and 0% Ficoll solution. The gradient was centrifuged at 100,000*g* for 80 min, 4 °C (Sorvall WX Ultracentrifuge). The enriched vacuoles at the 0–4% interface were collected and concentrated at 20,000*g* for 20 min, 4 °C and finally, the vacuoles were taken up in PS200 buffer.

## Mammalian cell culture conditions and cell line generation

The following mammalian cell lines were used in this study: cCE308: U2OS Flp-In T-REx cell line with stably integrated FRB–FIS1[93–152], used for the live fluorescent microscopy experiments. cCE377: U2OS Flp-In T-REx cell line with stably integrated FRB–FIS1[93–152], mt-mKeima and 2×FKBP–GFP–ULK1, used for the mKeima assay experiments. cCE1: ATG13KO U2OS Flp-In T-REx cells. The ATG13KO was generated using CRISPR-Cas9, introducing a 4-bp deletion at position 242–245, resulting in a premature stop codon at amino acid 102. cRB7: stable HEK293 Flp-In T-REx cell line with stably integrated FRB–FIS1[93–152], mt-mKeima and 2×FKBP–APEX2–ULK1. cRB12: stable HEK293 Flp-In T-REx cell line with stably integrated FRB–FIS1[93–152], mt-mKeima and 2×FKBP–APEX2. Both cRB7 and cRB12 were used for the affinity purification/MS experiments.

All three cell lines, cCE377, cRB7 and cRB12, were created by integration of Flp-In expression vectors into either HEK293 (R78007, Thermo Fisher Scientific) or U2OS (K650001, Thermo Fisher Scientific) Flp-In T-rex cells and are hygromycin and blasticidin resistant.

Both HEK293 and U2OS Flp-In T-REx cell lines were grown in Dulbecco's modified Eagle's medium (DMEM; Sigma-Aldrich, D6429) containing 10% fetal bovine serum (FBS; Sigma, F7524-500ML), 5 U ml⁻¹ penicillin (Sigma, P4333-100ML) and 50 μg ml⁻¹ streptomycin (Sigma, P4333-100ML). The cells were incubated at 37 °C in a humidified 5% (v/v) $CO_2$-air atmosphere.

To generate the HEK293 and U2OS Flp-In T-REx cell lines, cells were seeded in six-well plates (Sarstedt, 83.3920), at a density of 500,000 per well and allowed to grow overnight in antibiotic-free medium. The following day, the cells were co-transfected with a 1:10 ratio of Flp-In expression vector (pCE70 for 2×FKBP–GFP–ULK1, pRB7 for 2×FKBP–APEX2–ULK1 and pRB30 for 2×FKBP–APEX2) to Flp-recombinase plasmid pOG44 (Thermo Fisher Scientific, V600520). After 24 h, the medium was changed with one supplemented with 15 μg ml⁻¹ blasticidin (InvivoGen, ant-bl-05). Further 24 h later, cells were expanded from the six-well plates to p100 dishes (Sarstedt, 83.3902.300) and the medium was changed to one containing 15 μg ml⁻¹ blasticidin (InvivoGen, ant-bl-05) and 100 μg ml⁻¹ Hygromycin B Gold (InvivoGen, ant-hg-1). The Hygromycin B Gold treatment allowed for selection of the cells that stably integrated the construct. The selection medium were then changed every 5 days. Colonies were then picked and transferred to 24-well plates (Sarstedt, 83.3922.005) and expanded. The cells were then checked for doxycycline-inducible target gene expression using western blotting. All cells were periodically tested for mycoplasma contamination.

## siRNA treatment

To knockdown the expression of endogenous FIP200, cells were treated with two different siRNAs for FIP200: ON-TARGETplus Human RB1CC1 siRNA (Dharmacon, J-021117-05-0010) and Human RB1CC1 siRNA HSS114818 (Invitrogen, 91044774) at a final concentration of 20 nM each. As a negative control, the wild-type cell lines were treated with an ON-TARGETplus non-targeting siRNA no. 1 (Dharmacon, D-001810-01-05). siRNAs were transfected using ViaFect (E4981, Promega) according to the manufacturer's instructions.

## Sample processing for mass spectrometry analysis

HEK293 Flp-In T-REx cells were freshly thawed and seeded, and after 24 h, the cells were treated with the indicated siRNA (noncoding or siFIP200) for 72 h before collection. The day before collecting, the cells were treated with 1 μg ml⁻¹ doxycycline (Sigma, D9891-10g) to induce the expression of the Flp-In integrated construct: 2×FKBP–myc–APEX2 for cRB12 and 2×FKBP–myc–APEX2–ULK1 for cRB7. On the day of the collection, 72 h after siRNA treatment, the cells were treated with 500 μM of biotin–phenol (IrisBiotech, 41994-02-9/LS-3500.5000) and when indicated, with 0.5 μM rapalogue (A/C Heterodimerizer, TaKaRa, 635056) for 1 h at 37 °C. To induce the peroxidase activity of APEX2 and the formation of a biotinylated area, the cells were treated for 1 min with 1 mM $H_2O_2$ (Roth, CP26.5). Quickly after the 1 min $H_2O_2$ pulse, the cells were washed with DPBS (Sigma, D8537-500ML) and, to avoid further biotinylation from the APEX2, the cells were washed with quenching buffer (DPBS, 10 mM sodium ascorbate (Sigma, 11140-50G) and 5 mM Trolox (Sigma, 238813-25G)). The cells were then scraped, collected and lysed in RIPA buffer (50 mM Tris, 150 mM NaCl, 0.1% SDS, 0.5% sodium deoxycholate, 1% Triton X-100 and 1× protease inhibitor cocktail; Roche, 05056489001), supplemented with quenching reagents (10 mM sodium ascorbate and 1 mM Trolox). To clear the lysate, the cells were centrifuged at 10,000*g* at 4 °C for 15 min.

Magnetic streptavidin beads (Pierce, Thermo Scientific, 88816) were chemically acetylated using Sulfo-NHS-acetate (Pierce, Thermo Scientific, 26777) to prevent contamination with co-digested streptavidin peptides[51,52]. The supernatant of the cell lysate was incubated for 1 h at 4 °C with the S-NHS-Ac treated magnetic streptavidin beads. After incubation, the samples were washed five times with Tris buffer (50 mM Tris and 150 mM NaCl) and three times with 50 mM ABC buffer (ammonium bicarbonate, Sigma, 09830-1KG). The beads were transferred to a new tube and resuspended in 50 μl 1 M urea and 50 mM ammonium bicarbonate. Samples were reduced with 2 μl of 250 mM DDT (Roche, 10708984001) for 30 min at room temperature and alkylated with 2 μl 500 mM iodoacetamide (Sigma, I6125-5G) for 30 min at room temperature in the dark. The remaining iodoacetamide was quenched with 1 μl 250 mM DTT for 10 min. Proteins were digested with 150 ng LysC (FUJIFILM Wako Pure Chemical Corp., 125-02543) at 25 °C overnight. The supernatant was transferred to a new 0.2-ml vial and further digested for 5 h at 37 °C by the addition of 150 ng trypsin (Trypsin Gold, Mass Spectrometry grade, Promega, V5280). The digest was stopped by the addition of trifluoroacetic acid to a final concentration of 0.5%, and the peptides were desalted using a 96-well OASIS HLB μElution plate (Waters, 30-μm particle size, 186001828BA) following the manufacturer's protocol.

## Liquid chromatography–mass spectrometry analysis

Peptides were separated on an Ultimate 3000 RSLC nano-flow chromatography system (Thermo Fisher), using a pre-column for sample loading (Acclaim PepMap C18, 2 cm × 0.1 mm, 5 µm, Thermo Fisher) and a C18 analytical column (Acclaim PepMap C18, 50 cm × 0.75 mm, 2 µm, Thermo Fisher), applying a segmented linear gradient from 2% to 35% and finally 80% solvent B (80% acetonitrile, 0.1% formic acid; solvent A 0.1% formic acid) at a flow rate of 230 nl min$^{-1}$ over 60 min.

Eluting peptides were analysed on an Exploris 480 Orbitrap mass spectrometer (Thermo Fisher), which was coupled to the column with a FAIMS pro ion-source (Thermo Fisher) using coated emitter tips (PepSep, MSWil). The mass spectrometer was operated in DIA (Data-Independent Acquisition) mode with the FAIMS CV set to −45, the survey scans were obtained in a mass range of 350–1,200 m/z, at a resolution of 60,000 at 200 m/z and a normalized AGC target at 300%. A total of 31 MS/MS spectra were aquired with variable isolation width between 13 and 257 m/z covering 349.5–1,200.5 m/z range, including 1 m/z windows overlap, selected precursors were fragmented in the HCD (higher-energy collision induced dissociation) cell at 30% collision energy at a normalized AGC target of 1,000% and a resolution of 30,000. The maximum injection time was set to auto.

## Mass spectrometry data analysis

MS raw data were converted to the htrms format using HTRMS converter (v.18.3, Biognosys) and processed with Spectronaut (v.18.5, Biognosys). The library-free DirectDIA+ workflow was employed for analysis of the htrms files, utilizing the *Homo sapiens* one protein per gene reference proteome from UniProt (Proteome ID: UP000005640, release 2023.03), concatenated with a database of 379 common laboratory contaminants (in-house database) and an entry for the 2×FKBP–APEX2 construct. The cleavage specificity was set to full trypsin specificity (Trypsin/P), allowing for two missed cleavages. The thresholds for precursor *q*-value, precursor PEP, protein *q*-value per experiment, protein *q*-value per run and protein PEP were all set at 1% (ref. 53). Carbamidomethylation of cysteine residues was set as fixed modification; methionine oxidation and protein N-terminal acetylation were considered as variable modifications. Cross-run normalization was disabled and all other settings were used at their default values.

Computational analysis was performed using Python and the in-house developed Python library MsReport, v.0.0.23 (ref. 51). Only non-contaminant proteins identified with a minimum of two peptides and being quantified in at least two replicates of one experiment were considered for the analysis. MS2 LFQ protein intensities reported by Spectronaut were used for the quantitative analysis. Intensities below 1,000 were removed and treated as not quantified to exclude low-quality quantification. Intensities were log$_2$ transformed and normalized across samples using the ModeNormalizer from MsReport. This method involves calculating log$_2$ protein ratios for all pairs of samples and determining normalization factors based on the modes of all ratio distributions. Missing values were imputed by drawing random values from a normal distribution with $\mu = 9.96$ and $\sigma = 0.75$.

Statistical analysis comparing experiments FKBP–APEX2–ULK1 (WT) with FKBP–APEX2 (WT) and FKBP–APEX2–ULK1 (siFIP200) with FKBP–APEX2 (siFIP200) was performed using the linear models for microarray analysis (limma) v.3.54.2 (ref. 54) package in R. Moderated *t*-statistics were calculated using the limma-trend method, and multiple testing correction was applied using the Benjamini–Hochberg method. GO terms were obtained from UniProt (release 2023.05). The in-house Python library XlsxReport (https://github.com/hollenstein/xlsxreport) was used to create a formatted Excel file summarizing the results of protein quantification (Source data).

To assess the impact of FIP200 knockdown, ULK1 target proteins were selected based on an adjusted *P* value < 0.01 and log$_2$ ratio > 1 in FKBP–APEX2–ULK1 (WT) versus FKBP–APEX2 (WT). Average log$_2$ protein intensities were calculated for each control, FKBP–APEX2 (WT) and FKBP–APEX2 (siFIP200). The mean of the respective control was subtracted from each replicate of the FKBP–APEX2–ULK1 WT and FKBP–APEX2–ULK1 siFIP200 samples, resulting in values denoted as 'Signal over control (log$_2$)'. Focusing on ULK1-specific proteins associated with the GO term 'ER membrane' ($n = 44$), *t*-tests were calculated between FKBP–APEX2–ULK1 (WT) and FKBP–APEX2–ULK1 (siFIP200) experiments using log$_2$ 'signal over control' values. Multiple testing correction was applied using the Benjamini–Hochberg method with a false discovery rate-controlled *P* value cutoff of 0.05. The results are summarized in Source data.

## Fluorescence microscopy of mammalian cells

For live-cell imaging, cells were grown in 35-mm glass-bottom dishes (Ibidi, 81156) and preserved in an environmental chamber at 37 °C, 5% CO$_2$ during image acquisition. Fluorescent signal from U2OS cells was imaged using a DeltaVision OMX Flex Microscope with UPlanSApo ×60/1.4 oil Olympus objective, using a PCO Edge 4.2 sCMOS camera. Cells were treated with 4 µM antimycin A (A8674, Sigma-Aldrich) and 4 µM oligomycin (75351, Sigma-Aldrich) for 1 h; 0.5 µM rapalogue (B/B Homodimerizer, TaKaRa, 635059) treatment was performed for 24 h. Mitochondria were stained with MitoTracker DeepRed FM (Invitrogen, M22426) according to the manufacturer's instructions. ULK1 recruitment to p62 condensates was induced with EBSS (Sigma, E3024) for 4 h at 37 °C before imaging (Fig. 5c).

## Sample preparation for mKeima assay and sample analysis

U2OS cells (cCE377) were seeded in six-well plates at a density of 500,000 cells per well and cultured in DMEM supplemented with 10% FBS. After 24 h, the medium in the plates were replaced with one containing 1 µg ml$^{-1}$ of doxycycline to induce the expression of 2× FKBP–GFP–ULK1. Additionally, in the specified samples, 0.5 µM of rapalogue was added for a 24 h incubation period. Before collection, the indicated samples were treated with 200 nM bafilomycin A1 (Cell Signaling Technology, 54645) and rapalogue. Following the treatment, cells were washed with DPBS and detached with trypsin-EDTA (Sigma, T3924). The cells were then collected in FACS medium (phenol-red-free DMEM, Sigma, D1145-500ML, supplemented with 10% FBS). Cells were transferred to 1.5-ml tubes and centrifuged at 500g for 3 min at room temperature. The supernatant was removed, and the cell pellets were resuspended in 200 µl of FACS medium and transferred to U-bottom 96-well plates (Greiner Bio-One, 650970). Flow cytometry experiments were performed with a CytoFLEX S (Beckman Coulter, B75408) using CytExpert 2.3 analysis software.

Neutral pH mKeima and acidic pH mKeima were excited with the 405 nm and 561 nm lasers, respectively. For both mKeima forms, the emission was detected with a 610/20 bandpass filter. To detect GFP-positive cells, a 488 nm laser was used in combination with a 525/40 nm bandpass filter. For each sample, 200,000 events were collected and analysed using FlowJo (FlowJo v.10.9.0, 5 May 2023). The main population of live cells was selected via gating of SSC-A (side scatter area) versus FSC-A (forward scatter area) plots. Singlets were then selected in FSC-H (forward scatter height) versus FSC-W (forward scatter width) plots. Only GFP-positive cells were selected for the analysis, capturing a minimum of 80,000 cells per sample. Scale values of selected populations were then exported from FlowJo (Extended Data Fig. 8).

The in-house developed Python library mKeima (v.0.6.0, available at https://pypi.org/project/mkeima) was utilized for additional analysis.

The mKeima ratio was calculated for each individual event by dividing the acidic pH signal by the neutral pH signal.

For each sample, that is each replicate of each condition, the mean mKeima ratio was calculated and then scaled by using the average of the bafilomycin replicates as a low reference value and the average

of the highest rapalogue replicates as a high reference value. The low reference value was then set to 2 and the high reference value to 100.

$$x_{sc} = \frac{x - \bar{x}_{low}}{(\bar{x}_{high} - \bar{x}_{low})} \times (100 - 2) + 2$$

$x$, mean mKeima ratio of a sample

$x_{sc}$, scaled mean mKeima ratio of sample

$\bar{x}_{low}$, replicate average of the low scaling condition

$\bar{x}_{high}$, replicate average of the high scaling condition

## Plot profile quantification

The intensity peak-to-peak quantification was performed on similarly sized p62–FIP200 complexes. Profile plots were generated using Fiji, and the distance between intensity peaks was measured. To assess the difference in ER proximity between FIP200 and p62 structures, the distance of FIP200–ER was subtracted from the distance of p62–ER. The results were plotted as the frequency at which FIP200 was found to be closer to the ER than p62.

## 3D surface reconstruction

3D image reconstruction was performed using Imaris Image Analysis Software v.10.2 (Oxford Instruments Andor). The three-channel images (blue, green and red) were imported into Imaris, where surface segmentation was carried out using the machine-learning tool. Each channel was processed to isolate the specific structures, with the surfaces generated based on intensity thresholds and smoothed to reduce noise.

## FRAP in mammalian cells

FRAP analysis of mCherry–p62, GFP–p62–GFP and 2×FKBP–GFP–ULK1 structures was performed in an environmental chamber maintained at 37 °C and supplied with 5% CO$_2$. The image resolution was set to 128 pixels, and the excitation wavelengths were 561 nm for mCherry–p62 and 488 nm for 2×FKBP–GFP–ULK1 and GFP–p62–GFP. For each sample, a minimum of 13 frames were collected at a time interval of 4 s. Among these frames, one frame was captured before bleaching and at least 11 frames were collected immediately after bleaching. The average fluorescence intensities were recorded within the bleached region, the entire mCherry–p62 and GFP–p62–GFP or FKBP–GFP–ULK1-positive structures, and at a random condensate-free spot to serve as background for subtraction. These data were collected both before and immediately after photobleaching. Double normalization was performed as described previously[48].

## Cell lysis and immunoblot quantification

To induce bulk autophagy and degradation of p62, cells were starved for 4 h in EBSS (Sigma, E3024). After 4 h the cells were washed with DPBS and collected using a cell scraper. Cells were transferred to 15-ml tubes and centrifuged at 500$g$ for 3 min. After discarding the supernatant, cell pellets were resuspended in lysis buffer (50 mM Tris, pH 8, 1 mM EDTA, 150 mM NaCl, 0.5% sodium deoxycholate, 0.1% SDS, 1% Triton X-100, 1 mM PMSF, 1 mM NaF, 20 mM β-glycerophosphate and cOmplete protease inhibitor cocktail). The lysates were clarified by centrifugation at 10,000$g$ at 4 °C for 15 min. Protein concentrations were determined using the Pierce BCA Protein Assay kit (Thermo Fisher, 23227), and normalized to equal amounts across samples. Laemmli loading buffer was added and the samples were boiled at 95 °C for 5 min. Proteins were resolved by SDS–PAGE and analysed by immunoblotting. The ratio of free GFP/2×GFP–p62 was quantified from four independent replicates using FIJI and normalized to the wild-type control.

## Statistics and reproducibility

At least three successful independent biological replicates were performed for each experiment, as indicated in the Methods, except two successful biological replicates for Extended Data Fig. 7b, and three

technical replicates for Fig. 1e,f and Extended Data Figs. 3a,f,g and 4c,d. No data were excluded from analysis, except for clear technical failure. To assess statistical significance, one-tailed unpaired $t$-tests, two-tailed unpaired $t$-tests, one-way ANOVA or two-way-ANOVA tests followed by a Dunnett's post hoc test or Sidak's multiple comparisons test were performed, as specified in the figure legends. Data distribution was assumed to be normal but this was not formally tested. For all fluorescence microscopy image acquisition, cells were selected at random from the brightfield channel without bias toward the fluorescence signal. Samples for immunoblots were not randomized. For some experiments, blinding of the manual data analysis was not performed as all cells in the images were analysed and field of views were selected from the brightfield without previous knowledge of the fluorescence signal (Figs. 1d and 2f and Extended Data Figs. 3e and 5b). For all other fluorescence microscopy experiments cells were selected at random from the brightfield channel without bias toward the fluorescence signal. Analysis of fluorescence images was either performed computationally (no blinding necessary) (Figs. 1b,c and 3a,g and Extended Data Fig. 1c,g,h) or when manual performed quantification of fluorescence microscopy images was performed blindly after randomizing file names (all others). No statistical methods were used to predetermine sample sizes but our sample sizes are similar to those reported in previous publications[21,31,44].

## Correlative cryo-electron tomography

GFP-Atg19 cells expressing Ape1 under a copper-inducible promoter were grown at 30 °C in YPD supplemented with 250 μM CuSO$_4$ for induction of Ape1 overexpression. Upon reaching OD$_{600}$ = 0.8, rapamycin was added to the culture at a final concentration of 100 nM for 3 h to induce phagophore biogenesis. The culture was diluted to an OD$_{600}$ of 0.8 and autofluorescent 1 μm diameter Dynabeads (Dynabeads MyOne carboxylic acid no. 65011, Thermo Fisher Scientific) were added at a 1:20 dilution before vitrification. EM grids (200 Mesh Au SiO$_2$ R1/4, Quantifoil) were glow discharged for 90 s on both sides using a Pelco easiGlow device. Then, 3.5 μl cell suspension was applied to each grid and cells were vitrified using an EM GP2 grid plunger (Leica Microsystems) after 2-s blotting time at 20 °C and 90% relative humidity by plunge freezing in liquid ethane at −184 °C. Grids were clipped using modified autogrid rings with cut-outs suitable for cryo-FIB-milling. Cryo-confocal imaging of the plunge frozen samples was carried out on a TCS SP8 Cryo-CLEM (Leica Microsystems) equipped with a ×50/0.9 NA objective. For this, grids were initially mapped by collecting image stacks at 2 × 2 binning with a Z-spacing of 1.5 μm in brightfield and widefield GFP fluorescence channels. Grid maps were used to identify intact grid squares containing cells with the target fluorescent signal. For confocal imaging of grid squares containing cells with the target fluorescent signal, image stacks with a Z-spacing of 300 nm and 84.4 nm pixel size were acquired with 488 nm laser excitation. Grid mapping and cryo-confocal imaging were both carried out using the LAS X Navigator software (Leica Microsystems). Cryo-confocal stacks were deconvolved using Huygens Essential v.23.04.0p0 (Scientific Volume Imaging; http://svi.nl) and then resliced to achieve cubic voxels using Fiji[46]. For FIB-milling with an Aquilos dual-beam cryo-focused ion beam-scanning electron microscope (cryo-FIB-SEM) (Thermo Scientific), equipped with a cryo-stage cooled to −180 °C, grids were first coated with a protective organometallic platinum layer for 9 s and then mapped by cryo-scanning electron microscopy. Grid maps were correlated to the cryo-fluorescence maps using MAPS Software (Thermo Scientific) and lamella sites were placed in grid squares in which cryo-confocal stacks had been collected. Semi-automatic FIB-milling of cells with the target fluorescent signal was then carried out at 8° milling angle using a Gallium ion beam and AutoTEM software (Thermo Scientific). The ion beam current and distance between milling patterns was decreased in a stepwise manner to achieve a final lamella thickness of 150–200 nm.

Tilt series were acquired on a Krios G4 (Thermo Scientific) cryo-transmission electron microscope operated at 300 kV and equipped with a Selectris X energy filter, an energy slit set to 10 eV and Falcon 4i direct electron detector at a nominal magnification of ×64,000, corresponding to a 1.197 Å pixel size, using SerialEM[55]. A dose-symmetric acquisition scheme[56] was used with the start angle set to 8° and a nominal tilt range of 68° to −52° with 2° increments, a target defocus range of −1.5 to −4.5 μm and total dose of ~150 e/Å$^2$. Target areas for tilt series acquisition were identified via correlation of the cryo-confocal stacks with low-magnification transmission electron microscope grid square images acquired at a nominal magnification of ×470, using 3D-Correlation Toolbox (https://3dct.semper.space/), with the Dynabeads serving as fiducial markers visible in both imaging modalities.

### Tomogram reconstruction and membrane segmentation
Tilt series alignment and tomogram reconstruction via weighted back projection were carried out at bin4 using AreTomo v.1.3.3 (ref. 57). Tomogram denoising at bin4 was carried out with cryoCARE (https://github.com/juglab/cryoCARE_pip)[58] trained on tomograms reconstructed from odd and even tilt series frames with identical alignment parameters. Automatic segmentation of membranes in the denoised tomograms was carried out by MemBrain-Seg (https://github.com/teamtomo/membrain-seg)[59], followed by manual refinement in Amira 2022.2 (Thermo Scientific) and display in ChimeraX-1.6.1 (https://www.cgl.ucsf.edu/chimerax)[60].

### Modelling
The mathematical modelling is described in the Supplementary Note 1.

### Reporting summary
Further information on research design is available in the Nature Portfolio Reporting Summary linked to this article.

## Data availability
The mass spectrometry proteomics data have been deposited to the ProteomeXchange Consortium via the PRIDE[61] partner repository with the dataset identifier PXD047277. All other data supporting the findings of this study are available from the corresponding author on reasonable request. The Python source data are available from the GitHub repository at https://github.com/hollenstein/sourcecode_mkeima-assay_licheva-et-al-2024 Source data are provided with this paper.

## Code availability
We used cellular_raza (https://github.com/jonaspleyer/cellular_raza) to solve the equations of motion and implement neighbour counting. More details can be found in Supplementary Note 1. The code for the in-house pipeline to process the FRAP data can be found on GitHub (https://github.com/CraignRush/FRAP-Processing).
The Python scripts used for processing and analysis of the mKeima assay data are available from the GitHub repository at https://github.com/hollenstein/sourcecode_mkeima-assay_licheva-et-al-2024.

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

## Acknowledgements

We thank J. Lizarrondo, A. Bieber and C. Capitanio for advice on the correlation workflow membrane segmentation, data collection and data processing, L. Lacante for input on quantification and M. Rössler for help with the graphics. We thank A. Simonsen and S. Migliano from the Stenmark laboratory for providing plasmids, B. Lace from the Ott laboratory for help with 3D image reconstruction using

the Imaris software, I. Valiya-Parambath for help with cloning and M. McDowell for advice on mutating the GBP–GFP interaction surface. We thank all the members of the Central Electron Microscopy facility at the Max Planck Institute of Biophysiks for support and advice and B. Turonova from the Molecular Sociology Department for the use of custom scripts for tomography image processing. We also thank the members of the MPI Brain Research Imaging Facility for their support and D. Tetiker for her help in developing the in-house FRAP analysis pipeline. Proteomics analyses were performed using the instrument park of the Vienna Biocenter Core Facilities. The Kraft laboratory has received funding from the Deutsche Forschungsgemeinschaft (DFG, German Research Foundation) project ID 450216812, C.K.; project ID 409673687, C.K.; GRK 2606 (project ID 423813989), C.K.; from the European Research Council (ERC) under the European Union's Horizon 2020 research and innovation program under grant agreement no. 769065, C.K. The Kraft, Timmer and Hunte laboratories were supported by the DFG SFB 1381 (project ID 403222702) C.K., J.T. and C.H.; and under Germany's Excellence Strategy (CIBSS - EXC-2189 project ID 390939984), C.K., J.T. and C.H. The Kraft and Wilfling laboratories acknowledge funding from the DFG SFB 1177 (project ID 259130777), C.K. and F.W. The Wilfling laboratory is also funded by the Max Planck Society and by the European Union (ERC, IntrinsicReceptors, 101041982), F.W. Views and opinions expressed are, however, those of the author(s) only and do not necessarily reflect those of the European Union or the ERC Executive Agency. Neither the European Union nor the granting authority can be held responsible for them. The Kraft laboratory thanks the SGBM graduate school for supporting their students. Work included in this study has also been performed in partial fulfilment of the requirements for the doctoral theses of M.L., R.B., J.E., J.J.N., J. Brenneisen. and C.E. at the University of Freiburg.

## Author contributions

Conceptualization: M.L., J.P., R.B., H.M., F.W. and C.K.; Methodology: M.L., J.P., R.B., H.M., J.E., E.B., D.H., J.J.N., J.P., M. Heinrich, F.G.W., J. Brenneisen, C.E., J. Brenner, S.J., M. Hartl, S.W., C.H., J.T., F.W. and C.K.; Validation: M.L., J.P., R.B., H.M., J.E., E.B., D.H., J.J.N., J.P., M. Hartl, F.G.W., J. Brenneisen, C.E., J. Brenner, S.J., M.Heinrich, S.W., C.H., J.T., F.W. and C.K.; Investigation: M.L., J.P., R.B., H.M., J.E., E.B., D.H., J.J.N., J.P., M. Heinrich., F.G.W., J. Brenneisen, C.E., J. Brenner and S.J.; Writing – original draft: F.W. and C.K.; Writing – review and editing: M.L., J.P., R.B., H.M., F.W. and C.K.; Supervision: M. Hartl., S.W., C.H., J.T., F.W. and C.K.; Project administration: F.W. and C.K.; Funding acquisition: M. Hartl., C.H., J.T., F.W. and C.K.

## Competing interests

The authors declare no competing interests.

## Additional information

**Extended data** is available for this paper at https://doi.org/10.1038/s41556-024-01572-y.

**Correspondence and requests for materials** should be addressed to Florian Wilfling or Claudine Kraft.

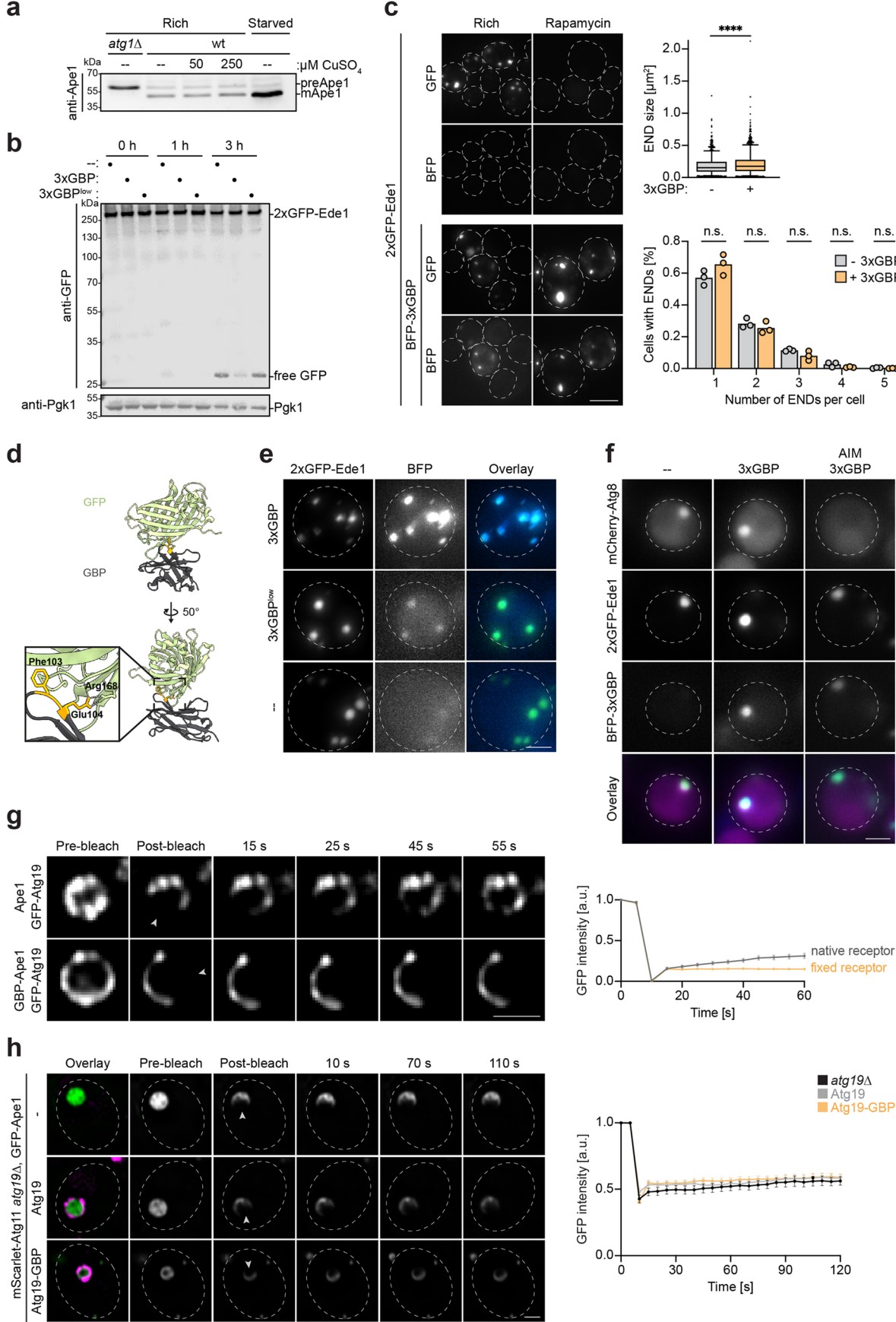

**Extended Data Fig. 1 | See next page for caption.**

**Extended Data Fig. 1 | High-affinity receptor–cargo interactions impair selective autophagy. a**, Wild-type and *atg1Δ* cells were grown to mid-log phase and treated as indicated. Cell extracts were analysed by immunoblotting. One out of three biological replicates with similar results is shown. **b**, Strains expressing 2×GFP-Ede1 and either no 3×GBP (−), +3×GBP or +3×GBP^low were treated with rapamycin and TCA precipitated. GFP cleavage was monitored by immunoblotting. One out of three biological replicates with similar results is shown. **c**, Representative fluorescence microscopy images from Fig. 1b. Scale bar: 5 μm. Quantification: area of ENDs in a box plot. Horizontal lines: median bound protein, box: 25th to 75th percentiles, whiskers: expand to 5th and 95th percentiles, outliers: black dots (n > 761 ENDs per condition and replicate, three biological replicates). Statistical analysis: two-tailed unpaired *t*-test. *P* value: *-3×GBP/+3×GBP P < 0.0001*, Quantification: number of ENDs per cell. Data are mean values (n > 170 cells per condition and replicate, three biological replicates). Statistical analysis: Multiple unpaired *t*-tests. *P* values: *-3×GBP:+3×GBP* (1) *P = 0.1425*, *-3×GBP:+3×GBP* (2) *P = 0.3669*, *-3×GBP/+3×GBP* (3) *P = 0.1243*, *-3×GBP/+3×GBP* (4) *P = 0.1087*, *-3×GBP/+3×GBP* (5) *P = 0.2158*. **d**, Crystal structure of GFP (green) bound to GBP (grey) (PDB: 3OGO). Inset shows binding interface disrupted by mutations of residues Phe103 and Glu104 (orange) made

in this study to create GBP^med and GBP^low. **e**, Representative images from Fig. 1b. Scale bar: 2 μm. **f**, Representative images from Fig. 1d. Scale bar: 2 μm. **g**, GFP–Atg19 cells expressing endogenous BFP–Ape1 and copper-inducible untagged Ape1 or GBP-BFP–Ape1 and copper-inducible untagged Ape1 were grown to mid-log phase in the presence of 50 μM CuSO₄. GFP-Atg19 structures were photobleached, and recovery of the signal was monitored. Scale bar: 1 μm. White arrowheads: photobleached area. Scale bar 1 μm. Quantification: recovery of the GFP signal. Data are mean values ± SEM (n = 28 structures per condition across replicates, three biological replicates). **h**, mScarlet–Atg11 *atg19Δ* cells expressing endogenous GFP–Ape1 and copper-inducible untagged Ape1 together with an empty control vector (-), Atg19 or Atg19–GBP were grown to mid-log phase in the presence of 50 μM CuSO₄. GFP–Ape1 structures were photobleached, and recovery of the signal was monitored Scale bar: 1 μm. White arrowheads: photobleached area. Scale bar 1 μm. Quantification: recovery of the GFP signal. Data are mean values ± SEM (n > 20 structures per condition across replicates, three biological replicates). For each panel, one example out of three biological replicates is shown. Source numerical data and unprocessed blots are available in source data, not significant (n.s.), arbitrary units (a.u.).

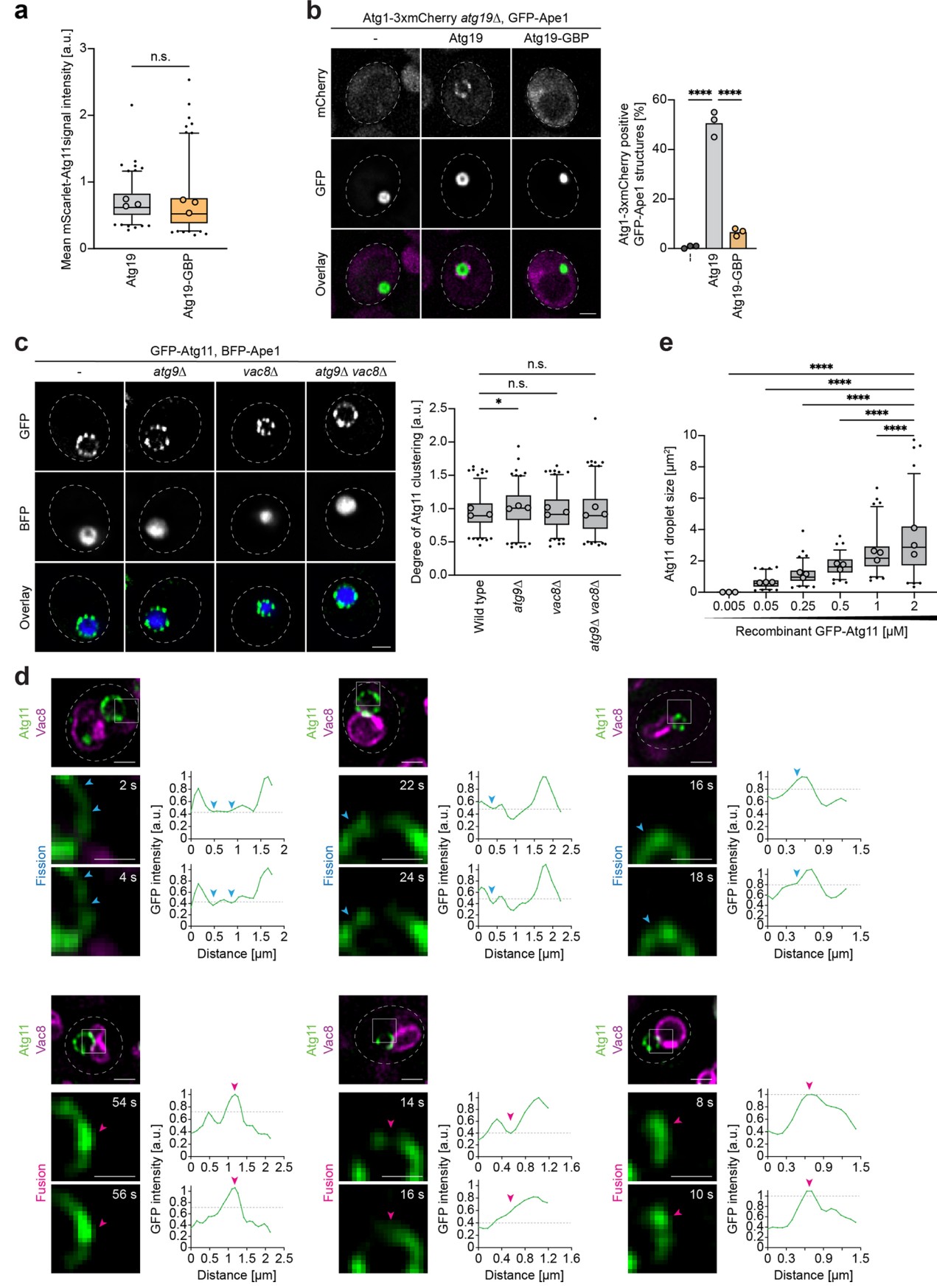

**Extended Data Fig. 2 | See next page for caption.**

**Extended Data Fig. 2 | Atg11 forms initiation hubs on selective cargo.**
**a**, mScarlet–Atg11 *atg19Δ* cells coexpressing either Atg19 or Atg19–GBP and endogenous GFP–Ape1 and copper-inducible untagged Ape1 were grown to mid-log phase in the presence of 50 μM CuSO₄. Quantification: mean mScarlet–Atg11 signal intensity in a box plot. Horizontal lines: median, box: 25th to 75th percentiles, whiskers: expand to 5th and 95th percentiles, circles: mean value of each replicate, outliers: black dots (n = 50 GFP–Ape1 positive particles per condition and replicate, three biological replicates). Statistical analysis: two-tailed unpaired *t*-test. **b**, Atg1-3xmCherry *atg19Δ* cells expressing endogenous GFP–Ape1 and copper-inducible untagged Ape1 together with an empty control vector (-), Atg19 or Atg19–GBP were grown to mid-log phase in the presence of 50 μM CuSO₄. Scale bar: 2 μm. Quantification: Atg1-3xmCherry positive GFP–Ape1 structures. Data are mean values (n = 100 structures per condition and replicate, three biological replicates). Circles: mean values of each replicate, bars: mean. Statistical analysis: Dunnett post hoc test. *P* values: –:Atg19 *P* < 0.0001, Atg19:Atg19–GBP *P < 0.0001*. **c**, Indicated cells expressing GFP–Atg11, endogenous BFP–Ape1 and copper-inducible untagged Ape1 were grown to mid-log phase in the presence of 50 μM CuSO₄. Scale bar: 2 μm. Quantification: GFP clustering as the coefficient of variance (SD/mean GFP intensity) in a box

plot. Horizontal lines: median, box: 25th to 75th percentiles, whiskers: expand to 5th and 95th percentiles, circles: mean value of each replicate, outliers: black dots (n = 50 structures per condition and replicate, three biological replicates). Statistical analysis: one-way ANOVA followed by Dunnett post hoc test. *P* values: *wild-type:atg9Δ P = 0.0498, wild-type:vac8Δ P = 0.9036, wild-type:atg9Δvac8Δ P = 0.9893*. **d**, Vac8-mCherry and GFP–Atg11 cells expressing endogenous BFP–Ape1 and copper-inducible untagged Ape1 were grown to mid-log phase in the presence of 50 μM CuSO₄. The dynamics of GFP–Atg11 foci were monitored and the fluorescence profile of GFP was measured along the Ape1 surface. Images of one out of three independent experiments are shown. Scale bar: 2 μm, scale bar inset: 1 μm. Time-lapse series are shown as Supplementary Videos 1–12. **e**, Quantification of GFP–Atg11 droplet size of Fig. 3c in a box plot. Horizontal lines: median, box: 25th to 75th percentiles, whiskers: expand to 5th and 95th percentiles, circles: mean value of each replicate, outliers: black dots (n = 30 structures per condition and replicate, three biological replicates). Statistical analysis: one-way ANOVA followed by a Dunnett post hoc test. *P* values: *0.005:2 P < 0.0001, 0.05:2 P < 0.0001, 0.25:2 P < 0.0001, 0.5:2 P < 0.0001, 1:2 P < 0.0001*. Source numerical data are available in source data, not significant (n.s.), arbitrary units (a.u.).

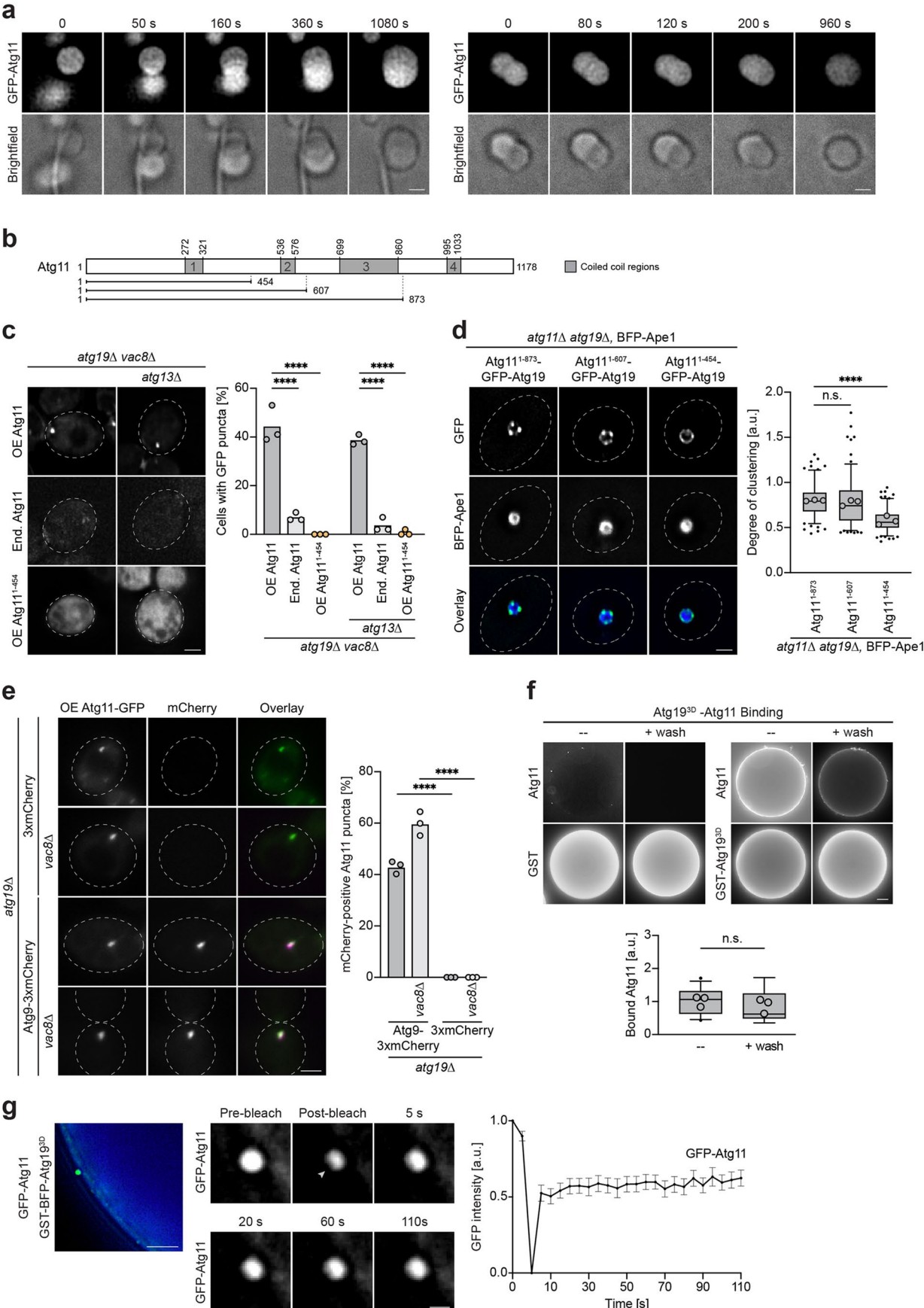

**Extended Data Fig. 3 | See next page for caption.**

**Extended Data Fig. 3 | Initiation hubs on selective cargo are dynamic.**
**a**, Further examples of coalescence of *in vitro* formed GFP–Atg11 droplets as shown and quantified in Fig. 3e. Scale bar: 2 μm. **b**, Schematic illustration of Atg11 domains. **c**, *atg19Δ vac8Δ* or *atg13Δ atg19Δ vac8Δ* were transformed with Atg11–GFP (OE Atg11) or Atg11$^{1-454}$-GFP (OE Atg11$^{1-454}$) overexpressed under a GPD promoter or Atg11–GFP expressed under its native promoter (End. Atg11) and grown to mid-log phase. Quantification: cells with GFP puncta. Data are mean values (n = 100 cells per condition and replicate, three biological replicates). Circles: mean values of each replicate, bars: mean. Statistical analysis: two-way ANOVA followed by a Tukey post hoc test. *P* values: *atg19Δvac8Δ: OE Atg11:End. Atg11 P < 0.0001, OE Atg11:OE Atg11$^{1-454}$ P < 0.0001; atg13Δ atg19Δ vac8Δ: OE Atg11:End. Atg11 P < 0.0001, OE Atg11:OE Atg11$^{1-454}$ P < 0.0001*. **d**, *atg11Δ atg19Δ* cells expressing endogenous BFP-Ape1 and copper-inducible untagged Ape1 along with Atg11$^{1-873}$-GFP–Atg19, Atg11$^{1-607}$-GFP–Atg19, or Atg11$^{1-454}$-GFP–Atg19 were grown to mid-log phase in the presence of 50 μM CuSO$_4$. Statistical analysis: one-way ANOVA followed by a Dunnett post hoc test. Scale bar: 2 μm. (n = 50 structures per condition and replicate, three biological replicates). *P* value: *Atg11$^{1-873}$-GFP–Atg19:Atg11$^{1-454}$-GFP–Atg19 P < 0.0001, Atg11$^{1-873}$-GFP–Atg19: Atg11$^{1-607}$-GFP-Atg1 P = 0.451*. **e**, Cells overexpressing Atg11–GFP under a GPD promoter and expressing Atg9-3xmCherry or 3xmCherry in *atg19Δ* or *atg19Δ vac8Δ* strains

were grown to mid-log phase. Scale bar: 2 μm. Quantification: co-localization of Atg9-3xmCherry and Atg11-GFP puncta. Data are mean values (n > 148 punctae per condition and replicate, three biological replicates). Circles: mean values of each replicate, bars: mean. Statistical analysis: two-tailed unpaired *t*-tests. *P* values: WT *P < 0.0001*, *vac8Δ P < 0.0001*. **f**, GST–BFP or GST–BFP-Atg19$^{3D}$ (a hosphor-mimetic mutant of Atg19 known to stably interact with Atg11, S390D, S391D, and S396D[29]) were expressed in *E. coli* and bound to Glutathione Sepharose (GSH) beads, incubated with Sf9 insect cell lysates containing overexpressed GFP–Atg11, and bound GFP–Atg11 was analysed. Scale bar: 20 μm. Quantification: ratio of bead-bound protein to soluble protein in a box plot. Horizontal lines: median, box: 25$^{th}$ to 75$^{th}$ percentiles, whiskers: expand to 5$^{th}$ and 95$^{th}$ percentiles, circles: mean value of each replicate, outliers: black dots (n > 18 beads per condition across replicates, three technical replicates). Statistical analysis: two-tailed unpaired *t*-test. *P values: P = 0.2102*. **g**, *In vitro* formed GFP–Atg11 droplets coupled to Atg19$^{3D}$ containing beads were photobleached, and recovery of the signal was measured. Quantification: recovery of the GFP signal. Data are mean ± SEM (n = 10 structures across replicates, three technical replicates). Source numerical data are available in source data, not significant (n.s.), arbitrary units (a.u.).

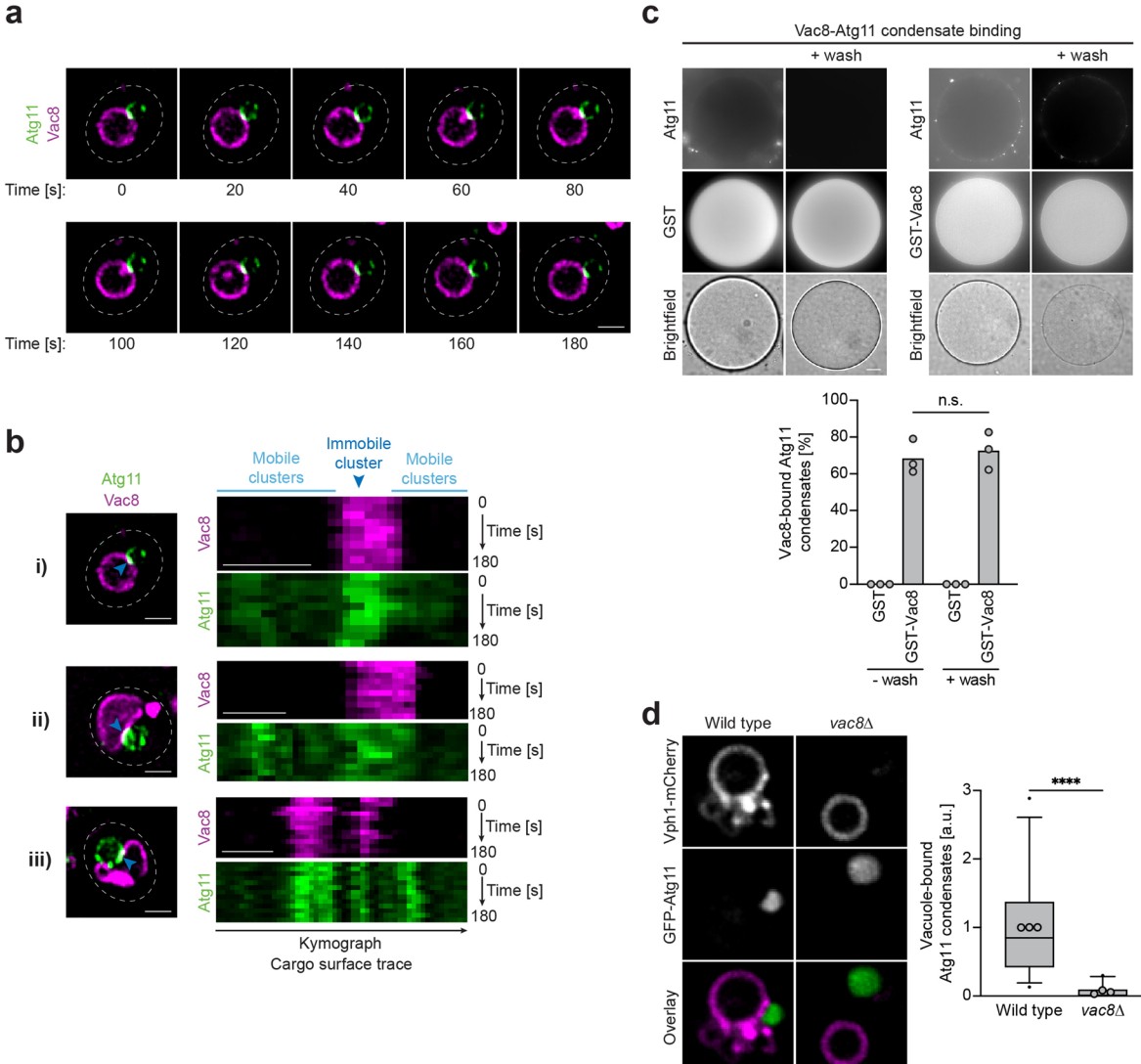

**Extended Data Fig. 4 | Initiation hubs coalesce at the vacuolar contact site to trigger phagophore initiation. a**, Vac8-mCherry GFP–Atg11 cells expressing endogenous BFP–Ape1 and copper-inducible untagged Ape1 were grown to mid-log phase in the presence of 50 µM CuSO₄. The dynamics of GFP–Atg11 foci were monitored. Scale bar: 2 µm. **b**, Kymograph corresponding to Extended Data Fig. 3a (i) and additional representative examples of kymographs (ii,iii). Scale bar: 2 µm. **c**, GST–BFP or GST–BFP–Vac8 were expressed in *E. coli* and bound to Glutathione Sepharose (GSH) beads. Protein-bound beads were incubated with Sf9 insect cell lysates containing overexpressed GFP–Atg11, and bound GFP–Atg11 was analysed before and after eight washing steps. Scale bar: 20 µm. Data

are mean values (n > 65 condensates per condition and replicate, three technical replicates). Circles: mean values of each replicate, bars: mean. Statistical analysis: two-tailed unpaired *t*-test. *P* values: *P* = 0.7. **d**, Purified vacuoles were incubated with recombinant GFP–Atg11 droplets. Scale bar: 1 µm. Quantification: Atg11 condensates bound to vacuoles in a box plot. Horizontal lines: median, box: 25th to 75th percentiles, whiskers: expand to 5th and 95th percentiles, circles: mean value of each replicate, outliers: black dots (n > 599 condensates per condition and replicate, three technical replicates). Statistical analysis: two-tailed unpaired *t*-test. *P* value: *vac8Δ:WT P < 0.0001*. Source numerical data are available in source data, not significant (n.s.), arbitrary units (a.u.).

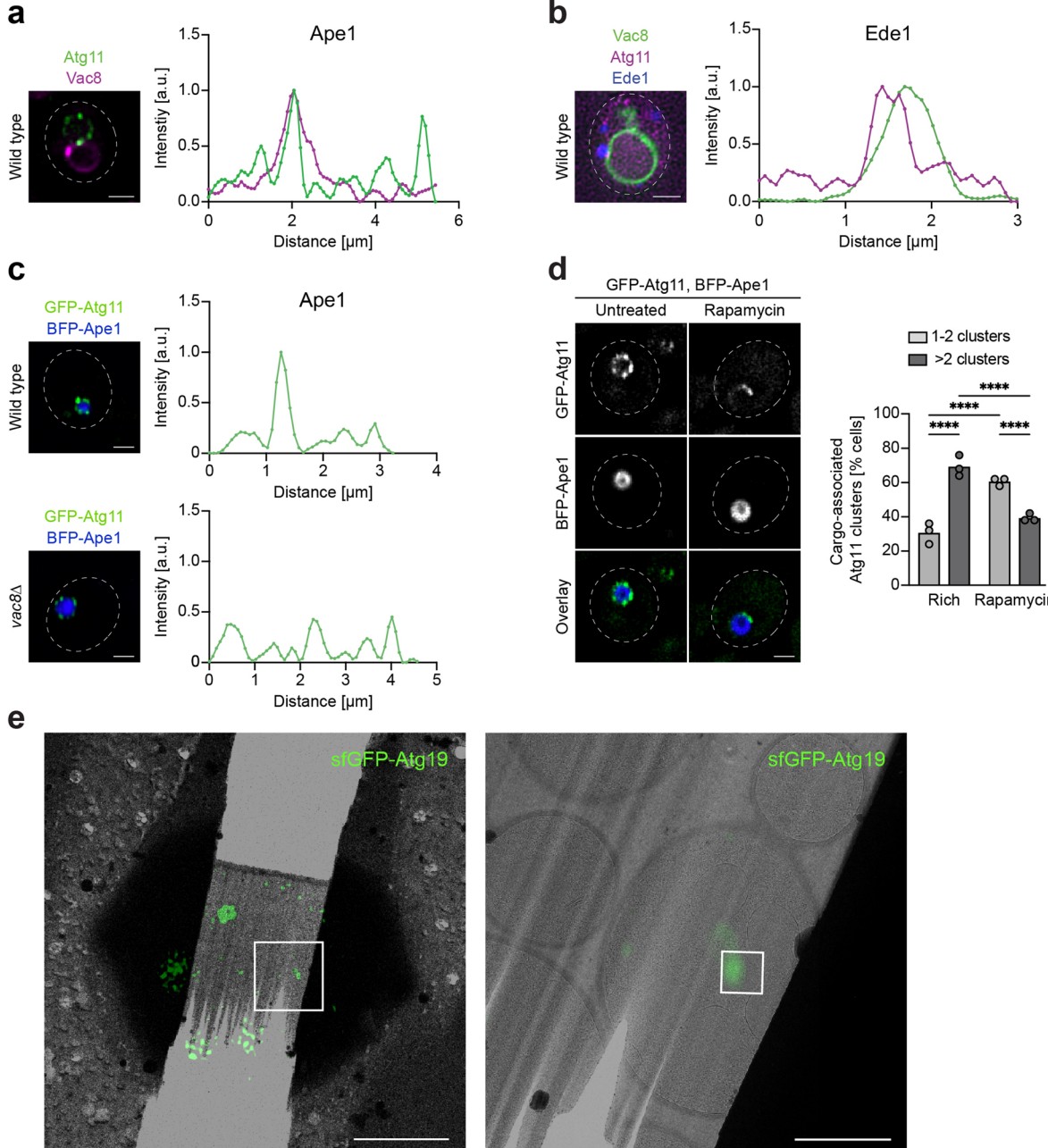

**Extended Data Fig. 5 | Autophagy initiation hubs are conserved in human cells. a**, GFP–Atg11 Vac8-mCherry cells expressing endogenous BFP–Ape1 and copper-inducible untagged Ape1 were grown to mid-log phase in the presence of 50 μM CuSO₄. The fluorescence profile of GFP (green) and mCherry (magenta) was measured along the Ape1 condensate surface. Images of one out of three independent experiments are shown. Scale bar: 2 μm. **b**, Cells expressing Ede1–BFP, mScarlet–Atg11, and Vac8-mNeon were grown to mid-log phase. The fluorescence profile of mNeon (green) and mScarlet (magenta) was measured along the Ede1 condensate surface. Images of one out of three independent experiments are shown. Scale bar: 2 μm. **c**, GFP–Atg11 or GFP–Atg11 *vac8Δ* cells expressing endogenous BFP–Ape1 and copper-inducible untagged Ape1 were grown to mid-log phase in the presence of 50 μM CuSO₄. The fluorescence profile of GFP (green) was measured along the Ape1 condensate surface. Images of one out of three independent experiments are shown. Scale bar: 2 μm. **d**, GFP–

Atg11 cells expressing endogenous BFP–Ape1 and copper-inducible untagged Ape1 were grown to mid-log phase in the presence of 50 μM CuSO₄. Atg11 rearrangement was monitored after rapamycin treatment. Quantification: BFP–Ape1 structures with rearranged Atg11 clusters. Scale bar: 2 μm. Data are mean values (n = 50 structures per condition and replicate, three biological replicates). Circles: mean values of each replicate, bars: mean. Statistical analysis: two-way ANOVA followed by Sidak's multiple comparison test. *P* values: *rich P < 0.0001, rapa P = 0.0005, rich(1-2):rapa(1-2) P < 0.0001, rich(>2):rapa(>2) P < 0.0001.* **e**, Overlays of maximum intensity projection of a GFP–Atg19 cryo-fluorescence stack on 470x (left) and 6500x (right) magnification TEM images used to select acquisition region for tomogram shown in Fig. 4e. Scale bars: 10 μm (left) and 2 μm (right). One out of three identified examples is shown. Source numerical data are available in source data, arbitrary units (a.u.).

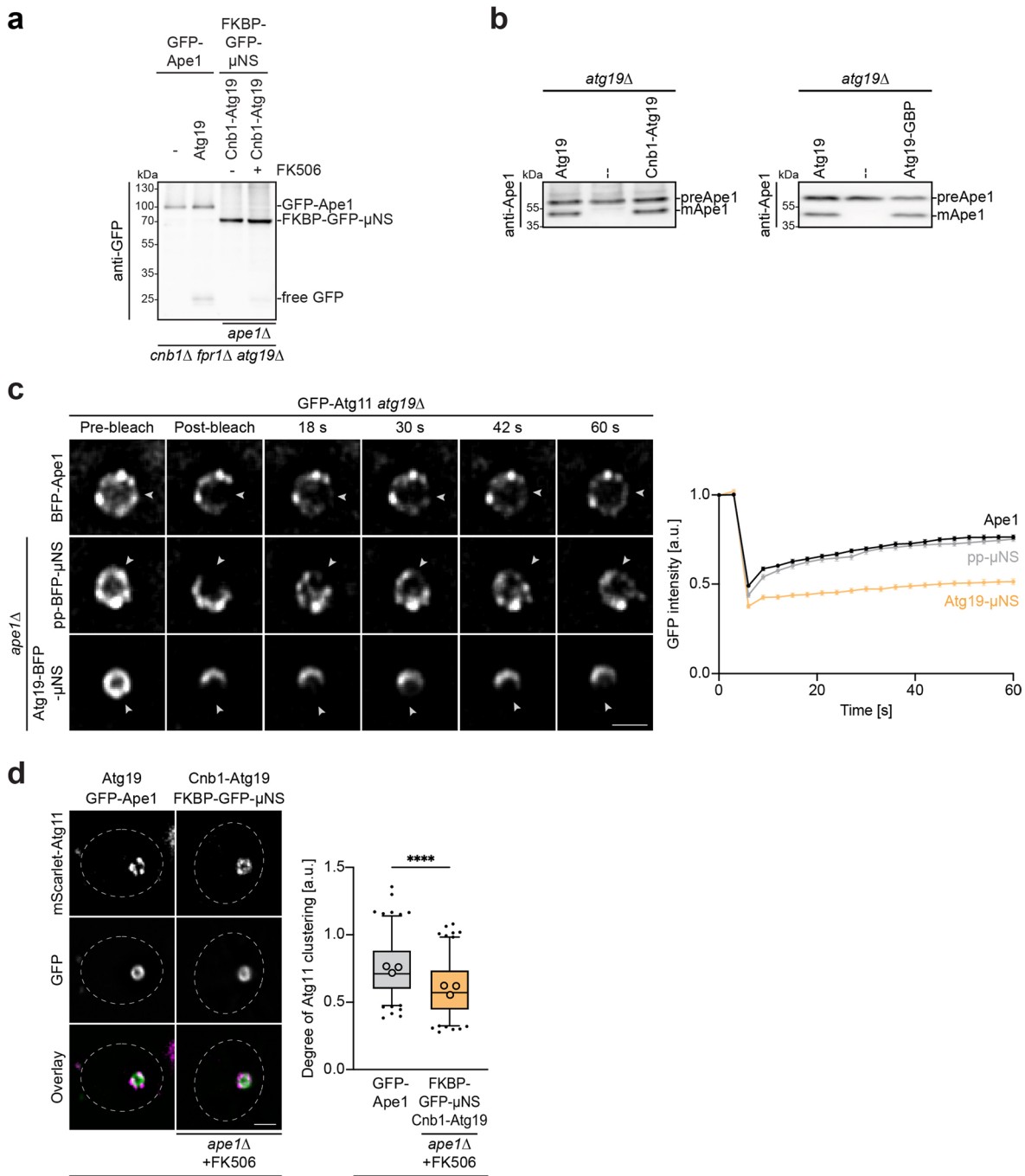

**Extended Data Fig. 6 | Degradation of a neo-cargo by selective autophagy.**
**a**, Indicated cells expressing Atg19 or Cnb1–Atg19, and GFP–Ape1 and copper-inducible untagged Ape1 or FKBP–GFP–μNS were grown to mid-log phase in the presence of 50 μM CuSO₄ and treated with FK506. GFP cleavage was monitored by TCA precipitation and immunoblotting. One out of three biological replicates with similar results is shown. **b**, atg19Δ cells expressing Atg19, an empty vector control (−), Cnb1–Atg19 or Atg19–GBP were grown to mid-log phase. Cell extracts were analysed by immunoblotting. One out of three biological replicates with similar results is shown. **c**, GFP–Atg11 atg19Δ cells expressing endogenous BFP–Ape1 and copper-inducible untagged Ape1 and GFP–Atg11 atg19Δ ape1Δ cells expressing pp-BFP–μNS or Atg19-BFP–μNS were grown to mid-log phase. GFP–Atg11 structures were photobleached, and recovery of the signal was monitored.

Scale bar: 1 μm. Quantification: recovery of the GFP signal. Data are mean values ± SEM (n > 26 structures per condition across replicates, three biological replicates). **d**, Cells from Extended Data Fig. 6a coexpressing mScarlet–Atg11 were analysed and quantified as in Fig. 2b. Scale bar: 2 μm. Quantification: GFP clustering as the coefficient of variance (SD/mean GFP intensity) in a box plot. Horizontal lines: median, box: 25th to 75th percentiles, whiskers: expand to 5th and 95th percentiles, circles: mean value of each replicate, outliers: black dots (n = 50 structures per condition and replicate, three biological replicates). One-tailed unpaired t-test. P value: P < 0.0001. For each panel, one out of three biological replicates is shown. Source numerical data and unprocessed blots are available in source data, arbitrary units (a.u.).

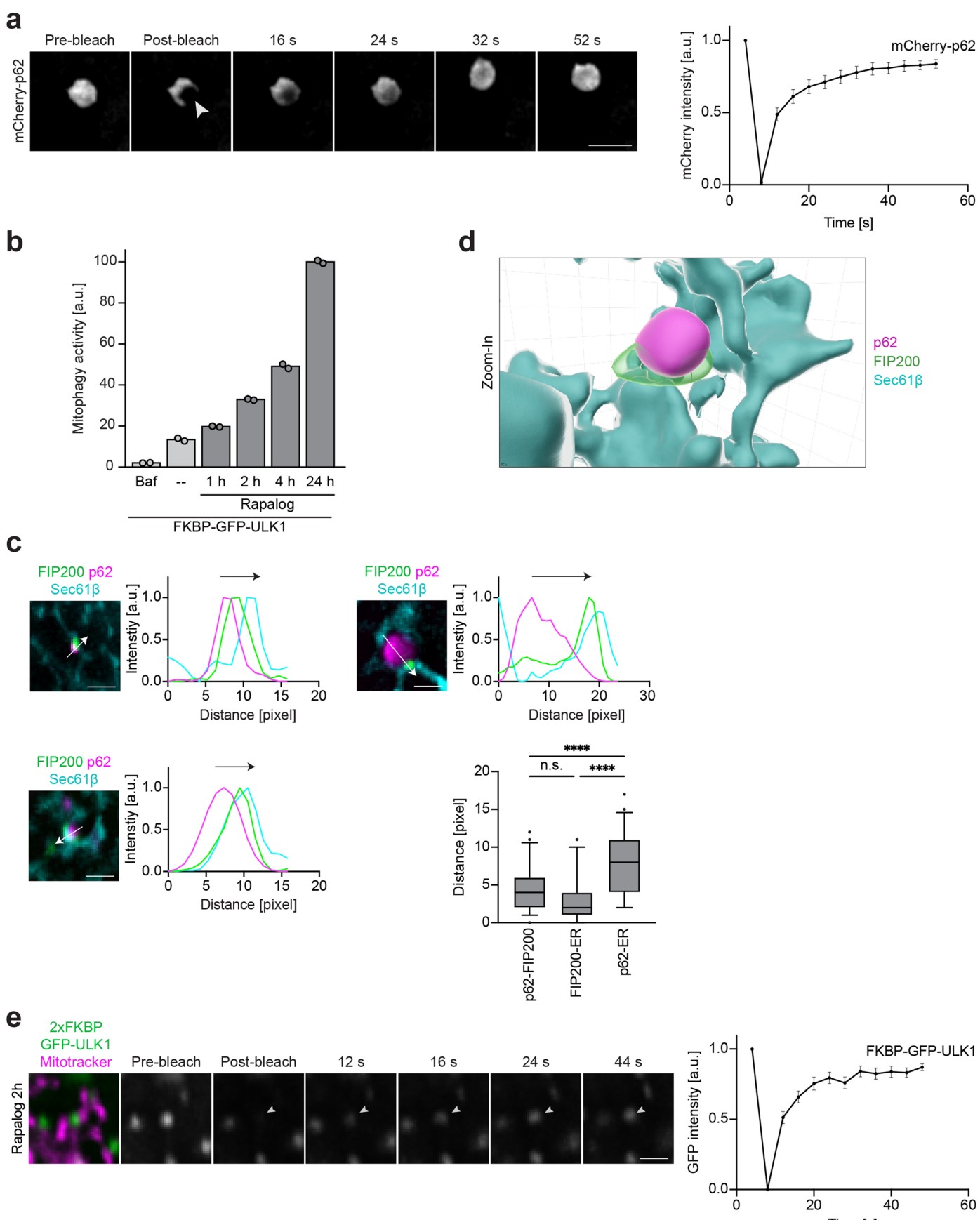

**Extended Data Fig. 7 | See next page for caption.**

**Extended Data Fig. 7 | Autophagy initiation hubs are conserved in human cells.**
**a**, U2OS cells were grown in nutrient-rich medium. mCherry–p62 condensates were photobleached and recovery of the signal was followed. A representative image is shown from one out of three biological replicates. The white arrowhead indicates the photobleached area. Scale bar: 1 µm. Quantification: recovery of the mCherry signal. Data are mean values ± SEM (n = 28 structures across three biological replicates). **b**, U2OS wild-type cells stably expressing mito-mKeima and FRB-FIS[93-152] and FKBP–GFP-ULK1 were grown in nutrient-rich medium and treated with Bafilomycin A1 (Baf) and rapalog as indicated. Gating for GFP-expressing cells was performed. The mito-mKeima ratio of lysosomal mitochondria (561 nm) to cytosolic mitochondria (488 nm) was analysed by flow cytometry and is shown as a ratio normalized to the Baf treatment. Data are mean values (n > 50,000 cells per condition and replicate, two biological replicates). Circles: mean values of each replicate, bars: mean. **c**, Further examples of Fig. 7f. Quantification: peak-to-peak signal distance in a box plot. Horizontal lines: median, box: 25th to 75th percentiles, whiskers: expand to 5th and 95th percentiles, outliers: black dots (n = 47 plot profiles across three

biological replicates). Statistical analysis: one-way ANOVA followed by Sidak's multiple comparison test. *P* values: *p62-FIP200:p62-ER P < 0.0001*, *FIP200-ER:p62-ER P < 0.0001*. **d**, Zoomed in 3D reconstruction of Fig. 7f. The 3D surface of BFP–Sec61β (blue), FKBP–GFP-FIP200 (green) and mCherry–p62 (magenta) was rendered with the IMARIS software using the machine learning tool for surface segmentation. A z-stack of 0.125 µm was taken to define the borders in z. **e**, U2OS cells stably expressing FRB-FIS1[93-152] and transfected with 2xFKBP-GFP-ULK1 were cultured in nutrient-rich medium and stained with MitoTracker DeepRed. Tethering of 2xFKBP-GFP-ULK1 to FRB-FIS[93-152] was induced by adding rapalog for 2 h. Mitochondria-associated 2xFKBP-GFP-ULK1 clusters were photobleached, and the recovery of the signal was monitored. Scale bar: 1 µm. Quantification: recovery of the GFP signal. Data are mean ± SEM (n = 28 structures across three biological replicates). White arrowheads indicate the photobleached cluster. The quantification and the representative image are the same as shown in Fig. 8b. Source numerical data are available in source data, not significant (n.s.), arbitrary units (a.u.).

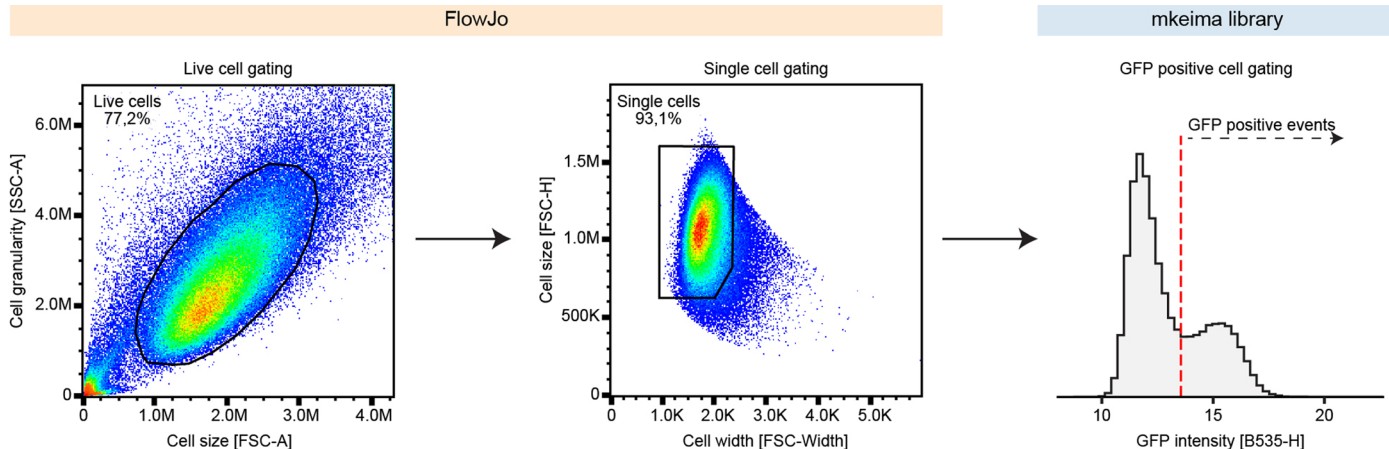

**Extended Data Fig. 8 | Gating strategy for flow cytometry.** Cell populations were gated through a series of steps to ensure data specificity and accuracy. First, live cells were selected using FlowJo software by plotting side scatter area (SSC-A) against forward scatter area (FSC-A), which allowed for the selection of a homogenous population based on cell granularity and size. Single cells were then gated by plotting forward scatter height (FSC-H) against forward scatter width (FSC-W) to eliminate doublets and aggregates. After gating, the data were exported as scale values and analysed using an in-house Python library mKeima (https://pypi.org/project/mkeima). GFP-positive cells were selected based on their signal in the B545-H channel. The analysis was then continued using the custom mKeima package.

| | |
|---|---|

# Reporting Summary

## Statistics

For all statistical analyses, confirm that the following items are present in the figure legend, table legend, main text, or Methods section.

| n/a | Confirmed | |
|---|---|---|
| ☐ | ☒ | The exact sample size (*n*) for each experimental group/condition, given as a discrete number and unit of measurement |
| ☐ | ☒ | A statement on whether measurements were taken from distinct samples or whether the same sample was measured repeatedly |
| ☐ | ☒ | The statistical test(s) used AND whether they are one- or two-sided<br>*Only common tests should be described solely by name; describe more complex techniques in the Methods section.* |
| ☒ | ☐ | A description of all covariates tested |
| ☒ | ☐ | A description of any assumptions or corrections, such as tests of normality and adjustment for multiple comparisons |
| ☐ | ☒ | A full description of the statistical parameters including central tendency (e.g. means) or other basic estimates (e.g. regression coefficient) AND variation (e.g. standard deviation) or associated estimates of uncertainty (e.g. confidence intervals) |
| ☐ | ☒ | For null hypothesis testing, the test statistic (e.g. *F*, *t*, *r*) with confidence intervals, effect sizes, degrees of freedom and *P* value noted<br>*Give P values as exact values whenever suitable.* |
| ☒ | ☐ | For Bayesian analysis, information on the choice of priors and Markov chain Monte Carlo settings |
| ☒ | ☐ | For hierarchical and complex designs, identification of the appropriate level for tests and full reporting of outcomes |
| ☒ | ☐ | Estimates of effect sizes (e.g. Cohen's *d*, Pearson's *r*), indicating how they were calculated |

*Our web collection on statistics for biologists contains articles on many of the points above.*

## Software and code

Policy information about availability of computer code

| | |
|---|---|
| Data collection | CryoEM: LAS X Navigator software (Leica Microsystems, version 3.5.7.23225); SerialEM, AutoTEM software (Thermo Scientific, version 2.4), 3D-Correlation Toolbox (https://3dct.semper.space/)<br>Fluorescence microscopy: Fluorescence microscopy images were captured using the DeltaVision OMX Flex microscope with AcquireSR software (version 4.5.10170-1), the DeltaVision Ultra High Resolution microscope with AcquireUltra software (version 1.2.2), the Leica Stellaris 5 system with Leica Application Suite X (version 4.6.,1.27508) software, or the Nikon Eclipse Ti2 microscope with NIS-Elements AR 5.10.00 software. Raw microscopy images acquired with the DeltaVision Ultra High Resolution microscope or DeltaVision OMX Flex microscope were deconvolved using the softWoRX deconvolution plugin (version 7.2.1 and version 7.2.0, respectively). Raw microscopy images acquired with the Nikon Eclipse Ti2 microscope were deconvolved using the NIS Elements Batch Deconvolution v5.20.00. Chromatic shift correction was done with Huygens compute engine 23.04.0p0.<br>Flow cytometry: CytExpert 2.3 analysis software |
| Data analysis | CryoEM: Huygens Essential version 23.04.0p0, MAPS Software (Thermo Scientific), AreTomo version 1.3.3; cryoCARE; MemBrain-Seg; Amira (Thermo Scientific, version 2022.2), https://github.com/juglab/cryoCARE_pip, https://github.com/teamtomo/membrain-seg<br>Fluorescence microscopy: FIJI, SoftWoRX (version 7.2.1 and version 7.2.0, respectively), NIS Elements Batch Deconvolution v5.20.00<br><br>Flow cytometry: FlowJo (FlowJo V.10.9.0 - May 5, 2023), in-house developed Python library mKeima (version 0.5.0, available at https://pypi.org/project/mkeima); Imaris Image Analysis Software Version 10.2 (Oxford Instruments Andor), https://github.com/hollenstein/sourcecode_mkeima-assay_licheva-et-al-2024<br><br>Mass spec: HTRMS converter (version 18.3, Biognosys), Spectronaut (version 18.5, Biognosys), the in-house developed Python library |

MsReport (version 0.0.23), ModeNormalizer from MsReport; Linear Models for Microarray Analysis (LIMMA, version 3.54.2) package in R.

Mathematical modeling: cellular_raza (https://github.com/jonaspleyer/cellular_raza)

FRAP:https://github.com/CraignRush/FRAP-Processing

For manuscripts utilizing custom algorithms or software that are central to the research but not yet described in published literature, software must be made available to editors and reviewers. We strongly encourage code deposition in a community repository (e.g. GitHub). See the Nature Portfolio guidelines for submitting code & software for further information.

## Data

Policy information about availability of data

All manuscripts must include a data availability statement. This statement should provide the following information, where applicable:

- Accession codes, unique identifiers, or web links for publicly available datasets
- A description of any restrictions on data availability
- For clinical datasets or third party data, please ensure that the statement adheres to our policy

We believe in making data publicly available whenever possible and do so for most of our data. The microscopy data is derived from hypothesis-driven experiments, unlike, for example, screening data, and mining it will be of limited use. However, as these images take up a lot of storage space and are cumbersome to deposit, we feel it's better to make the data available on request to those who need it.

No restrictions apply to the data collected for this manuscript. The mass spectrometry proteomics data have been deposited to the ProteomeXchange Consortium via the PRIDE 55 partner repository with the data set identifier PXD047277. Source data have been provided in Source Data. All other data supporting the findings of this study are available from the corresponding author on reasonable request.
The Python source data are available from the GitHub repository at https://github.com/hollenstein/sourcecode_mkeima-assay_licheva-et-al-2024
Datasets used: PDB: 3OGO, gene reference proteome from Uniprot (Proteome ID: UP000005640, release 2023.03), concatenated with a database of 379 common laboratory contaminants (in-house database), Gene Ontology (GO) term "ER membrane" (GO:0005789)

## Research involving human participants, their data, or biological material

Policy information about studies with human participants or human data. See also policy information about sex, gender (identity/presentation), and sexual orientation and race, ethnicity and racism.

| Reporting on sex and gender | n.a. |
|---|---|
| Reporting on race, ethnicity, or other socially relevant groupings | n.a. |
| Population characteristics | n.a. |
| Recruitment | n.a. |
| Ethics oversight | n.a. |

Note that full information on the approval of the study protocol must also be provided in the manuscript.

# Field-specific reporting

Please select the one below that is the best fit for your research. If you are not sure, read the appropriate sections before making your selection.

☒ Life sciences    ☐ Behavioural & social sciences    ☐ Ecological, evolutionary & environmental sciences

For a reference copy of the document with all sections, see nature.com/documents/nr-reporting-summary-flat.pdf

# Life sciences study design

All studies must disclose on these points even when the disclosure is negative.

| Sample size | No statistical methods were used to predetermine sample size. A sufficient sample size was determined based on variance between experiments. |
|---|---|
| Data exclusions | No data was excluded from analysis, except for clear technical failure. |
| Replication | At least three successful independent biological replicates were performed for each experiment, as indicated in the material and method section, except two successful biological replicates for Figure S7b, and three technical replicates for Fig. 1e,f and ED Fig. 3a,f,g, 4c,d. |
| Randomization | For all fluorescence microscopy image acquisition, cells were selected at random from the brightfield channel without bias towards the fluorescence signal. None microscopy samples (e.g. immunoblots) were analyzed based on genotypes, treatments, and/or time points with |

| | |
|---|---|
| | internal controls without randomization. |
| Blinding | For some experiments, blinding of the manual data analysis was not performed as all cells in the images were analyzed and field of views were selected from the brightfield without prior knowledge of the fluorescence signal (Fig. 1d, 2f; ExtData 3e, 5b). <br><br> For all other fluorescence microscopy experiments cells were selected at random from the brightfield channel without bias towards the fluorescence signal. Analysis of fluorescence images was either performed computationally (no blinding necessary) (Fig. 1b, 1c, 3a, 3g; ExtData 1c, 1g, 1h) or when manual performed quantification of fluorescence microscopy images was performed blindly after randomizing file names (all others). |

# Reporting for specific materials, systems and methods

We require information from authors about some types of materials, experimental systems and methods used in many studies. Here, indicate whether each material, system or method listed is relevant to your study. If you are not sure if a list item applies to your research, read the appropriate section before selecting a response.

### Materials & experimental systems

| n/a | Involved in the study |
|---|---|
| ☐ | ☒ Antibodies |
| ☐ | ☒ Eukaryotic cell lines |
| ☒ | ☐ Palaeontology and archaeology |
| ☒ | ☐ Animals and other organisms |
| ☒ | ☐ Clinical data |
| ☒ | ☐ Dual use research of concern |
| ☒ | ☐ Plants |

### Methods

| n/a | Involved in the study |
|---|---|
| ☒ | ☐ ChIP-seq |
| ☐ | ☒ Flow cytometry |
| ☒ | ☐ MRI-based neuroimaging |

## Antibodies

| | |
|---|---|
| Antibodies used | mouse monoclonal anti-GFP (1:100; 2B6, Monoclonal Antibody Facility, Max Perutz Labs, Vienna) <br> mouse monoclonal anti-GFP (1:5,000; 7.1 and 13.1, Ref No. 11814460001, Lot No. 70378300, Roche) <br> IRDye 800CW Goat anti-Mouse (1:1,000, Ref No. 926-32210, Lot No. D10825-15, Licor), <br> anti-Pgk1 (1:10,000, 22C5D8, Ref No. 459250 Lot NoVC2958788, Invitrogen) <br><br> rabbit polyclonal anti-Ape1 (1:15,000), was generated by immunizing rabbits with a synthetic peptide corresponding to amino acids 168-182. ref 43. <br><br> rabbit polyclonal anti-Atg19 (1:5,000, Sascha Martens, Monoclonal Antibody Facility, Max Perutz Labs, Vienna) ref 9 <br><br> mouse monoclonal anti-GST (1:1,000, 2H3-D10, Monoclonal Antibody Facility, Max Perutz Labs, Vienna) <br><br> rabbit polyclonal anti-ATG13, (1:50, 5HY-C1-F8, Monoclonal Antibody Facility, Max Perutz Labs, Vienna) ref 44 <br><br> mouse monoclonal anti-RFP (1:1,000, 6g6, Ref No. 6g6-100 Lot No51020014AB-05, Chromotek). validated by the company using transient expression of mRFP, mCherry, mPlum, mOrange, mRFPruby, DsRed, mScarlet and tdTomato on HEK 293T cells by western blot. Ref: Barucci G et al., Nat Cell Biol. 2020, doi:10.1038/s41556-020-0462-7. |
| Validation | mouse monoclonal anti-GFP (1:100; 2B6, Monoclonal Antibody Facility, Max Perutz Labs, Vienna) validated by the facility by western blot (GFP transfected vs. untransfected cells), used in Kiermaier et al., Science 2016, doi: 10.1126/science.aad0512 <br><br> mouse monoclonal anti-GFP (1:5,000; 7.1 and 13.1, Ref No. 11814460001, Lot No. 70378300, Roche), The mixture of two monoclonal antibodies, clones 7.1 and 13.1 was validated and tested by the company using western blot and immunoprecipitation of GFP fusion proteins. Ref: Wong et al., Blood. 2011, doi:10.1182/blood-2011-06-353938. <br><br> IRDye 800CW Goat anti-Mouse (1:1,000, Ref No. 926-32210, Lot No. D10825-15, Licor), Validated and tested by the company using dot blot and solid phase absorbed for minimal cross-reactivity with human, rabbit, goat, rat, and horse serum proteins. Ref: Wallroth et al., Nat Cell Biol 2019, doi:10.1038/s41556-019-0377-3. <br><br> anti-Pgk1 (1:10,000, 22C5D8, Ref No. 459250 Lot NoVC2958788, Invitrogen), validated by the company using Saccharomyces cerevisiae cell lysate and western blot. Ref: Montellà-Manuel et al., Int J Mol Sci. 2023, doi:10.3390/ijms24032438. <br><br> rabbit polyclonal anti-Ape1 (1:15,000), described and validated for yeast by western blot in ref 43 <br><br> rabbit polyclonal anti-Atg19 (1:5,000, Sascha Martens, Monoclonal Antibody Facility, Max Perutz Labs, Vienna) validated for yeast by western blot in ref 9 <br><br> mouse monoclonal anti-GST (1:1,000, 2H3-D10, Monoclonal Antibody Facility, Max Perutz Labs, Vienna), validated by western blot in Eisenhardt et al., Methods Enzymol, doi:10.1016/bs.mie.2018.12.025, |

rabbit polyclonal anti-ATG13, (1:50, 5HY-C1-F8, Monoclonal Antibody Facility, Max Perutz Labs, Vienna) validated for U2OS and HEK293 cells by western blot, described and validated in 44

mouse monoclonal anti-RFP (1:1,000, 6g6, Ref No. 6g6-100 Lot No51020014AB-05, Chromotek). validated by the company using transient expression of mRFP, mCherry, mPlum, mOrange, mRFPruby, DsRed, mScarlet and tdTomato on HEK 293T cells by western blot. Ref: Barucci G et al., Nat Cell Biol. 2020, doi:10.1038/s41556-020-0462-7.

# Eukaryotic cell lines

Policy information about cell lines and Sex and Gender in Research

| | |
|---|---|
| Cell line source(s) | HEK293 (R78007, ThermoFisher Scientific), U2OS (K650001, Thermo Fisher Scientific), spodoptera frugiperda Sf9 (94-001F, Biotrend) |
| Authentication | No authentication has been performed. |
| Mycoplasma contamination | All of the cell lines used were regularly checked for mycoplasma contamination and were always negative. |
| Commonly misidentified lines (See ICLAC register) | No commonly misidentified cell lines were used. |

# Plants

| | |
|---|---|
| Seed stocks | not used in the study |
| Novel plant genotypes | not used in the study |
| Authentication | not used in the study |

# Flow Cytometry

## Plots

Confirm that:

☒ The axis labels state the marker and fluorochrome used (e.g. CD4-FITC).

☒ The axis scales are clearly visible. Include numbers along axes only for bottom left plot of group (a 'group' is an analysis of identical markers).

☒ All plots are contour plots with outliers or pseudocolor plots.

☒ A numerical value for number of cells or percentage (with statistics) is provided.

## Methodology

| | |
|---|---|
| Sample preparation | U2OS cells (cCE377) were seeded in 6-well plates at a density of 500,000 cells per well and cultured in DMEM media supplemented with 10% FBS. After 24 h, the media in the plates were replaced with one containing 1 µg/ml of doxycycline to induce the expression of 2xFKBP-GFP-ULK1. Additionally, in the specified samples, 0.5 µM of rapalog was added for a 24 h incubation period. Before harvesting, the indicated samples were treated with 200 nM Bafilomycin A1 (Cell Signaling Technology, 54645) and rapalog. Following the treatment, cells were washed with DPBS and detached with trypsin-EDTA (Sigma, T3924). The cells were then harvested in FACS medium (phenol-red free DMEM, Sigma, D1145-500ML, supplemented with 10% FBS). Cells were transferred to 1.5 ml tubes and centrifuged at 500 x g for 3 min at room temperature. The supernatant was removed, and the cell pellets were resuspended in 200 µl of FACS media and transferred to U-bottom 96-well plates (Greiner Bio-One, 650970). |
| Instrument | CytExpert 2.3 analysis software, FlowJo V.10.9.0 - May 5, 2023 |
| Software | CytExpert 2.3 analysis software, FlowJo V.10.9.0 - May 5, 2023 |
| Cell population abundance | at least 50,000 events were analysed out of the initial at least 200,000 events, which accounts for 25% (alive, singlets, and GFP positive) |
| Gating strategy | The gating strategy is described in detail under https://github.com/hollenstein/sourcecode_mkeima-assay_licheva-et-al-2024 |

☒ Tick this box to confirm that a figure exemplifying the gating strategy is provided in the Supplementary Information.

