## [Peer Review File · Nature Cell Biology]

Phase separation of initiation hubs on cargo is a trigger switch for selective autophagy

Corresponding Author: Professor Claudine Kraft

Version 0:

Decision Letter:

*Please delete the link to your author homepage if you wish to forward this email to co-authors.

Dear Professor Kraft,

Thank you again for submitting your manuscript, "Phase separation of initiation hubs on cargo is a trigger switch for selective autophagy", to Nature Cell Biology. It has now been seen by 3 referees, who are experts in phase separation, autophagy (Referee #1); selective autophagy (Referee #2); and autophagy, yeast (Referee #3). As you will see from their comments (attached below), they found the work of potential interest but have raised substantial concerns, which in our view would need to be addressed with considerable revisions before we can consider publication in Nature Cell Biology.

As per our standard editorial process, Nature Cell Biology editors discuss the referee reports in detail within the editorial team, including the chief editor, to identify key referee points that should be addressed with priority, and requests that are overruled as being beyond the scope of the current study. To guide the scope of the revisions, I have listed these points below. Our standard revision period is six months, and we are committed to providing a fair and constructive peer-review process, so please feel free to contact me if you would like to discuss any of the referee comments further or if you anticipate any issues or delays addressing the reviews.

I should stress that the referees' concerns are significant and would need to be addressed thoroughly experimentally, and reconsideration of the study for this journal and re-engagement of referees will depend on the strength of these revisions. In our view, it would be essential to dedicate efforts in revision to address the following points:

1- Analyses of the initiation hubs were not yet convincing to the reviewers, and they requested a stronger test of the model of multivalent low-affinity interactions between autophagy receptors and cargo:

Rev#1 points #1, #2, #6, #8, #9, #10, #11

Rev#2 point #1

Rev#3 points #1, #2, #3, #4, #5, #6, #7, #10, #11, #12, #15

2- The reviewers were also critical of the analyses of Atg11 clusters and of the potential for phase separation of Atg11 to be occurring in cells with a functional role for hub formation:

Rev#1 points #3, #4, #5, #7

Rev#2 point #2

Rev#3 points #9, #13

3- Rev#2 (point #3) and Rev#3 (point #16) were not convinced by the analyses in mammalian cells, which need to be expanded to provide compelling evidence testing conservation.

4- All other referee concerns pertaining to strengthening existing data, providing controls, methodological details, clarifications and textual changes, should be addressed.

5- Finally, please pay close attention to our guidelines on statistical and methodological reporting (listed below) as failure to do so may delay the reconsideration of the revised manuscript. In particular, please provide:

- a Supplementary Table including all numerical source data in Excel format, with data for different figures provided as

different sheets within a single Excel file. The file should include source data giving rise to graphical representations and statistical descriptions in the paper and for all instances where the figures present representative experiments of multiple independent repeats, the source data of all repeats should be provided.

We would be happy to consider a revised manuscript that would satisfactorily address these points, unless a similar paper is published elsewhere or is accepted for publication in Nature Cell Biology in the meantime.

- ensure that it conforms to our format instructions and publication policies (see below and <https://www.nature.com/nature/for-authors>).
- provide a point-by-point rebuttal to the full referee reports verbatim, as provided at the end of this letter.
- provide the completed Reporting Summary (found here <https://www.nature.com/documents/nr-reporting-summary.pdf>). This is essential for reconsideration of the manuscript will be available to editors and referees in the event of peer review. For more information see <http://www.nature.com/authors/policies/availability.html> or contact me.

Nature Cell Biology is committed to improving transparency in authorship. As part of our efforts in this direction, we are now requesting that all authors identified as 'corresponding author' on published papers create and link their Open Researcher and Contributor Identifier (ORCID) with their account on the Manuscript Tracking System (MTS), prior to acceptance. ORCID helps the scientific community achieve unambiguous attribution of all scholarly contributions. You can create and link your ORCID from the home page of the MTS by clicking on 'Modify my Springer Nature account'. For more information please visit www.springernature.com/orcid.

This journal strongly supports public availability of data. Please place the data used in your paper into a public data repository, or alternatively, present the data as Supplementary Information. If data can only be shared on request, please explain why in your Data Availability Statement, and also in the correspondence with your editor. Please note that for some data types, deposition in a public repository is mandatory - more information on our data deposition policies and available repositories appears below.

Link Redacted

We hope that you will find our referees' comments and editorial guidance helpful. Please do not hesitate to contact me if there is anything you would like to discuss. Thank you again for considering our journal for your work.

Best wishes,

Melina

Melina Casadio, PhD
Senior Editor, Nature Cell Biology
ORCID ID: <https://orcid.org/0000-0003-2389-2243>

Reviewers' Comments:

Reviewer #1:

Remarks to the Author:

For starvation-induced autophagosome formation, core Atg proteins must gather to form the PAS on the vacuole, which is known to be mediated by liquid-liquid phase separation. However, it remained elusive how core Atg proteins behave during selective autophagy. In this manuscript, the authors studied Ape1 and END droplets as a model for selective autophagy cargo and showed that weak, but not strong interaction between the receptor and the cargo is important for selective autophagy. The authors then showed that Atg11 undergoes phase separation *in vitro* and in yeast cells and attaches to the cargo surface as puncta via strong interaction with the receptor on the cargo. Atg11 puncta coalesced to each other on the cargo to form "initiation hubs", for which the mobility of receptors on the cargo was shown to be important. Moreover, the authors showed that the concept of initiation hubs established here can be applicable to selective autophagy in mammals. The mechanism of selective autophagy initiation is one of the hottest topics in the field of autophagy and the proposed concept of initiation hubs is attractive. Moreover, the authors have constructed and studied a number of thoughtful experimental systems in order to strengthen their concept. On the other hand, the main conclusions are not sufficiently supported by the provided data as written below. It is essential to provide additional data that strengthen the main conclusions.

Major comments

- 1) The authors concluded that the affinity between Atg19 and Ape1 is low whereas that between Atg19 and Atg11 is high based on the GSH beads experiments. However, the used experiments are not quantitative and do not directly provide affinity information. To claim that Atg19-Atg11 interaction is stronger than Atg19-Ape1 interaction, determine each Kd value using ITC or SPR and compare the affinity quantitatively.
- 2) Figure 1F data are interesting. However, the complete block of the Cvt pathway by fusing GBP to Atg19 might be simply due to the loss of function of Atg19-GBP rather than the lowered mobility of the receptor. Perform control experiments expressing Atg19-GBP with mutation that impairs interaction with GFP and confirm that GFP-Ape1 is processed when expressing this Atg19-GBP mutant.
- 3) From the Kymograph data in Figure 3B, the authors wrote that morphological changes of the Atg11 clusters resemble fusion and fission of individual clusters. However, the Kymograph data may just show the movement of Atg11 clusters on the Ape1 complex. Provide time-lapse imaging data and show that Atg11 clusters actually fuse to each other.
- 4) The authors performed *in vitro* phase separation experiments using GFP-Atg11 and showed that GFP-Atg11 alone undergoes phase separation to form droplets. However, it is not known whether Atg11 alone undergoes phase separation to form the PAS in cells. In the case of starvation-induced autophagy, Atg13 is essential for phase separation to form the PAS in addition to an Atg11 paralog Atg17. Study the effect of Atg13 deletion on the Atg11 puncta formation in cells.
- 5) Based on Figure 3E data, the authors wrote that GFP-Atg11 droplets coalesced to each other. However, in Figure 3E, the two droplets seem not to be in the same z section and the left droplet appears to just pass over the right one from 0 to 70 seconds. Provide time-lapse imaging data and show that GFP-Atg11 droplets actually coalesced to each other.
- 6) Based on the data in Figure S2D, the authors wrote that Atg11-Atg19 interaction is high affinity. However, it was reported that Atg19 requires phosphorylation for high affinity interaction with Atg11 (PMID 24968893, 25287303). As written above, quantitative analysis is required in order to discuss the importance of affinity and so compare the Kd values between Atg11-Atg19, Atg11-phospho-Atg19, and Atg19-Ape1.
- 7) Based on Figure S2D data, the authors wrote that "GFP-Atg11 formed bright clusters on the Atg19-decorated beads, suggesting phase separation *in vitro*". However, the bright clusters of GFP-Atg11 remained on beads even after washing, suggesting that the clusters are aggregates of GFP-Atg11 rather than phase-separated droplets. To claim that GFP-Atg11 clusters on the Atg19-decorated beads are phase-separated ones, perform FRAP experiments on them and confirm the fluorescence recovery as observed in Figure 3D.
- 8) The authors developed a mathematical model and showed that low-affinity multivalent interactions between cargo and Atg11-Atg19 support the formation of initiation hubs (Figure 4A). This model assumes that the Atg11-Atg19 interaction is much stronger than that between Atg19 and Ape1. As written above, it is important to determine each Kd value to underline this mathematical model.
- 9) In Figure 4D, the authors provided only the cryo-EM image. Provide fluorescence image superimposed on the cryo-EM image in order to show that the phagophore is generated from the PAS on the vacuole.
- 10) In Figure 4E, the authors showed that co-expression of pp-GFP-uNS and Atg19 resulted in the autophagic degradation of pp-GFP-uNS whereas Atg19-GFP-uNS not. Moreover, co-expression of Cnb1-Atg19 and FKBP-GFP-uNS in the presence of FK506 did not result in the degradation of FKBP-GFP-uNS. In these experiments, there may be a possibility that only wild-type Atg19 is functional and that the affinity between Atg19 and cargo does not matter. It is necessary to show that Atg19-GFP-uNS or Cnb1-Atg19 is functional as a receptor, for example by studying the Ape1 delivery to the vacuole in *atg19delta* cells expressing these fusion proteins.
- 11) The authors concluded that "the mobility of receptors on selective cargo is essential for cargo degradability by selective autophagy, rather than the cargo property itself" in lines 311-312. However, the authors did not provide the data showing that the cargo property is not important for cargo degradation by selective autophagy. Cargo property should affect the mobility of receptors on the cargo and so this conclusion seems not accurate. If the authors do not provide the data about the effect of changing the cargo property on its selective autophagy efficiency, then reconsider the conclusion.

Minor comments

- 1) Provide images for Figure 2F as Supplementary.

Reviewer #2:

Remarks to the Author:

Phase separation was shown to organize autophagosome formation sites and mediate autophagy cargo assembly in bulk autophagy and selective autophagy, respectively. In this manuscript, Licheva et al described that the scaffold protein Atg11 phase separate at the surface of cargos to trigger initiation hub formation in yeast. The phase separation of Atg11 is mediated by low affinity interactions between cargo and cargo receptors, and this low affinity interactions were shown to be required for phagophore initiation and subsequent cargo degradation. This study reveals that receptor mobility rather than the cargo allows Atg11 condensation and initiation hub formation, and followed autophagic degradation. Overall, this discovery will be of general interest to the field. However, the conclusions are not entirely convincing, more evidences need to be provided to strengthen the key findings.

Major concerns:

1) One key finding of this study is that low-affinity cargo-receptor interactions promote receptor mobility, which enables initial hub formation during selective autophagy in yeast. However, first, the author manipulated the interactions between cargo and receptors solely by artificially inducing an exogenous GFP binding protein into cells. Mutants on both cargo and receptor side that change the interaction affinity should be generated, and the effects of such mutants on cargo degradation and initial hub formation should be tested. Second, the authors only showed the dynamics of Ede1 without checking mobility of other receptors like Atg19 in different conditions. The cargo mobility also needs to be shown, it could be possible that interaction affinity also changes the cargo properties. Third, the authors used copper-inducible promoter in several experiments, but previous study shows that copper is important for ULK1 activity regulation (Tsang et al, 2020), which could also affect Atg1 in yeast, the authors should check the effect of copper in their experiments.

2) Another finding is phase separation of Atg11 drives initial hub formation. The current evidences are not sufficient to demonstrate that Atg11 is the driving force, the authors need to show that Atg11 condensates are capable to recruit other core autophagy proteins like Atg1, Atg9 in vitro. And mutants of Atg11 that cannot phase separate into condensates should be generated, and the initial hub formation and cargo degradation in cells expressing such mutants should be analyzed.

3) To demonstrate that initiation hubs for selective autophagy are conserved in human cells, the authors only showed ULK1 foci on ER surface upon mitophagy induction. Initiation hub formation around more mammalian cargos should be examined. Moreover, it is important to test whether the formation of initiation hubs in mammalian cells is also mediated by low-affinity interactions between cargo and cargo receptors, whether the principle that receptor mobility regulate selective autophagy initiation is conserved from yeast to human cells need to be confirmed.

Minor points:

1) Except for dynamics of Ede1, whether other properties of Ede1 condensates like number or size changed in the +3xGBP cells. It looks that the condensates were bigger in the +3xGBP cells in Fig S1A.

2) In Fig 1F, the authors need to show that the interaction of GFP-Ape1 peptide and Atg19-GBP is high affinity using a similar bead binding assay as they showed in Fig 1E.

3) In Fig 2 A i) and iii), is there a particular reason to use atg19 Δ cells? would Atg11 form cluster around Ede1 condensates in wild type cells. Why did Atg11 form several clusters around Ape1 condensate, but only a single or even none around Ede1 condensates?

4) Fig 2B was not correctly described in the main text.

5) In Fig 2E and 3A, the author showed that Atg11 clusters on Ape1 condensates are highly dynamic in normal cells, how about the dynamics of Atg11 in Atg9-GBP expressing cells? Whether Atg1 or Atg9 colocalize with the less-clustered Atg11 in Atg9-GBP expressing cells?

6) The authors hypothesized that all the Atg11 clusters coalesce into a single initiation hub, but only showed that Atg8 and Atg9 formed a single focus upon rapamycin treatment in Fig 4, the localization of Atg11 upon rapamycin treatment need to be shown simultaneously.

Reviewer #3:

Remarks to the Author:

Cellular components to be degraded by autophagy (cargos) are sequestered within autophagosomes and delivered to lysosomes or vacuoles for degradation. Recent studies have revealed the involvement of phase separation in different events during autophagy, including that of cargo molecules and components of the autophagosome formation machinery (Atg proteins). In this study, Licheva et al. investigate the relationship between physical properties of cargos and receptors and the initiation of autophagosome formation via phase separation of related proteins. Based on the results, the authors provide several insights into the molecular mechanism of the initiation of autophagosome formation in selective autophagy, but most of the conclusions are not convincingly supported by the experimental data as described below in detail and therefore this work is too preliminary for publication.

Specific comments:

1. The authors showed that linking GFP-Ede1 molecules with BFP-3xGFP or linking Atg19 to Ape1 via the GBP-GFP interaction abolished vacuolar transport of Ede1 or Ape1, respectively, but it has not been shown that autophagosome formation is indeed hampered at the initiation step (before membrane expansion) as the authors assume.

2. Lines 120-122, "Despite their solidification, END assemblies still colocalized with the autophagy protein Atg8, suggesting that the autophagy receptor properties of Ede1 were not altered (Figure 1D and S1C).": To draw this conclusion, the authors should show that Atg8 localization to the END assemblies depends on the Ede1 AIM.

3. Lines 122-124, "We infer that high-affinity interactions between 2xGFP-Ede1 and BFP-3xGBP render the receptor Ede1 immobile and that receptor mobility is required for selective END degradation.": It is unclear whether BFP-3xGFP indeed inhibited Ede1 degradation by decreasing its mobility. For instance, BFP-3xGFP incorporation may dilute 2xGFP-Ede1 in END assemblies and thereby decrease the initiation of autophagosome formation.

4. Fig. 1E: The authors should examine the GFP-GBP interaction in this assay to compare it with the Ape1 1-45-Atg19 interaction.

5. Fig. 1F: Since GBP tagging of Atg19 itself could affect the efficiency of GFP-Ape1 degradation, a GBP mutant defective in GFP binding is necessary as a control.

6. Line 156, "Because these clusters contain multiple factors that are involved in phagophore initiation, we termed them "initiation hubs".": These criteria are insufficient to define "initiation hubs" as distinguished from the PAS. The authors should set up experimental conditions for the observation of initiation hubs separately from the PAS, PAS scaffold, or phagophore and then analyze how different changes in related molecules affect their formation.

7. Lines 158-160, "We noticed that Atg11 clustering was reduced in cells with increased affinity of receptor-cargo interactions, such as the strains co-expressing Atg19-GBP with GFP-Ape1 (Figure 2E) and those co-expressing 2xGFP-Ede1 with BFP-3xGBP (Figure 2F).": The authors rely on the coefficient of variance of the fluorescence intensity to evaluate Atg11 clustering, but the actual image in Fig. 2E appears that enlargement of Atg11 clusters or an increase in Atg11 levels on the Ape1 surface resulted in an apparent decrease in Atg11 clustering. It is also required to observe the localization of Atg1 and Atg9 in these experiments.

8. Fig. 2E: The raw data (fluorescence microscope images) should be presented.

9. Lines 170-171, "These morphological changes resemble fusion and fission of individual clusters, consistent with phase separation in vivo.": This statement comes from the results shown in Fig. 3B but does not make sense.

10. Lines 181-183, "GFP-Atg11 was efficiently recruited to GST-BFP-Atg19 immobilized on GSH beads and remained bound after subsequent washes, consistent with a high-affinity interaction between
183 the scaffold and receptor (Figure S2D).": This assay without any controls for comparison does not provide reliable information on whether the affinity is high or low; the authors should directly compare the strength of the interactions in the same assay.

11. Lines 181-187, "Moreover, GFP-Atg11 formed bright clusters on the Atg19-decorated beads (Figure S2D), suggesting phase separation in vitro. Given that increasing the affinity of the Atg19 receptor for the Ape1 cargo reduced Atg11 cluster formation (Figure 2E), we propose that low-affinity Atg19 receptor-cargo interactions enable phase separation of Atg11-Atg19 complexes and initiation hub formation.": Does GST-BFP-Atg19 bind to GSH-beads with high affinity? If so, the results shown in Fig. S2D seem to contradict this proposal.

12. Did the authors compare the two cases, in which the one was given low-affinity and high avidity and the other was given only high-affinity in mathematical modeling analysis? If so, can they attribute Atg11 clustering in the former to the difference in the affinity?

13. Fig. S3: The conclusion that Vac8 colocalizes with Atg11 clusters is not convincing, since Vac8 does not appear to form discrete puncta on the vacuolar membrane in the fluorescence images. The kymograph was made by analyzing the fluorescence intensity along the Ape1 surface, and therefore the fluorescence of Vac8 and Atg11 appears to match but this would not be regarded as their colocalization.

14. Fig. 4D: The fluorescence image should also be shown.

15. Figs. 4E, 4F, S4D, S4E: In these experiments, the authors compared the proteins with quite different architectures and therefore it is difficult to attribute changes in degradation and Atg11 clustering to their difference in cargo-receptor affinity. It would be better to compare proteins with an essentially same structure but containing mutations that are expected to affect the affinity and after experimentally confirming the affinity use them for these experiments.

16. Initiation hubs should also be more carefully defined in mammalian cells similar to the case in yeast as described above.

Methods should be written concisely, but should contain all elements necessary to allow interpretation and replication of the results. As a guideline, Methods sections typically do not exceed 3,000 words. The Methods should be divided into subsections listing reagents and techniques. When citing previous methods, accurate references should be provided and any alterations should be noted. Information must be provided about: antibody dilutions, company names, catalogue numbers and clone numbers for monoclonal antibodies; sequences of RNAi and cDNA probes/primers or company names and catalogue numbers if reagents are commercial; cell line names, sources and information on cell line identity and authentication. Animal studies and experiments involving human subjects must be reported in detail, identifying the committees approving the protocols. For studies involving human subjects/samples, a statement must be included confirming that informed consent was obtained. Statistical analyses and information on the reproducibility of experimental results should be provided in a section titled "Statistics and Reproducibility".

All Nature Cell Biology manuscripts submitted on or after March 21 2016 must include a Data availability statement as a separate section after Methods but before references, under the heading "Data Availability". For Springer Nature policies on data availability see <http://www.nature.com/authors/policies/availability.html>; for more information on this particular policy

see <http://www.nature.com/authors/policies/data/data-availability-statements-data-citations.pdf>. The Data availability statement should include:

- Accession codes for primary datasets (generated during the study under consideration and designated as "primary accessions") and secondary datasets (published datasets reanalysed during the study under consideration, designated as "referenced accessions"). For primary accessions data should be made public to coincide with publication of the manuscript. A list of data types for which submission to community-endorsed public repositories is mandated (including sequence, structure, microarray, deep sequencing data) can be found here <http://www.nature.com/authors/policies/availability.html#data>.
- Unique identifiers (accession codes, DOIs or other unique persistent identifier) and hyperlinks for datasets deposited in an approved repository, but for which data deposition is not mandated (see here for details <http://www.nature.com/sdata/data-policies/repositories>).
- At a minimum, please include a statement confirming that all relevant data are available from the authors, and/or are included with the manuscript (e.g. as source data or supplementary information), listing which data are included (e.g. by figure panels and data types) and mentioning any restrictions on availability.
- If a dataset has a Digital Object Identifier (DOI) as its unique identifier, we strongly encourage including this in the Reference list and citing the dataset in the Methods.

We recommend that you upload the step-by-step protocols used in this manuscript to the Protocol Exchange. More details can be found at www.nature.com/protocolexchange/about.

All imaging data should be accompanied by scale bars, which should be defined in the legend.

Cropped images of gels/blots are acceptable, but need to be accompanied by size markers, and to retain visible background signal within the linear range (i.e. should not be saturated). The boundaries of panels with low background have to be demarcated with black lines. Splicing of panels should only be considered if unavoidable, and must be clearly marked on the figure, and noted in the legend with a statement on whether the samples were obtained and processed simultaneously. Quantitative comparisons between samples on different gels/blots are discouraged; if this is unavoidable, it should only be performed for samples derived from the same experiment with gels/blots were processed in parallel, which needs to be stated in the legend.

Regardless of format, all figures must be vector graphic compatible files, not supplied in a flattened raster/bitmap graphics format, but should be fully editable, allowing us to highlight/copy/paste all text and move individual parts of the figures (i.e.

arrows, lines, x and y axes, graphs, tick marks, scale bars etc.). The only parts of the figure that should be in pixel raster/bitmap format are photographic images or 3D rendered graphics/complex technical illustrations.

The total number of Supplementary Figures (not including the “unprocessed scans” Supplementary Figure) should not exceed the number of main display items (figures and/or tables (see our Guide to Authors and March 2012 editorial <http://www.nature.com/nature/authors/submit/index.html#suppinfo>; <http://www.nature.com/nature/journal/v14/n3/index.html#ed>). No restrictions apply to Supplementary Tables or Videos, but we advise authors to be selective in including supplemental data.

GUIDELINES FOR EXPERIMENTAL AND STATISTICAL REPORTING

REPORTING REQUIREMENTS – We are trying to improve the quality of methods and statistics reporting in our papers. To that end, we are now asking authors to complete a reporting summary that collects information on experimental design and reagents. The Reporting Summary can be found here <https://www.nature.com/documents/nr-reporting-summary.pdf>. If you would like to reference the guidance text as you complete the template, please access these flattened versions at <http://www.nature.com/authors/policies/availability.html>.

Version 1:

Decision Letter:

*Please delete the link to your author homepage if you wish to forward this email to co-authors.

Dear Professor Kraft,

Thank you for submitting your revised manuscript, "Phase separation of initiation hubs on cargo is a trigger switch for selective autophagy", to the journal, which has been assessed by the original referees. As you will see from their comments (attached below), they continue to find the work interesting and appreciated the efforts made in revision. However, each had remaining, persisting concerns regarding how their previous comments were addressed, which we find significant at this stage. These concerns would need to be addressed before we can consider publication in Nature Cell Biology.

We strive to limit all our manuscripts to a single round of major experimental revision in order to limit the overall time spent in peer review. However, given interest in the study and overall support from the reviewers, we are open to a final round of revision if you can address the remaining reviewer comments, as follows:

-- You will see that Rev#1 is not convinced by the analyses provided to address their previous request for the Kd value of each interaction (point #1 in the first round of review). The reviewer suggests a published approach to resolve the purification issues. We continue to believe that this point should be addressed.

-- Rev#2 is not convinced by the analyses presented to address their previous point #3 and we agree that further tests as suggested by the reviewer would help support claims of conservation in human cells.

-- Rev#3 requested clarifications; in particular, the imaging data of fusion or splitting events for the clusters needs to be clearer and convincing to experts in the field.

-- Finally, please pay close attention to our guidelines on statistical and methodological reporting (listed below) as failure to do so may delay the reconsideration of the revised manuscript. In particular, please provide:

We would be happy to consider a revised manuscript that would satisfactorily address these points, unless a similar paper is published elsewhere, or is accepted for publication in Nature Cell Biology in the meantime.

Please do not hesitate to contact me if you would like to discuss the reviews or a revision plan, or if you have any questions. We hope these revisions could be finalized within 2 to 6 months, but please let us know if this is a concern.

- ensure that it conforms to our format instructions and publication policies (see below and <https://www.nature.com/nature/for-authors>).

- provide a point-by-point rebuttal to the full referee reports verbatim, as provided at the end of this letter.

- provide the completed Reporting Summary (found here <https://www.nature.com/documents/nr-reporting-summary.pdf>). This is essential for reconsideration of the manuscript will be available to editors and referees in the event of peer review. For more information see <http://www.nature.com/authors/policies/availability.html>

or contact me.

Nature Cell Biology is committed to improving transparency in authorship. As part of our efforts in this direction, we are now requesting that all authors identified as 'corresponding author' on published papers create and link their Open Researcher and Contributor Identifier (ORCID) with their account on the Manuscript Tracking System (MTS), prior to acceptance. ORCID helps the scientific community achieve unambiguous attribution of all scholarly contributions. You can create and link your ORCID from the home page of the MTS by clicking on 'Modify my Springer Nature account'. For more information please visit www.springernature.com/orcid.

This journal strongly supports public availability of data. Please place the data used in your paper into a public data repository, or alternatively, present the data as Supplementary Information. If data can only be shared on request, please explain why in your Data Availability Statement, and also in the correspondence with your editor. Please note that for some data types, deposition in a public repository is mandatory - more information on our data deposition policies and available repositories appears below.

Link Redacted

We hope that you will find our referees' comments and editorial guidance helpful. Please do not hesitate to contact me if there is anything you would like to discuss. Thank you again for your efforts in revision, and for considering the journal for your work.

Best wishes,

Melina

Melina Casadio, PhD
Senior Editor, Nature Cell Biology
ORCID ID: <https://orcid.org/0000-0003-2389-2243>

Reviewers' Comments:

Reviewer #1:

Remarks to the Author:

The authors have addressed many of my concerns. However, one major concern is left unresolved. Due to the technical difficulties, the authors failed to determine the Kd values between Atg19 and Ape1 and between Atg19 and Atg11 and instead they compared the Atg19-Ape1 and Atg19-Atg11 interactions with GFP-GBP (Kd=1 nM) and GFP-GBP mutant interactions (Kd is presumably larger than 1.4 μM; the actual affinity should also be determined) using the GSH beads experiments. However, this reviewer has a strong concern about the GSH beads experiments. In the rebuttal letter, the authors wrote that "GFP/GBP binding in wash-off experiments showed a similar behaviour as Atg19/Atg11, whereas GFP/GBP-F103 E104 (GBP-low) showed only very little binding, similar to the Atg19/PP interaction". If this is correct, then the Atg19-Atg11 interaction is an extremely strong interaction of Kd=1 nM, and the Atg19-Ape1 interaction is weaker than Kd=1.4 μM. However, a previous study purified the coiled-coil of Atg19 complexed with the propeptide (residues 1-20) of Ape1 using size-exclusion chromatography (SEC) and determined the crystal structure of the complex at high resolution (PMID 27320913). Stable complex formation during SEC suggests that the affinity is sufficiently high and the Kd value should not be higher than 1.4 μM. The affinity between Atg19 and Atg11 has not been reported, but in mammals the affinity between autophagic receptors-FIP200 (Atg11 homolog) has been extensively studied, which showed the Kd values in the order of 1~100 μM (even the phosphorylated (affinity-enhanced) CCPG1 FIR and Optineurin LIR gave the Kd value of 0.7 μM and 11 μM, respectively, to FIP200 (PMID 33692357)). Therefore, it is not reliable to use the GSH beads experiment as an

estimate of affinity. Because the purification method of Atg19 coiled-coil and Ape1 propeptide (1-20) has already been reported (PMID 27320913), it should be not difficult to measure their affinity by SPR or ITC or FP-based assays. The affinity between Atg19 and Atg11 must also be determined. Because the conclusion that the affinity between Atg19 and Atg11 is much higher than that between Atg19 and Ape1 is a major point in this manuscript, this conclusion must be supported by solid data. To make the matters worse, if this conclusion is different, several other experiments will become invalid since they are based on this (e.g., molecular simulation).

Reviewer #2:

Remarks to the Author:

In the revised manuscript, the authors provided additional data to support the main conclusion that the low-affinity cargo-receptor interaction is essential for initial hub formation and cargo degradation. Most of the reviewer's comments have been addressed, however, the evidences that support the importance of low cargo receptor mobility and initial hub formation is conserved in mammals are still not strong enough. To confirm the formation of initial hub in mammalian cells, they showed p62 condensates could recruit FIP200, which is kind of expected, as a direct interaction between p62 and FIP200 was reported previously. They stated that FIP200 could connect p62 condensates to ER, but the data are not robust and quantitative. To confirm the low mobility is required for initiation hub formation in mammalian cells, the authors manipulated the mobility of FRB-Fis1, but it would be better to use cargo receptor mutants that changed the mobility to do the experiments. Besides, only a working model in yeast was showed in Fig8e, a universal model of selective autophagy should include both yeast and mammals.

Reviewer #3:

Remarks to the Author:

The authors have addressed most of the concerns I raised in the review of the original manuscript, but I request the authors further clarify the following points:

>> 9. Lines 170-171, "These morphological changes resemble fusion and fission of individual clusters, consistent with phase separation in vivo."; This statement comes from the results shown in Fig. 3B but does not make sense.

> To address this point in more detail, we provide stills and time-lapse movies, in which one can clearly observe how certain clusters fuse and split again. These findings are shown as Supplementary Videos 1 and 2 and in Extended Data Fig. 2d, and support our findings shown by the kymograph analysis.

It is not clear at all what parts in the still images and movies represent fusion and split of Atg11 clusters. It appears that clusters are just moving around.

>> 11. Lines 181-187, "Moreover, GFP-Atg11 formed bright clusters on the Atg19-decorated beads (Figure S2D), suggesting phase separation in vitro. Given that increasing the affinity of the Atg19 receptor for the Ape1 cargo reduced Atg11 cluster formation (Figure 2E), we propose that low-affinity Atg19 receptor-cargo interactions enable phase separation of Atg11-Atg19 complexes and initiation hub formation.": Does GST-BFP-Atg19 bind to GSH-beads with high affinity? If so, the results shown in Fig. S2D seem to contradict this proposal.

> GST-tagged proteins bind with high affinity to GSH beads, which makes this setup widely used for affinity purifications. However, in this assay, we add recombinant GFP-Atg11 at the critical concentration for phase separation, which contains a mixture of soluble and condensated Atg11. This mixture is subsequently incubated with the Atg19 decorated GSH beads, and both soluble Atg11 and the condensates stably bind to Atg19 decorated beads. The purpose of this experiment is to compare the affinity of the Atg11/Atg19 interaction to the PP/Atg19 interaction.

I would like to confirm with the authors that they removed the descriptions like "GFP-Atg11 formed bright clusters on the Atg19-decorated beads, suggesting phase separation in vitro" in the revised manuscript, which did not correctly describe the experimental settings and was therefore misleading. In addition, the authors should clearly describe in the main text that they used recombinant Atg11 highly concentrated to form Atg11 clusters in the absence of Atg19 and observed binding of preformed Atg11 clusters to Atg19-decorated beads, to avoid readers' misunderstanding that Atg11 is phase-separated on the beads.

READABILITY OF MANUSCRIPTS – Nature Cell Biology is read by cell biologists from diverse backgrounds, many of

whom are not native English speakers. Authors should aim to communicate their findings clearly, explaining technical jargon that might be unfamiliar to non-specialists, and avoiding non-standard abbreviations. Titles and abstracts should concisely communicate the main findings of the study, and the background, rationale, results and conclusions should be clearly explained in the manuscript in a manner accessible to a broad cell biology audience. Nature Cell Biology uses British spelling.

Methods should be written concisely, but should contain all elements necessary to allow interpretation and replication of the results. As a guideline, Methods sections typically do not exceed 3,000 words. The Methods should be divided into subsections listing reagents and techniques. When citing previous methods, accurate references should be provided and any alterations should be noted. Information must be provided about: antibody dilutions, company names, catalogue numbers and clone numbers for monoclonal antibodies; sequences of RNAi and cDNA probes/primers or company names and catalogue numbers if reagents are commercial; cell line names, sources and information on cell line identity and authentication. Animal studies and experiments involving human subjects must be reported in detail, identifying the committees approving the protocols. For studies involving human subjects/samples, a statement must be included confirming that informed consent was obtained. Statistical analyses and information on the reproducibility of experimental results should be provided in a section titled "Statistics and Reproducibility".

All Nature Cell Biology manuscripts submitted on or after March 21 2016 must include a Data availability statement as a separate section after Methods but before references, under the heading "Data Availability". For Springer Nature policies on data availability see <http://www.nature.com/authors/policies/availability.html>; for more information on this particular policy see <http://www.nature.com/authors/policies/data/data-availability-statements-data-citations.pdf>. The Data availability statement should include:

- Accession codes for primary datasets (generated during the study under consideration and designated as "primary accessions") and secondary datasets (published datasets reanalysed during the study under consideration, designated as

"referenced accessions"). For primary accessions data should be made public to coincide with publication of the manuscript. A list of data types for which submission to community-endorsed public repositories is mandated (including sequence, structure, microarray, deep sequencing data) can be found here <http://www.nature.com/authors/policies/availability.html#data>.

- Unique identifiers (accession codes, DOIs or other unique persistent identifier) and hyperlinks for datasets deposited in an approved repository, but for which data deposition is not mandated (see here for details <http://www.nature.com/sdata/data-policies/repositories>).
- At a minimum, please include a statement confirming that all relevant data are available from the authors, and/or are included with the manuscript (e.g. as source data or supplementary information), listing which data are included (e.g. by figure panels and data types) and mentioning any restrictions on availability.
- If a dataset has a Digital Object Identifier (DOI) as its unique identifier, we strongly encourage including this in the Reference list and citing the dataset in the Methods.

We recommend that you upload the step-by-step protocols used in this manuscript to protocols.io. More details can be found at <https://www.protocols.io/help/publish-articles>.

All imaging data should be accompanied by scale bars, which should be defined in the legend. Cropped images of gels/blots are acceptable, but need to be accompanied by size markers, and to retain visible background signal within the linear range (i.e. should not be saturated). The boundaries of panels with low background have to be demarked with black lines. Splicing of panels should only be considered if unavoidable, and must be clearly marked on the figure, and noted in the legend with a statement on whether the samples were obtained and processed simultaneously. Quantitative comparisons between samples on different gels/blots are discouraged; if this is unavoidable, it should only be performed for samples derived from the same experiment with gels/blots were processed in parallel, which needs to be stated in the legend.

- For line art, graphs, charts and schematics we prefer Adobe Illustrator (.AI), Encapsulated PostScript (.EPS) or Portable Document Format (.PDF). Files should be saved or exported as such directly from the application in which they were made, to allow us to restyle them according to our journal house style.
- We accept PowerPoint (.PPT) files if they are fully editable. However, please refrain from adding PowerPoint graphical effects to objects, as this results in them outputting poor quality raster art. Text used for PowerPoint figures should be Helvetica (preferred) or Arial.
- We do not recommend using Adobe Photoshop for designing figures, but we can accept Photoshop generated (.PSD or .TIFF) files only if each element included in the figure (text, labels, pictures, graphs, arrows and scale bars) are on separate layers. All text should be editable in 'type layers' and line-art such as graphs and other simple schematics should be preserved and embedded within 'vector smart objects' - not flattened raster/bitmap graphics.
- Some programs can generate Postscript by 'printing to file' (found in the Print dialogue). If using an application not listed above, save the file in PostScript format or email our Art Editor, Allen Beattie for advice (a.beattie@nature.com).

All placed images (i.e. a photo incorporated into a figure) should be on a separate layer and independent from any superimposed scale bars or text. Individual photographic images must be a minimum of 300+ DPI (at actual size) or kept

constant from the original picture acquisition and not decreased in resolution post image acquisition. All colour artwork should be RGB format.

The total number of Supplementary Figures (not including the "unprocessed scans" Supplementary Figure) should not exceed the number of main display items (figures and/or tables (see our Guide to Authors and March 2012 editorial <http://www.nature.com/ncb/authors/submit/index.html#suppinfo>; <http://www.nature.com/ncb/journal/v14/n3/index.html#ed>). No restrictions apply to Supplementary Tables or Videos, but we advise authors to be selective in including supplemental data.

GUIDELINES FOR EXPERIMENTAL AND STATISTICAL REPORTING

REPORTING REQUIREMENTS – We are trying to improve the quality of methods and statistics reporting in our papers. To that end, we are now asking authors to complete a reporting summary that collects information on experimental design and reagents. The Reporting Summary can be found here <https://www.nature.com/documents/nr-reporting-summary.pdf> or <https://www.nature.com/documents/nr-reporting-summary.pdf>. If you would like to reference the guidance text as you complete the template, please access these flattened versions at <http://www.nature.com/authors/policies/availability.html>.

We strongly recommend the presentation of source data for graphical and statistical analyses as a separate Supplementary Table, and request that source data for all independent repeats are provided when representative experiments of multiple independent repeats, or averages of two independent experiments are presented. This supplementary table should be in Excel format, with data for different figures provided as different sheets within a single Excel file. It should be labelled and

numbered as one of the supplementary tables, titled "Statistics Source Data", and mentioned in all relevant figure legends.

Version 2:

Decision Letter:

Our ref: NCB-A53469B

3rd October 2024

Dear Dr. Kraft,

Thank you for submitting your revised manuscript "Phase separation of initiation hubs on cargo is a trigger switch for selective autophagy" (NCB-A53469B). It has now been seen by the original Referees #2-3 and their comments are below. Rev#3 kindly commented on Rev#1's points and found your responses adequately addressed the previous points by Rev#1. The reviewers find that the paper has improved in revision. Rev#3 still did not find some of the data clear, and therefore we would ask that you further edit the text to make the conclusions from the movies but also throughout the text closer to the data, softening claims of fission and fusion and making it clear which conclusions are strongly supported by the data and which claims are more speculative.

Overall, based on the advice we have received, we'll be happy in principle to publish the manuscript in Nature Cell Biology, pending minor revisions to satisfy the referees' final requests and to comply with our editorial and formatting guidelines.

Please note that the current version of your manuscript is in a PDF format. Could you please email us a copy of the file in an editable format (Microsoft Word or LaTeX), as we can not proceed with PDFs at this stage? Thank you in advance.

Once we have the Word file, we will begin performing detailed checks on your paper and will send you a checklist detailing our editorial and formatting requirements in about 1-2 weeks. Please do not upload the final materials and make any revisions until you receive this additional information from us.

Thank you again for your interest in Nature Cell Biology. Please do not hesitate to contact me if you have any questions.

Sincerely,

Melina

Melina Casadio, PhD
Senior Editor, Nature Cell Biology
ORCID ID: <https://orcid.org/0000-0003-2389-2243>

Reviewer #2 (Remarks to the Author):

The authors have addressed my comments and I have no further points. I support acceptance of the manuscript.

Reviewer #3 (Remarks to the Author):

Regarding the observation of fission and fusion of Atg11 clusters, I remain unconvinced even with the newly added data. The morphological changes observed in vivo do not appear to represent fission or fusion events; they cannot be distinguished from Atg11 clusters merely moving around. I think the authors' response to Rev 1's comments is sufficient and that no further revisions are required.

Version 3:

Decision Letter:

Dear Dr Kraft,

I am pleased to inform you that your manuscript, "Phase separation of initiation hubs on cargo is a trigger switch for selective autophagy", has now been accepted for publication in Nature Cell Biology.

Please note that *Nature Cell Biology* is a Transformative Journal (TJ). Authors may publish their research with us through the traditional subscription access route or make their paper immediately open access through payment of an article-processing charge (APC). Authors will not be required to make a final decision about access to their article until it has been accepted. [Find out more about Transformative Journals](https://www.springernature.com/gp/open-research/transformative-journals)

If you have not already done so, we strongly recommend that you upload the step-by-step protocols used in this manuscript to protocols.io (<https://protocols.io>), an open online resource that allows researchers to share their detailed experimental know-how. All uploaded protocols are made freely available and are assigned DOIs for ease of citation. Protocols and Nature Portfolio journal papers in which they are used can be linked to one another, and this link is clearly and prominently visible in the online versions of both. Authors who performed the specific experiments can act as primary authors for the Protocol as they will be best placed to share the methodology details, but the Corresponding Author of the present research

paper should be included as one of the authors. By uploading your Protocols onto protocols.io, you are enabling researchers to more readily reproduce or adapt the methodology you use, as well as increasing the visibility of your protocols and papers. You can also establish a dedicated workspace to collect your lab Protocols. Further information can be found at <https://www.protocols.io/help/publish-articles>.

Nature Cell Biology encourages authors presenting evidence for cell, biological, molecular, and genetic interactions to consider communicating these findings using Biofactoid (<https://biofactoid.org/>). This tool helps users share a searchable representation of interactions (e.g. binding, gene expression, post-translational modification) between genes, gene products, or chemicals. Information added to Biofactoid, with author attribution, is shared on social media and public databases, such as Pathway Commons, where it can be discovered and analyzed in the context of a large and growing corpus of knowledge.

With kind regards,

Melina Casadio, PhD
Senior Editor, Nature Cell Biology
Consulting Editor, Nature Structural & Molecular Biology
ORCID ID: <https://orcid.org/0000-0003-2389-2243>

** Visit the Springer Nature Editorial and Publishing website at http://editorial-jobs.springernature.com?utm_source=ejp_NCB_email&utm_medium=ejp_NCB_email&utm_campaign=ejp_NCB for more information about our career opportunities. If you have any questions please click [here](mailto:editorial.publishing.jobs@springernature.com).

Point-by-point response to the reviewers

We thank all referees for their insightful comments, which have helped to improve our manuscript. Below, we detail how we have addressed each point raised and incorporated extensive new validation data to further support our conclusions.

Reviewer #1:

Remarks to the Author:

For starvation-induced autophagosome formation, core Atg proteins must gather to form the PAS on the vacuole, which is known to be mediated by liquid-liquid phase separation. However, it remained elusive how core Atg proteins behave during selective autophagy. In this manuscript, the authors studied Ape1 and END droplets as a model for selective autophagy cargo and showed that weak, but not strong interaction between the receptor and the cargo is important for selective autophagy. The authors then showed that Atg11 undergoes phase separation in vitro and in yeast cells and attaches to the cargo surface as puncta via strong interaction with the receptor on the cargo. Atg11 puncta coalesced to each other on the cargo to form “initiation hubs”, for which the mobility of receptors on the cargo was shown to be important. Moreover, the authors showed that the concept of initiation hubs established here can be applicable to selective autophagy in mammals.

The mechanism of selective autophagy initiation is one of the hottest topics in the field of autophagy and the proposed concept of initiation hubs is attractive. Moreover, the authors have constructed and studied a number of thoughtful experimental systems in order to strengthen their concept. On the other hand, the main conclusions are not sufficiently supported by the provided data as written below. It is essential to provide additional data that strengthen the main conclusions.

Major comments

1) The authors concluded that the affinity between Atg19 and Ape1 is low whereas that between Atg19 and Atg11 is high based on the GSH beads experiments. However, the used experiments are not quantitative and do not directly provide affinity information. To claim that Atg19-Atg11 interaction is stronger than Atg19-Ape1 interaction, determine each Kd value using ITC or SPR and compare the affinity quantitatively.

This is a great suggestion by the reviewer, which we had already followed beforehand. We had successfully purified the Atg11 C-terminus to a high degree of purity. However, we encountered challenges with Atg19, which did not reach the desired purity levels needed for the biophysical analysis to get the Kds. Additionally, the propeptide failed to express without a tag. We revisited these expressions during the revision and explored various purification strategies, including different fusion proteins. Despite these efforts, purifying the untagged propeptide proved impossible, likely due to its tendency to aggregate in vivo. These difficulties were confirmed also by colleagues, who had tried similar purifications before. Also, it has already been reported that prApe1 spontaneously forms aggregates depending on the propeptide (Yamasaki et al., Cell Reports 2016). Therefore, the requested experiments are not feasible.

To strengthen our findings, we used an alternative strategy. We compared the high-affinity Atg19/Atg11 interaction with the high-affinity GFP/GBP interaction, which was previously shown to have a Kd of 1 nM (Liu et al., NAR 2021). To engineer a low-affinity GFP/GBP interaction, we introduced mutations in GBP at the binding interface: The E104 mutation has been previously shown to raise the Kd to 1.4 μ M. In addition, we mutated F103, which according to the GFP/GBP structure (Liu et al., NAR 2021; Extended Data Fig. 1d) is a key amino acid of the interface and therefore should further weaken the interaction and increase the Kd. Using the fluorescence-based bead binding assay, GFP/GBP binding in wash-off experiments showed a similar behaviour as Atg19/Atg11, whereas GFP/GBP-F103 E104 (GBP-low) showed only very little binding, similar to the Atg19/PP interaction. These results further support our findings that the affinity of the Atg19/Atg11 interaction is high compared to the low-affinity interaction of Atg19/PP. These new results are shown in Fig. 1e.

Next, we introduced the double F103 E104 mutation in the Atg19-GBP fusion protein or in the 3xGBP construct used to solidify ENDS. While the wild-type GBP (high-affinity) blocked autophagic turnover of Ape1 or ENDS, the GBP mutants in both systems resulted in a rescue, further supporting the model that low-affinity interactions are key for autophagic degradation (new Fig. 1b, 1c, 1f and Extended Fig. 1b).

2) Figure 1F data are interesting. However, the complete block of the Cvt pathway by fusing GBP to Atg19 might be simply due to the loss of function of Atg19-GBP rather than the lowered mobility of the receptor. Perform control experiments expressing Atg19-GBP with mutation that impairs interaction with GFP and confirm that GFP-Ape1 is processed when expressing this Atg19-GBP mutant.

We thank the reviewer for pointing out this important control. As mentioned above we have now compared Atg19-GBP to the Atg19 fusion with the point mutant GBP-F103-E104. Indeed, Atg19-GBP-F103E104 restored autophagy functionality, supporting that the failure in autophagy turnover by Atg19-GBP stems from the high-affinity interaction with GFP-Ape1 (new Fig. 1f). Notably, the single E104 mutation, which lowers the GFP-GBP affinity from 1.0 nM to 1.4 μ M (Liu et al., NAR 2021), was insufficient to restore autophagy, suggesting that the Atg19-propeptide interaction has an even higher K_d than 1.4 μ M.

Similarly, Atg19-GBP was also functional when no GFP tag was present on Ape1 (new Extended Data Fig. 6b). Also, Atg19-GBP recruited Atg11 with similar efficiency as non-tagged Atg19, further supporting that Atg19-GBP is functional (Reviewer Figure 1). These findings demonstrate that the phenotype observed is not simply due to the loss of function of Atg19-GBP.

Reviewer Figure 1

3) From the Kymograph data in Figure 3B, the authors wrote that morphological changes of the Atg11 clusters resemble fusion and fission of individual clusters. However, the Kymograph data may just show the movement of Atg11 clusters on the Ape1 complex. Provide time-lapse imaging data and show that Atg11 clusters actually fuse to each other.

To address this point in more detail, we provide stills and time-lapse movies, in which one can clearly observe how certain clusters fuse and split again. These findings are shown as Supplementary Videos 1 and 2 and in Extended Data Fig. 2d, and support our findings shown by the kymograph analysis.

4) The authors performed *in vitro* phase separation experiments using GFP-Atg11 and showed that GFP-Atg11 alone undergoes phase separation to form droplets. However, it is not known whether Atg11 alone undergoes phase separation to form the PAS in cells. In the case of starvation-induced autophagy, Atg13 is essential for phase separation to form the PAS in addition to an Atg11 paralog Atg17. Study the effect of Atg13 deletion on the Atg11 puncta formation in cells.

In vitro, we observed that Atg11 requires a certain critical concentration to promote phase separation. *In vivo*, when Atg11 was expressed under its endogenous promoter, we did not observe Atg11 clusters independent of cargo, suggesting that this critical concentration is not reached in the cytosol, but is promoted by its local concentration on cargo via Atg19 recruitment (Extended Data Fig. 3c). To test whether Atg11 undergoes phase separation in cells also in the absence of cargo binding, we overexpressed Atg11 in cells lacking the cargo receptor Atg19 under nutrient-rich conditions. This led to the formation of Atg11 clusters, suggesting that Atg11 can phase separate independently of cargo once a critical concentration is reached. FRAP experiments photobleaching these Atg11 clusters showed a dynamic exchange with the cytosolic pool as expected for a liquid-like condensate (Fig. 3g). In addition, these clusters also form in the absence of Atg13 or Vac8, suggesting that Atg11 phase separation occurs independently of these factors (Fig. 3f).

5) Based on Figure 3E data, the authors wrote that GFP-Atg11 droplets coalesced to each other. However, in Figure 3E, the two droplets seem not to be in the same z section and the left droplet appears to just pass over the right one from 0 to 70 seconds. Provide time-lapse imaging data and show that GFP-Atg11 droplets actually coalesced to each other.

Indeed, droplets are often in different planes. This comes from the necessity to image at the surface of the coverslip, where droplets eventually settle. To address the point raised, we took time lapse images and we quantified the size increase over time, which is a measure for coalescence. When quantifying the diameter, a clear increase was observed. In addition we provide the movies, which further support the coalescence and not only attachment of the droplets. This data is shown in the new Fig. 3e, Extended Data Fig. 3a and Supplementary Videos 3-5.

6) Based on the data in Figure S2D, the authors wrote that Atg11-Atg19 interaction is high affinity. However, it was reported that Atg19 requires phosphorylation for high affinity interaction with Atg11 (PMID 24968893, 25287303). As written above, quantitative analysis is required in order to discuss the importance of affinity and so compare the K_d values between Atg11-Atg19, Atg11-phospho-Atg19, and Atg19-Ape1.

The reviewer raises the important point that Atg19 requires phosphorylation to bind with high affinity to Atg11. This phosphorylation can be mimicked by aspartate mutations of three serine residues, which provide full functionality even in the absence of the phosphorylating kinase (Pfaffenwimmer et al., EMBO R 2014). In all our binding assays with Atg11, we used this aspartate mutant Atg19-3D. We realize that this was not sufficiently explained in the text, and have adjusted the text accordingly.

As explained in point 1), it was unfortunately not possible to produce proteins of sufficient quality for affinity measurements. We therefore compare the affinities by comparing to GFP-GBP and GFP-GBP-F103E104, as explained in point 1).

7) Based on Figure S2D data, the authors wrote that "GFP-Atg11 formed bright clusters on the Atg19-decorated beads, suggesting phase separation in vitro". However, the bright clusters of GFP-Atg11 remained on beads even after washing, suggesting that the clusters are aggregates of GFP-Atg11 rather than phase-separated droplets. To claim that GFP-Atg11 clusters on the Atg19-decorated beads are phase-separated ones, perform FRAP experiments on them and confirm the fluorescence recovery as observed in Figure 3D.

As suggested by the reviewer, we performed FRAP analysis of the Atg11 structures on beads. Indeed, these structures recovered over time, supporting that they form condensates (new Extended Data Fig. 3g).

8) The authors developed a mathematical model and showed that low-affinity multivalent interactions between cargo and Atg11-Atg19 support the formation of initiation hubs (Figure 4A). This model assumes that the Atg11-Atg19 interaction is much stronger than that between Atg19 and Ape1. As written above, it is important to determine each K_d value to underline this mathematical model.

As explained in point 1) above, it was impossible to purify the individual proteins and perform ITC or MST measurements. Alternatively, we performed bead binding assays with GBP and GBP mutants with GFP, to compare the binding strengths (see point 1).

9) In Figure 4D, the authors provided only the cryo-EM image. Provide fluorescence image superimposed on the cryo-EM image in order to show that the phagophore is generated from the PAS on the vacuole.

As suggested by the reviewer, the fluorescence image superimpositions are now shown in the new Extended Data Fig. 5e.

10) In Figure 4E, the authors showed that co-expression of pp-GFP-uNS and Atg19 resulted in the autophagic degradation of pp-GFP-uNS whereas Atg19-GFP-uNS not. Moreover, co-expression of Cnb1-Atg19 and FKBP-GFP-uNS in the presence of FK506 did not result in the degradation of FKBP-GFP-uNS. In these experiments, there may be a possibility that only wild-type Atg19 is functional and that the affinity between Atg19 and cargo does not matter. It is necessary to show that Atg19-GFP-uNS or Cnb1-Atg19 is functional as a receptor, for example by studying the Ape1 delivery to the vacuole in *atg19Δ* cells expressing these fusion proteins.

Indeed, this is an important control raised by the reviewer. To test if Cnb1-Atg19 is functional as a receptor, we expressed this fusion protein in *atg19Δ* cells. In this case, Cnb1-Atg19 promoted Cvt pathway function comparable to endogenous Atg19, suggesting that the Cnb1 tag doesn't inhibit its receptor function (new Extended Data Fig. 6b).

Cnb1-Atg19 furthermore recruited Atg11 when tethered to FKBP-uNS. However as for the GBP/GFP stiffening this strongly reduced the clustering, suggesting a general functionality in recruiting Atg11 but impaired mobility (Extended Data Fig. 6d). Similarly, also Atg19-GFP-uNS was proficient in Atg11 recruitment, supporting that Atg19's receptor function is not impaired also in this fusion protein (Fig. 4g).

11) The authors concluded that “the mobility of receptors on selective cargo is essential for cargo degradability by selective autophagy, rather than the cargo property itself” in lines 311-312. However, the authors did not provide the data showing that the cargo property is not important for cargo degradation by selective autophagy. Cargo property should affect the mobility of receptors on the cargo and so this conclusion seems not accurate. If the authors do not provide the data about the effect of changing the cargo property on its selective autophagy efficiency, then reconsider the conclusion.

We agree with the reviewer that this statement might be misinterpreted. We intended to state that the key feature for degradability is the mobility of receptors on cargo. As cargo properties can influence this mobility, at least indirectly, we therefore removed “..rather than the cargo property itself”.

Minor comments

1) Provide images for Figure 2F as Supplementary.

As suggested by the reviewer we provided representative images for Figure 2F.

Reviewer #2:

Remarks to the Author:

Phase separation was shown to organize autophagosome formation sites and mediate autophagy cargo assembly in bulk autophagy and selective autophagy, respectively. In this manuscript, Licheva et al described that the scaffold protein Atg11 phase separate at the surface of cargos to trigger initiation hub formation in yeast. The phase separation of Atg11 is mediated by low affinity interactions between cargo and cargo receptors, and this low affinity interactions were shown to be required for phagophore initiation and subsequent cargo degradation. This study reveals that receptor mobility rather than the cargo allows Atg11 condensation and initiation hub formation, and followed autophagic degradation. Overall, this discovery will be of general interest to the field. However, the conclusions are not entirely convincing, more evidences need to be provided to strengthen the key findings.

Major concerns:

1) One key finding of this study is that low-affinity cargo-receptor interactions promote receptor mobility, which enables initial hub formation during selective autophagy in yeast. However, first, the author manipulated the interactions between cargo and receptors solely by artificially inducing an exogenous GFP binding protein into cells. Mutants on both cargo and receptor side that change the interaction affinity should be generated, and the effects of such mutants on cargo degradation and initial hub formation should be tested. Second, the authors only showed the dynamics of Ede1 without checking mobility of other receptors like Atg19 in different conditions. The cargo mobility also needs to be shown, it could be possible that interaction affinity also changes the cargo properties. Third, the authors used copper-inducible promoter in several experiments, but previous study shows that copper is important for ULK1 activity regulation (Tsang et al, 2020), which could also affect Atg1 in yeast, the authors should check the effect of copper in their experiments.

To address the point concerning the interaction affinity, we engineered a low-affinity GFP/GBP interaction. We introduced mutations in GBP at the binding interface: The E104 mutation has been previously shown to raise the Kd to 1.4 μ M. In addition, we mutated F103, which according to the GFP/GBP structure (Liu et al., NAR 2021, Extended Data Fig. 1d) is a key amino acid of the interface and therefore should further weaken the interaction and increase the Kd. We have now compared Atg19-GBP to the Atg19 fusion with the point mutant GBP-F103E104. Indeed, Atg19-GBP-F103E104 restored autophagy functionality, supporting that the failure in autophagy turnover by Atg19-GBP stems from the high-affinity interaction with GFP-Ape1 (new Fig. 1f). Notably, the single E104 mutation, which lowers the GFP-GBP affinity from 1.0 nM to 1.4 μ M (Liu et al., NAR 2021), was insufficient to restore autophagy, suggesting that the Atg19-propeptide interaction has an even higher Kd than 1.4 μ M.

Similarly, Atg19-GBP was also functional when no GFP tag was present on Ape1 (new Extended Data Fig. 6b). Also, Atg19-GBP recruited Atg11 with similar efficiency as non-tagged Atg19, further supporting that Atg19-GBP is functional (Reviewer Figure 1). These findings demonstrate that the phenotype observed is not simply due to the loss of function of Atg19-GBP.

Reviewer Figure 1

As suggested by the reviewer, we also monitored the mobility of fixed and native Atg19 receptors. As expected, fixing receptor mobility by GBP-GFP binding resulted in abolished FRAP recovery (new Extended Data Fig. 1g).

We also monitored cargo mobility by FRAP, which, however, was not affected by receptor binding or fixation (new Extended Data Fig. 1h).

Tsang et al., 2020 showed that 10 μ M copper resulted in an enhancement of ULK1 kinase activity and in turn an increased autophagy flux. To test if copper in yeast also enhances the autophagy flux, we treated growing yeast

cell with 50 and 250 μM copper sulfate, the concentrations used in our yeast assays. Ape1 processing was unaffected by the copper treatment, suggesting that up to 250 μM copper does not enhance the autophagy flux (new Figure S1a). It should be noted that the -3xGBP cells, that don't contain the 3xGBP expression construct, were also treated with copper sulfate, therefore an influence of copper independent of the 3xGBP expression is controlled for in this setup.

2) Another finding is phase separation of Atg11 drives initial hub formation. The current evidences are not sufficient to demonstrate that Atg11 is the driving force, the authors need to show that Atg11 condensates are capable to recruit other core autophagy proteins like Atg1, Atg9 *in vitro*. And mutants of Atg11 that cannot phase separate into condensates should be generated, and the initial hub formation and cargo degradation in cells expressing such mutants should be analyzed.

To demonstrate that Atg11 is the driving force capable of recruiting other Atg proteins, we monitored the interaction of recombinant Atg11 condensates with recombinant Vac8. Indeed, when GST-Vac8 was immobilized on beads, Atg11 condensates bound, but not to GST alone (new Extended Data Fig. 4c).

We were unable to obtain recombinant Atg9 and Atg1 to perform similar assays. However, proteins driving liquid-liquid phase separation are called scaffolds and should phase separate in a concentration-dependent manner. We therefore rationalized that overexpression of Atg11 in the absence of Atg19 should lead to condensate formation. Indeed, overexpression of Atg11 in *atg19 Δ* cells resulted in condensate formation *in vivo* (new Fig. 3f), which was independent of Vac8 and Atg13. As suggested by the reviewer, we tested if these condensates were proficient in recruiting Atg9. Indeed Atg9 was recruited to the Atg11 condensates, as monitored by fluorescence microscopy, supporting that Atg11 is the driving force of initiation hub formation (new Extended Data Fig. 3e).

Based on findings by Yorimitsu et al., 2005, who showed that CC2 and CC3 in Atg11 are required for self-interaction, we speculated that this self-interaction is required also for phase separation. Therefore we generated truncations lacking these domains. We used fusion constructs with Atg19, to ensure cargo binding. We previously showed that Atg11 amino acids 1-873, which only lacks the Atg19 binding region, when fused to Atg19 is fully functional (Torggler et al., Mol Cell 2016). This construct was also proficient in forming initiation hubs on Ape1 (new Extended Data Fig. 3d). When truncating CC3 (Atg11 amino acids 1-607), initiation hubs still formed, similar to Atg11_1-873. However, when truncating both CC2 and CC3 (Atg11 amino acids 1-454), initiation hub formation was strongly reduced, suggesting that these regions in Atg11 are required for phase separation (new Extended Data Fig. 3b and 3d).

As overexpression of wild type Atg11 promotes phase separation *in vivo* in the absence of cargo binding (Fig. 3f), we also tested Atg11_1-454 localization upon overexpression in *atg13 Δ atg19 Δ vac8 Δ* cells. Whereas full-length Atg11 proficiently formed condensates, Atg11_1-454 failed to do so and rather showed homogenous cytoplasmic staining (Extended Data Fig. 3c).

Together, these findings suggest that Atg11 via its CC2 and CC3 independent of Atg13 and Vac8 phase separates and these condensates subsequently recruit Atg9.

3) To demonstrate that initiation hubs for selective autophagy are conserved in human cells, the authors only showed ULK1 foci on ER surface upon mitophagy induction. Initiation hub formation around more mammalian cargos should be examined. Moreover, it is important to test whether the formation of initiation hubs in mammalian cells is also mediated by low-affinity interactions between cargo and cargo receptors, whether the principle that receptor mobility regulate selective autophagy initiation is conserved from yeast to human cells need to be confirmed.

We now addressed this point in more detail. As another selective cargo, we looked at p62 condensates. In our hands, mostly small p62 condensates of around 0.5-1 μm diameter formed in U2OS cells, which rapidly recovered upon photobleaching, supporting their fluid or semi-fluid state (Fig. 5c, Extended Data Fig. 7a). Next, we monitored FIP200 localization, which formed similar foci on p62 as found for Atg11 on Ape1 and ENDS (Fig. 6e). Upon starvation, also ULK1 formed foci on p62 condensates (Fig. 5c). As we found FIP200 to make the connection between mitochondria and the ER, we analyzed if FIP200 also connected p62 condensates to the ER. Indeed, fluorescence profile plots of FIP200, p62, and ER (Sec61 β) revealed FIP200 sitting between p62 and the ER, as expected for a connecting factor (Fig. 8a, Extended Data Fig. 7c). 3D reconstructions furthermore support these findings (Extended Data Fig. 7d and Supplementary Video 9), similar to our observations on mitochondria. Together, these findings support that initiation hubs form on different selective cargo also in mammals.

One reason we initially chose to use artificial tethering of ULK1 to the mitochondrial membrane was that the engineered mitochondrial tether FRB-FIS1, similar to a receptor, has a high mobility due to lateral diffusion in the

membrane. This mobility of FRB-Fis1 was now confirmed by FRAP experiments as ULK1 foci on mitochondria showed fast recovery after photobleaching (Extended Data Fig. 7e). To test if this mobility is required for initiation hub formation in mammalian cells, we created an oligomerization construct by expressing 2xFKBP-ULK1 and adding a FKBP-homodimerizing drug, AP20187, which results in oligomerization of ULK1. Upon homo-oligomerization and rapalog treatment, distinct FKBP-GFP-ULK1 foci formed along the mitochondrial network, resembling those observed without the homo-oligomerizer (new Fig. 8b). FRAP experiments confirmed the reduced mobility of these hubs. Importantly, a comparison of the mitophagy flux with and without homo-oligomerization showed a drastic reduction when the homo-oligomerizer was added to the cells (new Fig. 8C). Together, these findings support that the principle of receptor mobility is conserved in mammals.

Finally, additional supporting evidence for this avidity and mobility-mediated principle in selective autophagy comes from an independent study by the Mizushima lab (Yang et al. BioRxiv). In this study they observe that NDP52 and optineurin form condensates on the mitochondrial surface at distinct sites of phagophore grows, rather than being homogeneously distributed all around mitochondria. Although studied from a different angle of view, these observations are in line with our model.

Minor points:

1) Except for dynamics of Ede1, whether other properties of Ede1 condensates like number or size changed in the +3xGBP cells. It looks that the condensates were bigger in the +3xGBP cells in Fig S1A.

As suggested, we quantified the number and size of Ede1 condensates with and without 3xGBP. We could not observe any major changes. These quantifications are included in the new Extended Data Fig. 1c.

2) In Fig 1F, the authors need to show that the interaction of GFP-Ape1 peptide and Atg19-GBP is high affinity using a similar bead binding assay as they showed in Fig 1E.

To address the point of affinities between Ape1, PP and Atg19 in more detail, we now compared these interactions to that of the high-affinity interaction between GBP and GFP, and the low-affinity interaction of GFP with the point mutant GBP-F103E104 (GBP^{low}) both *in vivo* and in the *in vitro* bead binding assay. The bead binding assays are shown in Fig. 1e. We found that Atg19 was recruited to the Ape1 propeptide on GSH beads, but this interaction was lost after washing, consistent with a low-affinity interaction. Similarly, GFP was washed-off from GST-GBP^{low} bound beads, but not from GST-GBP bound beads (Fig. 1e). These findings support our *in vivo* findings, that receptor mobility in the CvT pathway is established by low-affinity receptor-cargo interactions, allowing a high on- and off binding rate.

3) In Fig 2 A i) and iii), is there a particular reason to use *atg19Δ* cells? would Atg11 form cluster around Ede1 condensates in wild type cells. Why did Atg11 form several clusters around Ape1 condensate, but only a single or even none around Ede1 condensates?

The reason for using *atg19Δ* cells was

1. to prevent Atg11 localization to Ape1 to exclude false positive events
2. to prevent competition of Atg11 binding to Ape1 cargo thereby reducing the frequency on ENDs.

We see Atg11 condensates on ENDs also in wild type cells, however, the frequency is lower likely due to the competition with Ape1 under rich conditions (see below). Therefore, we used *atg19Δ* cells in our experiments.

Reviewer Figure 2

We also see ENDs with multiple Atg11 foci (see below), but more often there is only one. This could be due to a different regulation of ENDs or to their higher mobility compared to Ape1. This will be an interesting aspect to investigate in future studies.

Reviewer Figure 3

4) Fig 2B was not correctly described in the main text.

To clarify the intention of Fig 2b, in the revised text, we now explain Figure 2b in more detail. We explain the comparison of GFP-Atg11 to Nup170-GFP, a protein known to form foci around the nucleus, and Vph1-GFP, a protein known to distribute rather homogenously around the vacuole.

5) In Fig 2E and 3A, the author showed that Atg11 clusters on Ape1 condensates are highly dynamic in normal cells, how about the dynamics of Atg11 in Atg19-GBP expressing cells? Whether Atg1 or Atg9 colocalize with the less-clustered Atg11 in Atg19-GBP expressing cells?

To address this point, we first measured mScarlet-Atg11 in Atg19-GBP GFP-Ape1 cells. However, we were unable to receive solid FRAP data due to the rapid photobleaching of mScarlet under the experimental conditions. As GFP was used to tether to GBP, a fluorophore switch to GFP was not possible. Therefore we switched to the μ NS setup: We compared GFP-Atg11 on PP- μ NS and on Atg19- μ NS. GFP-Atg11 formed clusters that were mobile on PP- μ NS, but not on Atg19- μ NS, as expected (new Extended Data Fig. 6c).

We also analyzed Atg1-mCherry localization on Atg19 wild type and Atg19-GBP containing GFP-Ape1 cells. Whereas Atg1 localized to initiation hubs in the case of wild type Atg19, only little Atg1 recruitment could be observed in Atg19-GBP mutants (Extended Data Fig. 2b).

6) The authors hypothesized that all the Atg11 clusters coalesce into a single initiation hub, but only showed that Atg8 and Atg9 formed a single focus upon rapamycin treatment in Fig 4, the localization of Atg11 upon rapamycin treatment need to be shown simultaneously.

To follow the reviewer's suggestion, we compared Atg11 cluster formation under nutrient-rich conditions and upon rapamycin treatment. As expected Atg11 does rearrange significantly to form intense clusters at the vacuolar contact site (Extended Data Fig. 5d).

Reviewer #3:

Remarks to the Author:

Cellular components to be degraded by autophagy (cargos) are sequestered within autophagosomes and delivered to lysosomes or vacuoles for degradation. Recent studies have revealed the involvement of phase separation in different events during autophagy, including that of cargo molecules and components of the autophagosome formation machinery (Atg proteins). In this study, Licheva et al. investigate the relationship between physical properties of cargos and receptors and the initiation of autophagosome formation via phase separation of related proteins. Based on the results, the authors provide several insights into the molecular mechanism of the initiation of autophagosome formation in selective autophagy, but most of the conclusions are not convincingly supported by the experimental data as described below in detail and therefore this work is too preliminary for publication.

Specific comments:

1. The authors showed that linking GFP-Ede1 molecules with BFP-3xGFP or linking Atg19 to Ape1 via the GBP-GFP interaction abolished vacuolar transport of Ede1 or Ape1, respectively, but it has not been shown that autophagosome formation is indeed hampered at the initiation step (before membrane expansion) as the authors assume.

To address if autophagosome formation is impaired at the initiation step, we monitored membrane elongation by monitoring mScarlet-Atg8 on Atg19 wild type or Atg19-GBP containing GFP-Ape1 cells. Only a few cells showed Atg8 localization to cargo in Atg19-GBP containing cells upon rapamycin treatment, which, in contrast to Atg19 wild type-containing cells, didn't elongate. These results suggest that autophagosome formation is impaired at the initiation step. We now show these findings in the new Fig. 4c.

2. Lines 120-122, "Despite their solidification, END assemblies still colocalized with the autophagy protein Atg8, suggesting that the autophagy receptor properties of Ede1 were not altered (Figure 1D and S1C).": To draw this conclusion, the authors should show that Atg8 localization to the END assemblies depends on the Ede1 AIM.

In previous work, we showed that Atg8 localization to ENDS depends on its AIM (Wilfling et al., Mol Cell 2020). We now tested if the Atg8 localization to 3xGBP-stiffened ENDS also depends on the AIM, which indeed was the case. These results are now shown in the revised Fig. 1d, Extended Data Fig. 1f).

3. Lines 122-124, "We infer that high-affinity interactions between 2xGFP-Ede1 and BFP-3xGBP render the receptor Ede1 immobile and that receptor mobility is required for selective END degradation.": It is unclear whether BFP-3xGFP indeed inhibited Ede1 degradation by decreasing its mobility. For instance, BFP-3xGFP incorporation may dilute 2xGFP-Ede1 in END assemblies and thereby decrease the initiation of autophagosome formation.

To address this point, we introduced point mutations in GBP that strongly reduce its affinity for GFP (GBP^{low}, GBP F103A E104R, Liu et al NAR 2021). Using these GBP mutants resulted in the restoration of END fluidity (new Fig. 1c) and END turnover despite targeting the GBP mutant to ENDS (new Fig. 1b, Extended Data Fig. 1b, 1e).

4. Fig. 1E: The authors should examine the GFP-GBP interaction in this assay to compare it with the Ape1 1-45-Atg19 interaction.

To address the point of affinities between Ape1, PP and Atg19 in more detail, we now compared these interactions to that of the high-affinity interaction between GBP and GFP, and the low-affinity interaction of GFP with the point mutant GBP-F103E104 both in vivo and the bead binding assay. The bead binding assays are shown in Fig. 1e. We found that Atg19 was recruited to the Ape1 propeptide on GSH beads, but this interaction was lost after washing, consistent with a low-affinity interaction. Similarly, GFP was washed-off from GST-GBP^{low} bound beads, but not from GST-GBP bound beads (Fig. 1e). These findings support our in vivo findings, that receptor mobility in the Cvt pathway is established by low-affinity receptor-cargo interactions, allowing a high on- and off binding rate.

5. Fig. 1F: Since GBP tagging of Atg19 itself could affect the efficiency of GFP-Ape1 degradation, a GBP mutant defective in GFP binding is necessary as a control.

We thank the reviewer for pointing out this important control. As mentioned above we have now compared Atg19-GBP to the Atg19 fusion with the point mutant GBP-F103E104. Indeed, Atg19-GBP-F103E104 restored autophagy functionality, supporting that the failure in autophagy turnover by Atg19-GBP stems from the high-

affinity interaction with GFP-Ape1 (new Fig. 1f). Notably, the single E104 mutation, which lowers the GFP-GBP affinity from 1.0 nM to 1.4 μ M (Liu et al., NAR 2021), was insufficient to restore autophagy, suggesting that the Atg19-propeptide interaction has an even higher Kd than 1.4 μ M.

Similarly, Atg19-GBP was also functional when no GFP tag was present on Ape1 (new Figure S6B). Also, Atg19-GBP recruited Atg11 with similar efficiency as non-tagged Atg19, further supporting that Atg19-GBP is functional (Reviewer Figure 1). These findings demonstrate that the phenotype observed is not simply due to the loss of function of Atg19-GBP.

Reviewer Figure 1

6. Line 156, "Because these clusters contain multiple factors that are involved in phagophore initiation, we termed them "initiation hubs".". These criteria are insufficient to define "initiation hubs" as distinguished from the PAS. The authors should set up experimental conditions for the observation of initiation hubs separately from the PAS, PAS scaffold, or phagophore and then analyze how different changes in related molecules affect their formation.

We consider initiation hubs to be a precursor of the PAS, forming multiple foci around the cargo to recruit autophagy machinery proteins such as Atg1 and Atg9. These hubs, however, are not yet capable of promoting phagophore formation. These multiple hubs then converge at the vacuole, forming the PAS, where further downstream factors such as the PI3K are recruited, which then allow phagophore nucleation. We now explain the difference between initiation hubs and the PAS more carefully in the revised text.

7. Lines 158-160, "We noticed that Atg11 clustering was reduced in cells with increased affinity of receptor-cargo interactions, such as the strains co-expressing Atg19-GBP with GFP-Ape1 (Figure 2E) and those co-expressing 2xGFP-Ede1 with BFP-3xGBP (Figure 2F).": The authors rely on the coefficient of variance of the fluorescence intensity to evaluate Atg11 clustering, but the actual image in Fig. 2E appears that enlargement of Atg11 clusters or an increase in Atg11 levels on the Ape1 surface resulted in an apparent decrease in Atg11 clustering. It is also required to observe the localization of Atg1 and Atg9 in these experiments.

The overall fluorescence intensity of Atg11 on cargo is not changed between Atg19 and Atg19-GBP containing cells. We show this quantification in the new Extended Data Fig. 2a. These findings support that Atg11 molecules distribute more evenly around the cargo rather than that more Atg11 is recruited.

We also localized Atg1-3xmCherry to these structures. Whereas Atg1 localizes with initiation hubs in the case of wild type Atg19, it hardly localizes to cargo in Atg19-GBP containing cells (Extended Data Fig. 2b.), suggesting that also this interaction is avidity mediated. We were unable to localize Atg9, due to the need of using red or blue fluorophores in this setup, which made Atg9 hardly visible at all.

8. Fig. 2E: The raw data (fluorescence microscope images) should be presented.

The FM pictures are provided for Fig. 2e. All pictures were taken with the same microscope settings and are presented with the same intensity range. We have now also analyzed the overall fluorescence intensity of Atg11 on cargo, which is unchanged (Extended Data Fig. 2a)

9. Lines 170-171, "These morphological changes resemble fusion and fission of individual clusters, consistent

with phase separation *in vivo*."; This statement comes from the results shown in Fig. 3B but does not make sense.

To address this point in more detail, we provide stills and time-lapse movies, in which one can clearly observe how certain clusters fuse and split again. These findings are shown as Supplementary Videos 1 and 2 and in Extended Data Fig. 2d, and support our findings shown by the kymograph analysis.

10. Lines 181-183, "GFP-Atg11 was efficiently recruited to GST-BFP-Atg19 immobilized on GSH beads and remained bound after subsequent washes, consistent with a high-affinity interaction between the scaffold and receptor (Figure S2D).": This assay without any controls for comparison does not provide reliable information on whether the affinity is high or low; the authors should directly compare the strength of the interactions in the same assay.

As suggested by the reviewer and explained in point 4), we have now compared these interactions to the known high-affinity interaction of GFP and GBP as well as to the low affinity interaction of GFP with the GBP-F103E104 mutant. These findings are shown in the new Fig. 1e and Extended Data Fig. 4c, and support our previous findings.

11. Lines 181-187, "Moreover, GFP-Atg11 formed bright clusters on the Atg19-decorated beads (Figure S2D), suggesting phase separation *in vitro*. Given that increasing the affinity of the Atg19 receptor for the Ape1 cargo reduced Atg11 cluster formation (Figure 2E), we propose that low-affinity Atg19 receptor-cargo interactions enable phase separation of Atg11-Atg19 complexes and initiation hub formation.": Does GST-BFP-Atg19 bind to GSH-beads with high affinity? If so, the results shown in Fig. S2D seem to contradict this proposal.

GST-tagged proteins bind with high affinity to GSH beads, which makes this setup widely used for affinity purifications. However, in this assay, we add recombinant GFP-Atg11 at the critical concentration for phase separation, which contains a mixture of soluble and condensed Atg11. This mixture is subsequently incubated with the Atg19 decorated GSH beads, and both soluble Atg11 and the condensates stably bind to Atg19 decorated beads. The purpose of this experiment is to compare the affinity of the Atg11/Atg19 interaction to the PP/Atg19 interaction.

This is different to the *in vivo* situation where we show that endogenously expressed Atg11 does not phase separate in the cytosol in absence of receptor due to not reaching a critical concentration (current Fig. 3c). Currently, we suggest that the mobility of Atg19 on cargo allows the local concentration of Atg11 to be increased above the critical concentration to start the condensation process on cargo. This notion is supported by new experiments showing that in the absence of Atg19, overexpression of Atg11 does lead to phase separation (new Extended Data Fig. 3e). Therefore, the results shown in former Figure S2D now Extended Data Fig. 3f are not contradictory but support our model. These clusters remain stable for some time also after wash-off experiments and maintain their phase-separated character (new Extended Data Fig. 3g).

12. Did the authors compare the two cases, in which the one was given low-affinity and high avidity and the other was given only high-affinity in mathematical modeling analysis? If so, can they attribute Atg11 clustering in the former to the difference in the affinity?

We thank the reviewer for this comment and realize that we didn't explain well enough what we modeled. This is exactly what the current model shows. In general, low affinity interactions can lead to an overall low binding strength, or if multiple low affinity interactions accumulate, this can lead to avidity which results in an overall high binding strength. High affinity is always also high binding strength. We have now adjusted the figure to better explain these situations. We also show now a third case, in which very-low affinity does not result in avidity and therefore results in an overall low binding strength (new Fig. 4a and Supplementary Videos 6-8).

13. Fig. S3: The conclusion that Vac8 colocalizes with Atg11 clusters is not convincing, since Vac8 does not appear to form discrete puncta on the vacuolar membrane in the fluorescence images. The kymograph was made by analyzing the fluorescence intensity along the Ape1 surface, and therefore the fluorescence of Vac8 and Atg11 appears to match but this would not be regarded as their colocalization.

We have previously characterized this in more detail and shown that Vac8 enriches at Atg11 contact sites in live cells (Hollenstein et al., Nat Com 2021). This enrichment is not simple to see for native cargo, however, for large cargo with a high amount of Atg11, it is very clear (Hollenstein et al., Nat Com 2021). However, the point we want to make here is not that Vac8 clusters, but rather that Atg11 enriches at the sites on the cargo, at which the cargo

is in contact with the vacuole via Vac8. We state this now more clearly in the revised version of the manuscript. We have furthermore strengthened this observation by showing that Atg11 condensates bind to recombinant Vac8 (Extended Data Fig. 4c).

14. Fig. 4D: The fluorescence image should also be shown.

As suggested by the reviewer, the fluorescence image superimpositions are now shown in the new Extended Data Fig. 5e.

15. Figs. 4E, 4F, S4D, S4E: In these experiments, the authors compared the proteins with quite different architectures and therefore it is difficult to attribute changes in degradation and Atg11 clustering to their difference in cargo-receptor affinity. It would be better to compare proteins with an essentially same structure but containing mutations that are expected to affect the affinity and after experimentally confirming the affinity use them for these experiments.

We have now compared Atg19-GBP to the Atg19 fusion with the point mutant GBP-F103-E104, which lowers the affinity. Indeed, Atg19-GBP-F103E104 restored autophagy functionality, supporting that the failure in autophagy turnover by Atg19-GBP stems from the high-affinity interaction with GFP-Ape1 (new Fig. 1f). Notably, the single E104 mutation, which lowers the GFP-GBP affinity from 1.0 nM to 1.4 μ M (Liu et al., NAR 2021), was insufficient to restore autophagy, suggesting that the Atg19-propeptide interaction has an even higher K_d than 1.4 μ M.

Similarly, Atg19-GBP was also functional when no GFP tag was present on Ape1 (new Extended Data Fig. 6b). Also, Atg19-GBP recruited Atg11 with similar efficiency as non-tagged Atg19, further supporting that Atg19-GBP is functional (Reviewer Figure 1). These findings demonstrate that the phenotype observed is not simply due to the loss of function of Atg19-GBP.

To test if Cnb1-Atg19 is functional as a receptor, we expressed this fusion protein in *atg19 Δ* cells. In this case, Cnb1-Atg19 promoted Cvt pathway function, suggesting that the Cnb1 tag doesn't inhibit its receptor function (new Extended Data Fig. 6b). Cnb1-Atg19 furthermore recruited Atg11 when tethered to FKBP-uNS, further supporting its receptor functionality (Extended Data Fig. 6d). Similarly, also Atg19-GFP-uNS was proficient in Atg11 recruitment, supporting that Atg19's receptor function is not impaired also in this fusion protein (Fig. 4g).

16. Initiation hubs should also be more carefully defined in mammalian cells similar to the case in yeast as described above.

We now performed several additional experiments in mammalian cells.

As another selective cargo, we looked at p62 condensates. In our hands, mostly small p62 condensates of around 0.5-1 μ m diameter formed in U2OS cells, which rapidly recovered upon photobleaching, supporting their fluid or semi-fluid state (Extended Data Fig. 7a). Next, we monitored FIP200 localization, which formed similar foci on p62 as found for Atg11 on Ape1 and ENDS (Fig. 6e). Upon starvation, also ULK1 formed foci on p62 condensates (Fig. 5c)

As we found FIP200 to make the connection between mitochondria and the ER, we analyzed if FIP200 also connected p62 condensates to the ER. Indeed, fluorescence profile plots of FIP200, p62, and ER (Sec61b) revealed FIP200 sitting between p62 and the ER, as expected for a connecting factor (Fig. 8a and Extended Data Fig. 7c). 3D reconstructions furthermore support these findings (Extended Data Fig. 7d and Supplementary Video 9), similar to our observations on mitochondria.

Together, these findings support that initiation hubs form on different selective cargo also in mammals.

One reason we initially chose to use artificial tethering of ULK1 to the mitochondrial membrane was that the engineered mitochondrial tether FRB-FIS1, similar to a receptor, has a high mobility due to lateral diffusion in the membrane. This mobility of FRB-Fis1 was now confirmed by FRAP experiments as ULK1 foci on mitochondria showed fast recovery after photobleaching (Extended Data Fig. 7e). To test if this mobility is required for initiation hub formation in mammalian cells, we created an oligomerization construct by expressing 2xFKBP-ULK1 and adding a FKBP-homodimerizing drug, AP20187, which results in oligomerization of ULK1. Upon homo-oligomerization and rapalog treatment, distinct FKBP-GFP-ULK1 foci formed along the mitochondrial network, resembling those observed without the homo-oligomerizer (new Fig. 8b). FRAP experiments confirmed the reduced mobility of these hubs. Importantly, a comparison of the mitophagy flux with and without homo-oligomerization showed a drastic reduction when the homo-oligomerizer was added to the cells (new Fig. 8c). Together, these findings support that the principle of receptor mobility is conserved in mammals.

Finally, additional supporting evidence for this avidity and mobility-mediated principle in selective autophagy comes from an independent study by the Mizushima lab (Yang et al. BioRxiv). In this study they observe that NDP52 and optineurin form condensates on the mitochondrial surface at distinct sites of phagophore growth, rather than being homogeneously distributed all around mitochondria. Although studied from a different angle of view, these observations are in line with our model.

Together, these experiments strongly support that initiation hub formation is conserved in mammals.

Reviewer #1:

Remarks to the Author:

The authors have addressed many of my concerns. However, one major concern is left unresolved. Due to the technical difficulties, the authors failed to determine the K_d values between Atg19 and Ape1 and between Atg19 and Atg11 and instead they compared the Atg19-Ape1 and Atg19-Atg11 interactions with GFP-GBP ($K_d=1\text{nM}$) and GFP-GBP mutant interactions (K_d is presumably larger than $1.4\mu\text{M}$; the actual affinity should also be determined) using the GSH beads experiments. However, this reviewer has a strong concern about the GSH beads experiments. In the rebuttal letter, the authors wrote that "GFP/GBP binding in wash-off experiments showed a similar behaviour as Atg19/Atg11, whereas GFP/GBP-F103 E104 (GBP-low) showed only very little binding, similar to the Atg19/PP interaction". If this is correct, then the Atg19-Atg11 interaction is an extremely strong interaction of $K_d\sim 1\text{nM}$, and the Atg19-Ape1 interaction is weaker than $K_d=1.4\mu\text{M}$. However, a previous study purified the coiled-coil of Atg19 complexed with the propeptide (residues 1-20) of Ape1 using size-exclusion chromatography (SEC) and determined the crystal structure of the complex at high resolution (PMID 27320913). Stable complex formation during SEC suggests that the affinity is sufficiently high and the K_d value should not be higher than $1.4\mu\text{M}$. The affinity between Atg19 and Atg11 has not been reported, but in mammals the affinity between autophagic receptors-FIP200 (Atg11 homolog) has been extensively studied, which showed the K_d values in the order of $1\sim 100\mu\text{M}$ (even the phosphorylated (affinity-enhanced) CCPG1 FIR and Optineurin LIR gave the K_d value of $0.7\mu\text{M}$ and $11\mu\text{M}$, respectively, to FIP200 (PMID 33692357)). Therefore, it is not reliable to use the GSH beads experiment as an estimate of affinity. Because the purification method of Atg19 coiled-coil and Ape1 propeptide (1-20) has already been reported (PMID 27320913), it should be not difficult to measure their affinity by SPR or ITC or FP-based assays. The affinity between Atg19 and Atg11 must also be determined. Because the conclusion that the affinity between Atg19 and Atg11 is much higher than that between Atg19 and Ape1 is a major point in this manuscript, this conclusion must be supported by solid data. To make the matters worse, if this conclusion is different, several other experiments will become invalid since they are based on this (e.g., molecular simulation).

We appreciate the reviewer's insightful feedback. After considering this input, we now recognize that our initial statements may have been misleading. We apologize for that and appreciate the opportunity to clarify our logic.

In the first part of the manuscript, our goal was to show that increasing the affinity between Ape1 and Atg19 impedes autophagy. We did this by co-expressing GFP-Ape1 with Atg19 fused either to wild-type GBP (an ectopic high-affinity interaction) or to mutant GBP (GBP^{low}, an ectopic low-affinity interaction, as suggested by Reviewer 3). GFP-GBP affinities have been described previously (Liu, J. *et al. Nucleic Acids Res.* 2021 PMID: 34255844) and we also show that we can detect the GFP-GBP interaction *in vitro*, whereas we cannot detect the GFP-GBP^{low} interaction *in vitro* (Fig 1e). Importantly, we show that autophagy was inhibited in cells expressing GFP-Ape1 /Atg19-GBP and was restored in cells expressing GFP-Ape1 /Atg19-GBP^{low} (Fig 1f). We also show that initiation hub formation of Atg11 is inhibited in cells expressing GFP-Ape1 /Atg19-GBP (Fig 2e).

We agree with the reviewer that estimating absolute affinities with the GSH bead experiment is not reliable, and we will tone this aspect down. We use this assay mainly to show that GFP interacts with GBP but not with GBP^{low} *in vitro*, as expected, but do not make a claim about specific affinities. We also show that we can detect an interaction between Atg19-Atg11 by this assay, but not between Ape1-Atg19 (Fig S3f and 1e).

The key point, in our mind, is that a high-affinity interaction (GFP-Ape1 /Atg19-GBP) impedes the formation of initiation hubs and inhibits autophagy in cells and that this is rescued in cells by the low-affinity interaction (GFP-Ape1 /Atg19-GBP^{low}). These data suggest that a

high-affinity receptor-cargo interaction blocks autophagy. Further support for this conclusion is provided by using the same GFP-GBP approach to analyze additional autophagy cargo (Ede1, Fig 1b,c and S1b), as well as high/low affinities on artificial cargo (μ NS, Fig 4f,g and S6a,c).

Although we agree with Reviewer 1 that measuring Kds would be valuable, we think this kind of analysis could shift the emphasis of the paper towards the *in vitro* context, which likely is an oversimplification of the process *in vivo*. For instance, these proteins interact in a multivalent manner *in vivo*, and measuring the Kds of fragments that instead bind in a 1:1 ratio may not accurately reflect the *in vivo* context. The previous study of the purified coiled-coil fragment of Atg19 complexed with the propeptide (residues 1-20) fragment of Ape1 (Yamasaki et al., Cell Rep. 2016 PMID 27320913) provides striking structural insight into the interaction, but to what extent the affinity of these fragments would represent the full-length proteins, *in vitro* or *in vivo*, is unclear. Determining absolute affinities for multivalent interactions is technically challenging and highly variable (as illustrated in Amacher et al., JCB 2020, Figure 7A PMID: 32289118). We would therefore like to suggest that these experiments are beyond the scope of this study, though it remains an interesting question for future investigation.

We have however included a co-IP comparison, in which we compare Atg19 binding to either the propeptide or the Atg11-Cterminus. In this setup, Atg19 stably bound to the C-terminal fragment of Atg11 immobilized on GSH beads but showed weaker binding and was largely washed off from GST-BFP-Ape1¹⁻⁴⁵ coated beads (Fig. 1f), similar to our observations by microscopy.

We believe that the experiments presented, along with the new *in vivo* experiments modulating affinities, strongly support the main conclusions of the paper.

Modified paragraph:

To determine if receptor mobility is an important feature in selective autophagy more generally, we manipulated the Cvt pathway. First, we used a pull-down assay to examine the interaction between Ape1 cargo and the Atg19 receptor. This assay detects the high-affinity interaction between GFP and GST-BFP-GBP, but not the low-affinity interaction between GFP and GST-BFP-GBP^{low} (Fig. 1e). Subsequently, we immobilized a GST-BFP-tagged Ape1 propeptide (Ape1 residues 1-45, which interacts with Atg19²⁰). After washing, we did not detect the interaction between mCherry-Atg19 and GST-BFP-Ape1¹⁻⁴⁵. Similarly, Atg19 bound stably to a C-terminal fragment of Atg11 immobilized on GSH beads (GST-BFP-Atg11⁶⁸⁵⁻¹¹⁷⁸), but showed substantially reduced binding to GST-BFP-Ape1¹⁻⁴⁵ coated beads in co-immunoprecipitation experiments (Fig. 1f), consistent with a low-affinity interaction between Ape1 and Atg19. This suggests that receptor mobility in the Cvt pathway is established by low-affinity receptor-cargo interactions.

To examine the impact of cargo-receptor affinity on degradation, we co-expressed GFP-Ape1 with either untagged Atg19 or GBP-tagged Atg19 in atg19 Δ cells. This is expected to change in the case of the Atg19-GBP to a high-affinity (ectopic) interaction compared to the endogenous interaction. Strikingly, free GFP was generated in atg19 Δ cells co-expressing GFP-Ape1 and untagged Atg19, but strongly reduced in those co-expressing GFP-Ape1 with Atg19-GBP (Fig. 1g). Moreover, expression of Atg19-GBP^{low} restored free GFP generation, further suggesting that the high-affinity interaction of Atg19-GBP with GFP-Ape1 inhibits its delivery to the vacuole and autophagic turnover. Notably, the single E104 mutation (GBP^{med}), was not sufficient to restore autophagic Ape1 delivery to the vacuole to the extent of the wild type (Fig. 1g).

Reviewer #2:

Remarks to the Author:

In the revised manuscript, the authors provided additional data to support the main conclusion that the low-affinity cargo-receptor interaction is essential for initial hub formation and cargo degradation. Most of the reviewer's comments have been addressed, however, the evidences that support the importance of low cargo receptor mobility and initial hub formation is conserved in mammals are still not strong enough. To confirm the formation of initial hub in mammalian cells, they showed p62 condensates could recruit FIP200, which is kind of expected, as a direct interaction between p62 and FIP200 was reported previously. They stated that FIP200 could connect p62 condensates to ER, but the data are not robust and quantitative. To confirm the low mobility is required for initiation hub formation in mammalian cells, the authors manipulated the mobility of FRB-Fis1, but it would be better to use cargo receptor mutants that changed the mobility to do the experiments. Besides, only a working model in yeast was showed in Fig8e, a universal model of selective autophagy should include both yeast and mammals.

The Cargo-ER connection has been quantitatively assessed by mass spectrometry (Fig. 7c,d,e). We now added a further quantification to Fig. 7f.

To address whether cargo mobility affects its turnover, we co-expressed 2xGFP-p62 with and without 3xGBP. Whereas 2xGFP-p62 shows fast recovery in FRAP and is turned over upon starvation, the co-expression of 3xGBP largely abolishes this, supporting our findings in yeast (new Fig. 8e).

We also adapted the model to a more universal model, as suggested, and revised the text to ensure our observations are accurately represented without overstating the findings.

Reviewer #3:

Remarks to the Author:

The authors have addressed most of the concerns I raised in the review of the original manuscript, but I request the authors further clarify the following points:

>> 9. Lines 170-171, "These morphological changes resemble fusion and fission of individual clusters, consistent with phase separation *in vivo*."; This statement comes from the results shown in Fig. 3B but does not make sense.

> To address this point in more detail, we provide stills and time-lapse movies, in which one can clearly observe how certain clusters fuse and split again. These findings are shown as Supplementary Videos 1 and 2 and in Extended Data Fig. 2d, and support our findings shown by the kymograph analysis.

It is not clear at all what parts in the still images and movies represent fusion and split of Atg11 clusters. It appears that clusters are just moving around.

Our observation of fission and fusion events *in vivo* provides the first direct evidence suggesting that Atg11 exhibits liquid-like behavior, which is subsequently validated through additional experiments including *in vitro* reconstitutions. To further substantiate this first observation, we now present several example movies focusing on zoomed-in areas of Ape1, capturing both fission and fusion events of Atg11 condensates. These are accompanied by still images and intensity plots. In the movies and plots, fusion is evident as parts of clusters merge, marked by an increase in intensity, for example when two distinct peaks combine into one higher-intensity peak. Similarly, fission events are observable when a single peak splits into two. We have also revised the text to ensure our observations are accurately represented:

*"Some of these morphological changes may resemble fusion and fission of individual clusters, suggesting a liquid-like behavior of Atg11 *in vivo*"*

>> 11. Lines 181-187, "Moreover, GFP-Atg11 formed bright clusters on the Atg19-decorated beads (Figure S2D), suggesting phase separation *in vitro*. Given that increasing the affinity of the Atg19 receptor for the Ape1 cargo reduced Atg11 cluster formation (Figure 2E), we propose that low-affinity Atg19 receptor-cargo interactions enable phase separation of Atg11-Atg19 complexes and initiation hub formation.": Does GST-BFP-Atg19 bind to GSH-beads with high affinity? If so, the results shown in Fig. S2D seem to contradict this proposal.

> GST-tagged proteins bind with high affinity to GSH beads, which makes this setup widely used for affinity purifications. However, in this assay, we add recombinant GFP-Atg11 at the critical concentration for phase separation, which contains a mixture of soluble and condensed Atg11. This mixture is subsequently incubated with the Atg19 decorated GSH beads, and both soluble Atg11 and the condensates stably bind to Atg19 decorated beads. The purpose of this experiment is to compare the affinity of the Atg11/Atg19 interaction to the PP/Atg19 interaction.

I would like to confirm with the authors that they removed the descriptions like "GFP-Atg11 formed bright clusters on the Atg19-decorated beads, suggesting phase separation *in vitro*" in the revised manuscript, which did not correctly describe the experimental settings and was therefore misleading.

In addition, the authors should clearly describe in the main text that they used recombinant Atg11 highly concentrated to form Atg11 clusters in the absence of Atg19 and observed binding of preformed Atg11 clusters to Atg19-decorated beads, to avoid readers' misunderstanding that Atg11 is phase-separated on the beads.

We have adjusted this text as indicated below to make clear that Atg11 condensates are recruited. The statement of "bright clusters...suggesting phase separation..." has been removed, as requested.

"Recombinant, preformed GFP-Atg11 condensates were efficiently recruited to GST-BFP-Atg19^{3D}, a phospho-mimetic mutant of Atg19 known to stably interact with Atg11²⁹, when immobilized on GSH beads. GFP-Atg11 remained bound after subsequent washes, consistent with a high-affinity interaction between the scaffold and receptor (Extended Data Fig. 3f). Moreover, GFP-Atg11 condensates on the Atg19^{3D}-decorated beads recovered upon photobleaching (Extended Data Fig. 3f, 3g), supporting its phase separation. Given that increasing the affinity of the Atg19 receptor for the Ape1 cargo reduced Atg11 cluster formation (Fig. 2e), we propose that low-affinity Atg19 receptor-cargo interactions enable phase separation of Atg11-Atg19 complexes and initiation hub formation."

Reviewer #2:

The authors have addressed my comments and I have no further points. I support acceptance of the manuscript.

Reviewer #3:

Regarding the observation of fission and fusion of Atg11 clusters, I remain unconvinced even with the newly added data. The morphological changes observed in vivo do not appear to represent fission or fusion events; they cannot be distinguished from Atg11 clusters merely moving around. I think the authors' response to Rev 1's comments is sufficient and that no further revisions are required.

To address the remaining reviewer's concern on the fission and fusion events, we have balanced the text further, highlighting the mobile aspect of Atg11 foci, without claiming fission and fusion associated with Fig. 3b and the videos:

Phase separation of Atg11 drives initiation hub formation

*Intriguingly, FRAP analysis revealed less mobility of the cargo GFP-Ape1 when compared to that of GFP-Atg11 (Fig. 3a). Time-lapse microscopy analysis of GFP-Atg11 revealed that most of the Atg11 clusters on the surface of the Ape1 complex dynamically change their size and morphology (Fig. 3b, Extended Data Fig. 2d, Supplementary Videos 1-12). ~~Some of these morphological changes may resemble fusion and fission of individual clusters, suggesting a liquid-like behaviour of Atg11 in vivo.~~ To investigate **these morphological changes** in more detail, we purified GFP-Atg11 from insect cells and found that it formed round droplets in the presence of physiological salt concentrations, **resembling a liquid-like behavior** (Fig. 3c, Extended Data Fig. 2e). Droplet formation was absent at low protein concentrations but became visible at around 0.05 μM , and droplet size increased with increased concentration of Atg11. FRAP experiments showed that GFP-Atg11 is largely mobile within the droplet (Fig. 3d). In addition, we observed coalescence of individual droplets, another typical feature of phase-separating proteins (Fig. 3e, Extended Data Fig. 3a, Supplementary Video 13-15).*